# EXPRESSIVE: A SPATIO-FUNCTIONAL EMBEDDING FOR KNOWLEDGE GRAPH COMPLETION

**Aleksandar Pavlović & Emanuel Sallinger**
Research Unit of Databases and Artificial Intelligence
TU Wien
Vienna, Austria
`{aleksandar.pavlovic,emanuel.sallinger}@tuwien.ac.at`

## ABSTRACT

Knowledge graphs are inherently incomplete. Therefore substantial research has been directed toward knowledge graph completion (KGC), i.e., predicting missing triples from the information represented in the knowledge graph (KG). KG embedding models (KGEs) have yielded promising results for KGC, yet any current KGE is incapable of: (1) *fully* capturing vital *inference patterns* (e.g., composition), (2) *capturing* prominent patterns *jointly* (e.g., hierarchy and composition), and (3) providing an *intuitive interpretation* of captured patterns. In this work, we propose *ExpressivE*, a fully expressive *spatio-functional* KGE that solves all these challenges simultaneously. ExpressivE embeds pairs of entities as *points* and relations as *hyper-parallelograms* in the virtual triple space $\mathbb{R}^{2d}$. This model design allows ExpressivE not only to capture a rich set of inference patterns jointly but additionally to display any supported inference pattern through the spatial relation of hyper-parallelograms, offering an intuitive and consistent geometric interpretation of ExpressivE embeddings and their captured patterns. Experimental results on standard KGC benchmarks reveal that ExpressivE is competitive with state-of-the-art KGEs and even significantly outperforms them on WN18RR.

## 1 INTRODUCTION

Knowledge graphs (KGs) are large collections of triples $r_i(e_h, e_t)$ over relations $r_i \in \mathbf{R}$ and entities $e_h, e_t \in \mathbf{E}$ used for representing, storing, and processing information. Real-world KGs such as Freebase (Bollacker et al., 2007) and WordNet (Miller, 1995) lie at the heart of numerous applications such as recommendation (Cao et al., 2019), question answering (Zhang et al., 2018), information retrieval (Dietz et al., 2018), and natural language processing (Chen & Zaniolo, 2017).

**KG Completion**. Yet, KGs are inherently *incomplete*, hindering the immediate utilization of their stored knowledge. For example, 75% of the people represented in Freebase lack a nationality (West et al., 2014). Therefore, much research has been directed toward the problem of automatically inferring missing triples, called *knowledge graph completion* (KGC). KG embedding models (KGEs) that *embed* entities and relations of a KG into latent spaces and quantify the plausibility of unknown triples by computing scores based on these learned embeddings have yielded promising results for KGC (Wang et al., 2017). Moreover, they have shown excellent knowledge representation capabilities, concisely capturing complex graph structures, e.g., entity hierarchies (Nickel & Kiela, 2017).

**Inference Patterns**. Substantial research has been invested in understanding which KGEs can capture which *inference patterns*, as summarized in Table 1. For instance, KGEs such as TransE (Bordes et al., 2013) and RotatE (Sun et al., 2019) can capture fundamental patterns such as composition. Recently, however, it was discovered that these two models can only capture a fairly limited notion of composition (Zhang et al., 2019; Abboud et al., 2020; Lu & Hu, 2020; Gao et al., 2020), cf. also Appendix K.1. Thus, multiple extensions have been proposed to tackle some of these limitations, focusing, e.g., on modeling non-commutative composition (Lu & Hu, 2020; Gao et al., 2020). Yet, while these extensions solved some limitations, the purely functional nature of TransE, RotatE, and any of their extensions still limits them to capture solely *compositional definition*, not *general composition* (see Table 1 for the defining formulas, and cf. also Appendix K.1 for details).

Therefore, capturing general composition is still an open problem. Even more, composition patterns describe paths, which are fundamental for navigation within a graph. Hence, the ability to capture general composition is vital for KGEs. In contrast, approaches such as SimplE (Kazemi & Poole, 2018), ComplEx (Trouillon et al., 2016), and BoxE (Abboud et al., 2020) have managed to capture other vital patterns, such as hierarchy, yet are unable to capture any notion of composition.

Table 1: This table lists patterns that several KGEs can capture. Specifically, ✓ represents that the pattern is supported and ✗ that it is not supported. Furthermore, "Comp. def." stands for compositional definition and "Gen. comp." for general composition.

| Inference Pattern | ExpressivE | BoxE | RotatE | TransE | DistMult | ComplEx |
|---|---|---|---|---|---|---|
| Symmetry: $r_1(X,Y) \Rightarrow r_1(Y,X)$ | ✓ | ✓ | ✓ | ✗ | ✓ | ✓ |
| Anti-symmetry: $r_1(X,Y) \Rightarrow \neg r_1(Y,X)$ | ✓ | ✓ | ✓ | ✓ | ✗ | ✓ |
| Inversion: $r_1(X,Y) \Leftrightarrow r_2(Y,X)$ | ✓ | ✓ | ✓ | ✓ | ✗ | ✓ |
| Comp. def.: $r_1(X,Y) \wedge r_2(Y,Z) \Leftrightarrow r_3(X,Z)$ | ✓ | ✗ | ✓ | ✓ | ✗ | ✗ |
| Gen. comp.: $r_1(X,Y) \wedge r_2(Y,Z) \Rightarrow r_3(X,Z)$ | ✓ | ✗ | ✗ | ✗ | ✗ | ✗ |
| Hierarchy: $r_1(X,Y) \Rightarrow r_2(X,Y)$ | ✓ | ✓ | ✗ | ✗ | ✓ | ✓ |
| Intersection: $r_1(X,Y) \wedge r_2(X,Y) \Rightarrow r_3(X,Y)$ | ✓ | ✓ | ✓ | ✓ | ✗ | ✗ |
| Mutual exclusion: $r_1(X,Y) \wedge r_2(X,Y) \Rightarrow \bot$ | ✓ | ✓ | ✓ | ✓ | ✓ | ✓ |

**Challenge**. While the extensive research on composition (Bordes et al., 2013; Sun et al., 2019; Zhang et al., 2019; Lu & Hu, 2020) and hierarchy (Yang et al., 2015a; Trouillon et al., 2016; Kazemi & Poole, 2018; Abboud et al., 2020) highlights their importance, any KGE so far is incapable of: (1) capturing general composition, (2) capturing composition and hierarchy jointly, and (3) providing an intuitive geometric interpretation of captured inference patterns.

**Contribution**. This paper focuses on solving all the stated limitations simultaneously. In particular:

- We introduce the *spatio-functional* embedding model **ExpressivE**. It embeds pairs of entities as points and relations as *hyper-parallelograms* in the space $\mathbb{R}^{2d}$, which we call the *virtual triple space*. The virtual triple space allows ExpressivE to represent patterns through the spatial relationship of hyper-parallelograms, offering an intuitive and consistent geometric interpretation of ExpressivE embeddings and their captured patterns.

- We prove that ExpressivE can capture **any pattern listed in Table 1**. This makes ExpressivE the first model capable of capturing both general composition and hierarchy jointly.

- We prove that our model is **fully expressive**, making ExpressivE the first KGE that both supports composition and is fully expressive.

- We evaluate ExpressivE on the two standard KGC benchmarks WN18RR (Dettmers et al., 2018) and FB15k-237 (Toutanova & Chen, 2015), revealing that ExpressivE is competitive with state-of-the-art (SotA) KGEs and even significantly outperforms them on WN18RR.

**Organization**. Section 2 introduces the KGC problem and methods for evaluating KGEs. Section 3 embeds ExpressivE in the context of related work. Section 4 introduces ExpressivE, the virtual triple space, and interprets our model's parameters within it. Section 5 analyzes our model's expressive power and inference capabilities. Section 6 discusses experimental results together with our model's space complexity and Section 7 summarizes our work. The appendix contains all proofs of theorems.

## 2 KNOWLEDGE GRAPH COMPLETION

This section introduces the KGC problem and evaluation methods (Abboud et al., 2020). Let us first introduce the *triple vocabulary* $\boldsymbol{T}$, consisting of a finite set of *entities* $\boldsymbol{E}$ and *relations* $\boldsymbol{R}$. We call an expression of the form $r_i(e_h, e_t)$ a triple, where $r_i \in \boldsymbol{R}$ and $e_h, e_t \in \boldsymbol{E}$. Furthermore, we call $e_h$ the *head* and $e_t$ the *tail* of the triple. Now, a KG $G$ is a finite set of triples over $\boldsymbol{T}$ and KGC is the problem of predicting missing triples. KGEs can be evaluated by means of an: (1) *experimental* evaluation on benchmark datasets, (2) analysis of the model's *expressiveness*, and (3) analysis of the *inference patterns* that the model can capture. We will discuss each of these points in what follows.

**Experimental Evaluation.** The experimental evaluation of KGEs requires a set of true and corrupted triples. True triples $r_i(e_h, e_t) \in G$ are corrupted by replacing either $e_h$ or $e_t$ with any $e_c \in \boldsymbol{E}$ such

that the corrupted triple does not occur in $G$. KGEs define scores over triples and are optimized to score true triples higher than false ones, thereby estimating a given triple's truth. A KGE's KGC performance is measured with *the mean reciprocal rank* (MRR), the average of inverse ranks ($1/rank$) and Hits@k, the proportion of true triples within the predicted triples whose rank is at maximum k.

**Expressiveness.** A KGE is fully expressive if for any finite set of disjoint true and false triples, a parameter set can be found such that the model classifies the triples of the set correctly. Intuitively, a fully expressive model can represent any given graph. However, this is not necessarily correlated with its inference capabilities (Abboud et al., 2020). For instance, while a fully expressive model may express the entire training set, it may have poor generalization capabilities (Abboud et al., 2020). Conversely, a model that is not fully expressive may underfit the training data severely (Abboud et al., 2020). Hence, KGEs should be both fully expressive and support important inference patterns.

**Inference Patterns.** The generalization capabilities of KGEs are commonly analyzed using inference patterns (short: patterns). They represent logical properties that allow to infer new triples from the ones in $G$. Patterns are of the form $\psi \Rightarrow \phi$, where we call $\psi$ the body and $\phi$ the head of the pattern. For instance, composition $r_1(X, Y) \wedge r_2(Y, Z) \Rightarrow r_3(X, Z)$ is a prominent pattern. Intuitively, it states that if the body of the pattern is satisfied, then the head needs to be satisfied, i.e., if for some entities $e_x, e_y, e_z \in \mathbf{E}$ the triples $r_1(e_x, e_y)$ and $r_2(e_y, e_z)$ are contained in $G$, then also $r_3(e_x, e_z)$ needs to be in $G$. Further patterns are listed in Table 1 and discussed in Section 5. Analyzing the patterns that a KGE supports helps estimate its *inference capabilities* (Abboud et al., 2020).

## 3 RELATED WORK

As our work focuses on KGEs that can *intuitively represent* inference patterns, we have excluded neural models that are less interpretable (Dettmers et al., 2018; Socher et al., 2013; Nathani et al., 2019). We investigate relevant literature to embed ExpressivE in its scientific context below:

**Functional Models.** So far, solely a subset of translational models supports composition. We call this subset *functional models*, as they embed relations as functions $\boldsymbol{f_{r_i}} : \mathbb{K}^d \rightarrow \mathbb{K}^d$ and entities as vectors $\boldsymbol{e_j} \in \mathbb{K}^d$ over some field $\mathbb{K}$. These models represent true triples $r_i(e_h, e_t)$ as $\boldsymbol{e_t} = \boldsymbol{f_{r_i}}(\boldsymbol{e_h})$. Thereby, they can capture composition patterns via functional composition. TransE (Bordes et al., 2013) is the pioneering functional model, embedding relations $r_i$ as $\boldsymbol{f_{r_i}}(\boldsymbol{e_h}) = \boldsymbol{e_h} + \boldsymbol{e_{r_i}}$ with $\boldsymbol{e_{r_i}} \in \mathbb{K}^d$. However, it is neither fully expressive nor can it capture *1-N, N-1, N-N*, nor symmetric relations. RotatE (Sun et al., 2019) embeds relations as rotations in complex space, allowing it to capture symmetry patterns but leaving it otherwise with TransE's limitations. Recently, it was discovered that TransE and RotatE may only capture a fairly limited notion of composition (Zhang et al., 2019; Abboud et al., 2020; Lu & Hu, 2020; Gao et al., 2020), cf. also Appendix K.1. Therefore, extensions have been proposed to tackle some limitations, focusing, e.g., on modeling non-commutative composition (Lu & Hu, 2020; Gao et al., 2020). While these extensions solved some limitations, the purely functional nature of TransE, RotatE, and any of their extensions limits them to capture solely *compositional definition* and not *general composition* (see Table 1 for the defining formulas and cf. also Appendix K.1 for details). Therefore, capturing general composition is still an open problem. Even more, functional models are incapable of capturing vital patterns, such as hierarchies, completely (Abboud et al., 2020).

**Bilinear Models.** Bilinear models factorize the adjacency matrix of a graph with a bilinear product of entity and relation embeddings. The pioneering bilinear model is RESCAL (Nickel et al., 2011). It embeds relations with full-rank $d \times d$ matrices $\boldsymbol{M}$ and entities with $d$-dimensional vectors. DistMult (Yang et al., 2015a) constrains RESCAL's relation matrix $\boldsymbol{M}$ to a diagonal matrix for efficiency reasons, limiting DistMult to capture symmetric relations only. HolE (Nickel et al., 2016) solves this limitation by combining entity embeddings via circular correlation, whereas ComplEx (Trouillon et al., 2016) solves this limitation by embedding relations with a complex-valued diagonal matrix. HolE and ComplEx have subsequently been shown to be equivalent (Hayashi & Shimbo, 2017). SimplE (Kazemi & Poole, 2018) is based on canonical polyadic decomposition (Hitchcock, 1927). TuckER (Balazevic et al., 2019) is based on Tucker decomposition (Tucker, 1966) and extends the capabilities of RESCAL and SimplE (Balazevic et al., 2019). While all bilinear models, excluding DistMult, are fully expressive, they cannot capture any notion of composition.

**Spatial Models.** Spatial models define semantic regions within the embedding space that allow the intuitive representation of certain patterns. In entity classification, for example, bounded axis-aligned

hyper-rectangles (boxes) represent entity classes, capturing class hierarchies naturally through the spatial subsumption of these boxes (Vilnis et al., 2018; Subramanian & Chakrabarti, 2018; Li et al., 2019). Also, query answering systems — such as Query2Box (Ren et al., 2020) — have used boxes to represent answer sets due to their intuitive interpretation as sets of entities. Although Query2Box can be used for KGC, entity classification approaches cannot scalably be employed in the general KGC setting, as this would require an embedding for each entity tuple (Abboud et al., 2020). BoxE (Abboud et al., 2020) is the first spatial KGE dedicated to KGC. It embeds relations as a pair of boxes and entities as a set of points and bumps in the embedding space. The usage of boxes enables BoxE to capture any inference pattern that can be described by the intersection of boxes in the embedding space, such as hierarchy. Moreover, boxes enable BoxE to capture *1-N*, *N-1*, and *N-N* relations naturally. Yet, BoxE cannot capture any notion of composition (Abboud et al., 2020).

**Our Work**. These research gaps, namely that any KGE cannot capture general composition and hierarchy jointly, have motivated our work. In contrast to prior work, our model defines for each relation a hyper-parallelogram, allowing us to combine the benefits of both spatial and functional models. Even more, prior work primarily analyzes the embedding space itself, while we propose the novel *virtual triple space* that allows us to display any captured inference pattern — *including general composition* — through the spatial relation of hyper-parallelograms.

## 4 EXPRESSIVE AND THE VIRTUAL TRIPLE SPACE

This section introduces ExpressivE, a KGE targeted toward KGC with the capabilities of capturing a rich set of inference patterns. ExpressivE embeds entities as *points* and relations as hyper-parallelograms in the virtual triple space $\mathbb{R}^{2d}$. More concretely, instead of analyzing our model in the $d$-dimensional embedding space $\mathbb{R}^d$, we construct the novel *virtual triple space* that grants ExpressivE's parameters a geometric meaning. Above all, the virtual triple space allows us to intuitively interpret ExpressivE embeddings and their captured patterns, as discussed in Section 5.

**Representation.** Entities $e_j \in E$ are embedded in ExpressivE via a vector $\boldsymbol{e_j} \in \mathbb{R}^d$, representing points in the latent embedding space $\mathbb{R}^d$. Relations $r_i \in \boldsymbol{R}$ are embedded as hyper-parallelograms in the virtual triple space $\mathbb{R}^{2d}$. More specifically, ExpressivE assigns to a relation $r_i$ for each of its arity positions $p \in \{h, t\}$ the following vectors: (1) a *slope vector* $\boldsymbol{r_i^p} \in \mathbb{R}^d$, (2) a *center vector* $\boldsymbol{c_i^p} \in \mathbb{R}^d$, and (3) a *width vector* $\boldsymbol{d_i^p} \in (\mathbb{R}_{\geq 0})^d$. Intuitively, these vectors define the slopes $\boldsymbol{r_i^p}$ of the hyper-parallelogram's boundaries, its center $\boldsymbol{c_i^p}$ and width $\boldsymbol{d_i^p}$. A triple $r_i(e_h, e_t)$ is captured to be true in an ExpressivE model if its relation and entity embeddings satisfy the following inequalities:

$$(\boldsymbol{e_h} - \boldsymbol{c_i^h} - \boldsymbol{r_i^t} \odot \boldsymbol{e_t})^{|.|} \preceq \boldsymbol{d_i^h} \tag{1}$$

$$(\boldsymbol{e_t} - \boldsymbol{c_i^t} - \boldsymbol{r_i^h} \odot \boldsymbol{e_h})^{|.|} \preceq \boldsymbol{d_i^t} \tag{2}$$

Where $\boldsymbol{x}^{|.|}$ represents the element-wise absolute value of a vector $\boldsymbol{x}$, $\odot$ represents the Hadamard (i.e., element-wise) product and $\preceq$ represents the element-wise less or equal operator. It is very complex to interpret this model in the embedding space $\mathbb{R}^d$. Hence, we construct followingly a *virtual triple space* in $\mathbb{R}^{2d}$ that will ease reasoning about the parameters and inference capabilities of ExpressivE.

**Virtual Triple Space.** We construct this virtual space by concatenating the head and tail entity embeddings. In detail, this means that any pair of entities $(e_h, e_t) \in \boldsymbol{E} \times \boldsymbol{E}$ defines a point in the virtual triple space by concatenating their entity embeddings $\boldsymbol{e_h}, \boldsymbol{e_t} \in \mathbb{R}^d$, i.e., $(\boldsymbol{e_h} || \boldsymbol{e_t}) \in \mathbb{R}^{2d}$, where $||$ is the concatenation operator. A set of important sub-spaces of the virtual triple space are the 2-dimensional spaces, created from the $j$-th embedding dimension of head entities and the $j$-th dimension of tail entities — i.e., the $j$-th and $(d + j)$-th virtual triple space dimensions. We call them *correlation subspaces*, as they visualize the captured relation-specific dependencies of head and tail entity embeddings as will be discussed followingly. Moreover, we call the correlation subspace spanned by the $j$-th and $(d + j)$-th virtual triple space dimension the $j$-th correlation subspace.

**Parameter Interpretation.** Inequalities 1 and 2 construct each an intersection of two parallel half-spaces in any correlation subspace of the virtual triple space. We call the intersection of two parallel half-spaces a *band*, as they are limited by two parallel boundaries. Henceforth, we will denote with $\boldsymbol{v}(j)$ the $j$-th dimension of a vector $\boldsymbol{v}$. For example, $(\boldsymbol{e_h}(j) - \boldsymbol{c_i^h}(j) - \boldsymbol{r_i^t}(j) \odot \boldsymbol{e_t}(j))^{|.|} \preceq \boldsymbol{d_i^h}(j)$ defines a band in the $j$-th correlation subspace. The intersection of two bands results either in a band (if one band subsumes the other) or a parallelogram. Since we are interested in constructing ExpressivE

embeddings that capture certain inference patterns, it is sufficient to consider parallelograms for these constructions. Figure 1a visualizes a relation parallelogram (green solid) and its parameters (orange dashed) in the $j$-th correlation subspace. In essence, the parallelogram is the result of the intersection of two bands (thick blue and magenta lines), where its boundaries' slopes are defined by $r_i^p$, the center of the parallelogram is defined by $c_i^p$, and finally, the widths of each band are defined by $d_i^p$.

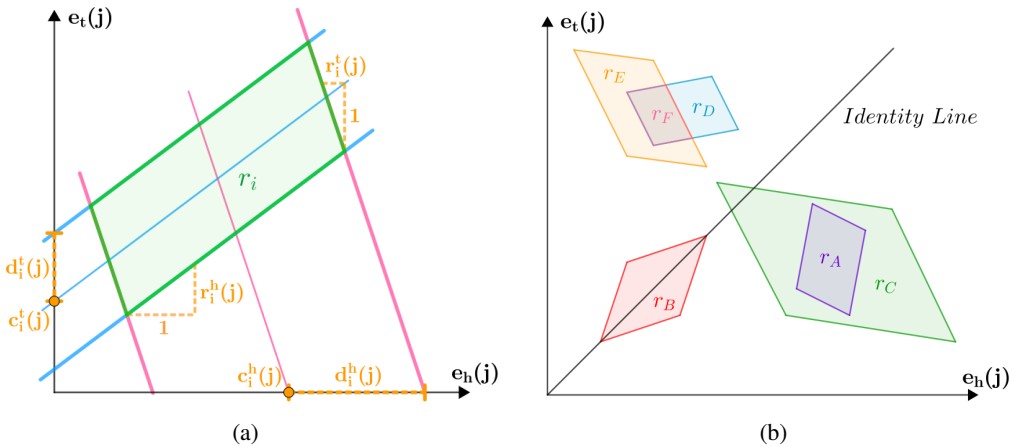

Figure 1: (a) Interpretation of relation parameters (orange dashed) as a parallelogram (green solid) in the $j$-th correlation subspace; (b) Multiple relation embeddings with the following properties: Symmetry ($r_B$), Anti-Symmetry ($r_A, r_D, r_E, r_F$), Inversion ($r_D = r_A^{-1}$), Hierarchy $r_A(X, Y) \Rightarrow r_C(X, Y)$, Intersection $r_D(X, Y) \wedge r_E(X, Y) \Rightarrow r_F(X, Y)$, Mutual Exclusion (e.g., $r_A \cap r_B = \emptyset$).

Since Inequalities 1 and 2 solely capture dependencies within the same dimension, any two different dimensions $j \neq k$ of head and tail entity embeddings are independent. Thus, relations are embedded as hyper-parallelograms in the virtual triple space, whose edges are solely crooked in any $j$-th correlation subspace. Intuitively, the crooked edges represent relation-specific dependencies between head and tail entities and are thus vital for the expressive power of ExpressivE. Note that each correlation subspace represents one dimension of the element-wise Inequalities 1 and 2. Since the sum of all correlation subspaces represents all dimensions of Inequalities 1 and 2, it is sufficient to analyze all correlation subspaces to identify the captured inference patterns of an ExpressivE model.

**Scoring Function.** Let $\boldsymbol{\tau}_{r_i(h,t)}$ denote the embedding of a triple $r_i(h, t)$, i.e., $\boldsymbol{\tau}_{r_i(h,t)} = (e_{ht} - c_i^{ht} - r_i^{th} \odot e_{th})^{|\cdot|}$, with $e_{xy} = (e_x || e_y)$ and $a_i^{xy} = (a_i^x || a_i^y)$ for $a \in \{c, r, d\}$ and $x, y \in \{h, t\}$.

$$D(h, r_i, t) = \begin{cases} \boldsymbol{\tau}_{r_i(h,t)} \oslash w_i, & \text{if } \boldsymbol{\tau}_{r_i(h,t)} \preceq d_i^{ht} \\ \boldsymbol{\tau}_{r_i(h,t)} \odot w_i - k, & \text{otherwise} \end{cases} \tag{3}$$

Equation 3 states the typical distance function of spatial KGEs (Abboud et al., 2020), where $w_i = 2 \odot d_i^{ht} + 1$ is a width-dependent factor and $k = 0.5 \odot (w_i - 1) \odot (w_i - 1 \oslash w_i)$. If a triple $r_i(h, t)$ is captured to be true by an ExpressivE embedding, i.e., if $\boldsymbol{\tau}_{r_i(h,t)} \preceq d_i^{ht}$, then the distance correlates inversely with the hyper-parallelogram's width, keeping low distances/gradients within the parallelogram. Otherwise, the distance correlates linearly with the width to penalize points outside larger parallelograms. Appendix J provides further details on the distance function. The *scoring function* is defined as $s(h, r_i, t) = -||D(h, r_i, t)||_2$. Following Abboud et al. (2020), we optimize the self-adversarial negative sampling loss (Sun et al., 2019) using the Adam optimizer (Kingma & Ba, 2015). We have provided more details on the training setup in Appendix M.

## 5 KNOWLEDGE CAPTURING CAPABILITIES

This section analyzes ExpressivE's expressive power and supported patterns. In what follows, we assume the standard definition of capturing patterns (Sun et al., 2019; Abboud et al., 2020). This means intuitively that a KGE captures a pattern if a set of parameters exists such that the pattern is captured *exactly* and *exclusively*. Appendix C formalizes this notion for our model.

## 5.1 EXPRESSIVENESS

This section analyzes whether ExpressivE is *fully expressive* (Abboud et al., 2020), i.e., can capture any graph $G$ over $\boldsymbol{R}$ and $\boldsymbol{E}$. Theorem 5.1 proves that this is the case by constructing for any graph $G$ an ExpressivE embedding that captures any triple within $G$ to be true and any other triple to be false. Specifically, the proof uses induction, starting with an embedding that captures the complete graph, i.e., any triple over $\boldsymbol{E}$ and $\boldsymbol{R}$ is true. Next, each induction step shows that we can alter the embedding to make an arbitrarily picked triple of the form $r_i(e_j, e_k)$ with $r_i \in \boldsymbol{R}$, $e_j, e_k \in \boldsymbol{E}$ and $e_j \neq e_k$ false. Finally, we add $|\boldsymbol{E}| * |\boldsymbol{R}|$ dimensions to make any self-loop — i.e., any triple of the form $r_i(e_j, e_j)$ with $r_i \in \boldsymbol{R}$ and $e_j \in \boldsymbol{E}$ — false. The full, quite technical proof can be found in Appendix D.

**Theorem 5.1 (Expressive Power)** *ExpressivE can capture any arbitrary graph $G$ over $\boldsymbol{R}$ and $\boldsymbol{E}$ if the embedding dimensionality $d$ is at least in $O(|\boldsymbol{E}| * |\boldsymbol{R}|)$.*

## 5.2 INFERENCE PATTERNS

This section proves that ExpressivE can capture any pattern from Table 1. First, we discuss how ExpressivE represents inference patterns with at most two variables. Next, we introduce the notion of compositional definition and continue by identifying how this pattern is described in the virtual triple space. Then, we define general composition, building on both the notion of compositional definition and hierarchy. Finally, we conclude this section by discussing the key properties of ExpressivE.

**Two-Variable Patterns.** Figure 1b displays several one-dimensional relation embeddings and their captured patterns in a correlation subspace. Intuitively, ExpressivE represents: (1) symmetry patterns $r_1(X, Y) \Rightarrow r_1(Y, X)$ via symmetric hyper-parallelograms, (2) anti-symmetry patterns $r_1(X, Y) \Rightarrow \neg r_1(Y, X)$ via hyper-parallelograms that do not overlap with their mirror image, (3) inversion patterns $r_1(X, Y) \Leftrightarrow r_2(Y, X)$ via $r_2$'s hyper-parallelogram being the mirror image of $r_1$'s, (4) hierarchy patterns $r_1(X, Y) \Rightarrow r_2(X, Y)$ via $r_2$'s hyper-parallelogram subsuming $r_1$'s, (5) intersection patterns $r_1(X, Y) \wedge r_2(X, Y) \Rightarrow r_3(X, Y)$ via $r_3$'s hyper-parallelogram subsuming the intersection of $r_1$'s and $r_2$'s, and (6) mutual exclusion patterns $r_1(X, Y) \wedge r_2(X, Y) \Rightarrow \bot$ via mutually exclusive hyper-parallelograms of $r_1$ and $r_2$. We have formally proven that ExpressivE can capture any of these two-variable inference patterns in Theorem 5.2 (see Appendices F and G).

**Theorem 5.2** *ExpressivE captures (a) symmetry, (b) anti-symmetry, (c) inversion, (d) hierarchy, (e) intersection, and (f) mutual exclusion.*

**Compositional Definition.** A compositional definition pattern is of the form $r_1(X, Y) \wedge r_2(Y, Z) \Leftrightarrow r_d(X, Z)$, where we call $r_1$ and $r_2$ the *composing* and $r_d$ the *compositionally defined relation*. In essence, this pattern defines a relation $r_d$ that describes the start and end entities of a path $X \xrightarrow{r_1} Y \xrightarrow{r_2} Z$. Since any two relations $r_1$ and $r_2$ can instantiate the body of a compositional definition pattern, any such pair may produce a new compositionally defined relation $r_d$. Interestingly, compositional definition translates analogously into the virtual triple space: Intuitively, this means that the embeddings of any two relations $r_1$ and $r_2$ define for $r_d$ a *convex* region — which we call the *compositionally defined region* — that captures $r_1(X, Y) \wedge r_2(Y, Z) \Leftrightarrow r_d(X, Z)$, leading to Theorem 5.3 (proven in Appendix E). Based on this insight, ExpressivE captures compositional definition patterns by embedding the compositionally defined relation $r_d$ with the compositionally defined region, defined by the relation embeddings of $r_1$ and $r_2$. We have formally proven that ExpressivE can capture compositional definition in Theorem 5.4 (see Appendices F and G).

**Theorem 5.3** *Let $r_1, r_2, r_d \in \boldsymbol{R}$ be relations, $\boldsymbol{s_1}, \boldsymbol{s_2}$ be their ExpressivE embeddings, and assume $r_1(X, Y) \wedge r_2(Y, Z) \Leftrightarrow r_d(X, Z)$ holds. Then there exists a region $\boldsymbol{s_d}$ in the virtual triple space $\mathbb{R}^{2d}$ such that (i) $\boldsymbol{s_1}, \boldsymbol{s_2},$ and $\boldsymbol{s_d}$ capture $r_1(X, Y) \wedge r_2(Y, Z) \Leftrightarrow r_d(X, Z)$ and (ii) $\boldsymbol{s_d}$ is* convex.

**General Composition.** In contrast to compositional definition, general composition $r_1(X, Y) \wedge r_2(Y, Z) \Rightarrow r_3(X, Z)$ does not specify the composed relation $r_3$ completely. Specifically, general composition allows the relation $r_3$ to include additional entity pairs not described by the start and end entities of the path $X \xrightarrow{r_1} Y \xrightarrow{r_2} Z$. Therefore, to capture general composition, we need to combine hierarchy and compositional definition. Formally this means that we express general composition as: $\{r_1(X, Y) \wedge r_2(Y, Z) \Leftrightarrow r_d(X, Z), r_d(X, Y) \Rightarrow r_3(X, Y)\}$. We have proven that ExpressivE can capture general composition in Theorem 5.4 (see Appendices F and G for the full proofs).

**Theorem 5.4** *ExpressivE captures compositional definition and general composition.*

We argue that hierarchy and general composition are very tightly connected as hierarchies are hidden within general composition. If, for instance, $r_1$ were to represent the relation that solely captures self-loops, then the general composition $r_1(X, Y) \wedge r_2(Y, Z) \Rightarrow r_3(X, Z)$ would reduce to a hierarchy $r_2(X, Y) \Rightarrow r_3(X, Y)$. This hints at why our model is the first to support general composition, as ExpressivE can capture both hierarchy and composition jointly in a single embedding space.

**Key Properties**. ExpressivE's way of capturing inference patterns has several interesting implications:

1. We observe that ExpressivE embeddings offer an intuitive geometric interpretation: there is a natural correspondence between (a) relations in the KG – and – regions (representing mathematical relations) in the virtual triple space, (b) relation containment, intersection, and disjointness in the KG – and – region containment, intersection, and disjointness in the virtual triple space, (c) symmetry, anti-symmetry, and inversion in the KG – and – symmetry, anti-symmetry, and reflection in the virtual triple space, (d) compositional definition in the KG – and – the composition of mathematical relations in the virtual triple space.

2. Next, we observe that ExpressivE captures a general composition pattern if the hyper-parallelogram of the pattern's head relation subsumes the compositionally defined region defined by its body relations. Thereby, ExpressivE assigns a novel spatial interpretation to general composition patterns, generalizing the spatial interpretation that is directly provided by set-theoretic patterns such as hierarchy, intersection, and mutual exclusion.

3. Finally, capturing general composition patterns through the subsumption of spatial regions allows ExpressivE to provably capture composition patterns for *1-N*, *N-1*, and *N-N* relations. We provide further empirical evidence to this in Appendix I.1.

## 6 EXPERIMENTAL EVALUATION AND SPACE COMPLEXITY

In this section, we evaluate ExpressivE on the standard KGC benchmarks WN18RR (Dettmers et al., 2018) and FB15k-237 (Toutanova & Chen, 2015) and report SotA results, providing strong empirical evidence for the theoretical strengths of ExpressivE. Furthermore, we perform an ablation study on ExpressivE's parameters to quantify the importance of each parameter and finally perform a relation-wise performance comparison on WN18RR to provide an in-depth analysis of our results.

### 6.1 KNOWLEDGE GRAPH COMPLETION

**Experimental Setup.** As in Abboud et al. (2020), we compare ExpressivE to the functional models TransE (Bordes et al., 2013) and RotatE (Sun et al., 2019), spatial model BoxE (Abboud et al., 2020), and bilinear models DistMult (Yang et al., 2015a), ComplEx (Trouillon et al., 2016), and TuckER (Balazevic et al., 2019). ExpressivE is trained with gradient descent for up to 1000 epochs, stopping the training if after 100 epochs the Hits@10 score did not increase by at least 0.5% for WN18RR and 1% for FB15k-237. We use the model of the final epoch for testing. Each experiment was repeated 3 times to account for small performance fluctuations. In particular, the MRR values fluctuate by less than 0.003 between runs for any dataset. We maintain the fairness of our result comparison by considering KGEs with a dimensionality $d \leq 1000$ (Balazevic et al., 2019; Abboud et al., 2020). To allow a direct comparison of ExpressivE's performance and parameter efficiency to its closest functional relative RotatE and spatial relative BoxE, we employ the same embedding dimensionality for the benchmarks as RotatE and BoxE. Appendix M lists further setup details, hyperparameters, libraries (Ali et al., 2021), hardware details, definitions of metrics, and properties of datasets.

Table 2: Model sizes of ExpressivE, BoxE, and RotatE models of equal dimensionality.

| Benchmark | Dimensionality | ExpressivE | BoxE | RotatE |
|-----------|----------------|------------|------|--------|
| WN18RR | 500 | **467MB** | 930MB | 930MB |
| FB15k-237 | 1000 | **366MB** | 687MB | 687MB |

**Space Complexity.** For a $d$-dimensional embedding, RotatE and BoxE have $(2|\boldsymbol{E}| + 2|\boldsymbol{R}|)d$, whereas ExpressivE has $(|\boldsymbol{E}| + 6|\boldsymbol{R}|)d$ parameters, where $|\boldsymbol{E}|$ is the number of entities and $|\boldsymbol{R}|$ the number

of relations. Since $|\boldsymbol{R}| << |\boldsymbol{E}|$ in most graphs, (e.g., FB15k-237: $|\boldsymbol{R}|/|\boldsymbol{E}| = 0.016$) ExpressivE almost *halves* the number of parameters for a $d$-dimensional embedding compared to BoxE and RotatE. Table 2 lists the model sizes of trained ExpressivE, BoxE, and RotatE models of the same dimensionality, empirically confirming that ExpressivE almost *halves* BoxE's and RotatE's sizes.

Table 3: KGC performance of ExpressivE and SotA KGEs on FB15k-237 and WN18RR. The table shows the best-published results of the competing models per family, specifically: TransE and RotatE (Sun et al., 2019), BoxE (Abboud et al., 2020), DistMult and ComplEx (Ruffinelli et al., 2020; Yang et al., 2015b), and TuckER (Balazevic et al., 2019).

| Family | Model | WN18RR | | | | FB15k-237 | | | |
|---|---|---|---|---|---|---|---|---|---|
| | | H@1 | H@3 | H@10 | MRR | H@1 | H@3 | H@10 | MRR |
| Func. & Spatial | Base ExpressivE | **.464** | **.522** | .597 | **.508** | .243 | .366 | .512 | .333 |
| | Func. ExpressivE | .407 | .519 | **.619** | .482 | **.256** | **.387** | .535 | **.350** |
| | BoxE | .400 | .472 | .541 | .451 | .238 | .374 | **.538** | .337 |
| | RotatE | .428 | .492 | .571 | .476 | .241 | .375 | .533 | .338 |
| | TransE | .013 | .401 | .529 | .223 | .233 | .372 | .531 | .332 |
| Bilinear | DistMult | - | - | .531 | .452 | - | - | .531 | .343 |
| | ComplEx | - | - | **.547** | **.475** | - | - | .536 | .348 |
| | TuckER | **.443** | **.482** | .526 | .470 | **.266** | **.394** | **.544** | **.358** |

**Benchmark Results.** We use two versions of ExpressivE in the benchmarks, one where the width parameter $d_i^{ht}$ is learned and one where $d_i^{ht} = 0$, called Base ExpressivE and Functional ExpressivE. Tables 2 and 3 reveal that Functional ExpressivE, with only *half* the number of parameters of BoxE and RotatE, performs best among spatial and functional models on FB15k-237 and is competitive with TuckER, especially in MRR. Even more, Base ExpressivE *outperforms all* competing models significantly on WN18RR. The significant performance increase of Base ExpressivE on WN18RR is likely due to WN18RR containing both hierarchy and composition patterns in contrast to FB15k-237 (similar to the discussion of Abboud et al. (2020)). We will empirically investigate the reasons for ExpressivE's performances on FB15k-237 and WN18RR in Section 6.2 and Section 6.3.

**Discussion.** Tables 2 and 3 reveal that ExpressivE is highly parameter efficient compared to related spatial and functional models while reaching competitive performance on FB15k-237 and even new SotA performance on WN18RR, supporting the extensive theoretical results of our paper.

## 6.2 ABLATION STUDY

This section analyses how constraints on ExpressivE's parameters impact its benchmark performances. Specifically, we analyze the following constrained ExpressivE versions: (1) *Base ExpressivE*, which represents ExpressivE without any parameter constraints, (2) *Functional ExpressivE*, where the width parameter $d_i^{ht}$ of each relation $r_i$ is zero, (3) *EqSlopes ExpressivE*, where all slope vectors are constrained to be equal — i.e., $r_i^{ht} = r_k^{ht}$ for any relations $r_i$ and $r_k$, (4) *NoCenter ExpressivE*, where the center vector $c_i^{ht}$ of any relation $r_i$ is zero, and (5) *OneBand ExpressivE*, where each relation is embedded by solely one band instead of two — i.e., OneBand ExpressivE captures a triple $r_i(e_h, e_t)$ to be true if its relation and entity embeddings only satisfy Inequality 1.

Table 4: Ablation study on ExpressivE's parameters.

| Model | WN18RR | | | | FB15k-237 | | | |
|---|---|---|---|---|---|---|---|---|
| | H@1 | H@3 | H@10 | MRR | H@1 | H@3 | H@10 | MRR |
| Base ExpressivE | **.464** | **.522** | .597 | **.508** | .243 | .366 | .512 | .333 |
| Func. ExpressivE | .407 | .519 | **.619** | .482 | **.256** | **.387** | **.535** | **.350** |
| EqSlopes ExpressivE | .254 | .415 | .528 | .353 | .237 | .361 | .510 | .328 |
| NoCenter ExpressivE | .457 | .514 | .591 | .501 | .224 | .349 | .494 | .314 |
| OneBand ExpressivE | .435 | .480 | .538 | .470 | .230 | .352 | .491 | .318 |

**Ablation Results.** Table 4 provides the results of the ablation study on WN18RR and FB15k-237. It reveals that each component of ExpressivE is vital as setting all slopes $r_i^{ht}$ to be equal (EqSlopes ExpressivE) or removing the center $c_i^{ht}$ (NoCenter ExpressivE), width $d_i^{ht}$ (Functional ExpressivE), or a band (OneBand ExpressivE) results in performance losses on at least one benchmark. Interestingly, Functional outperforms Base ExpressivE on FB15k-237. Since Functional ExpressivE sets $d_i^{ht} = 0$, the relation embeddings reduce from a hyper-parallelogram to a function. Intuitively, this means that Functional ExpressivE loses the spatial capabilities of Base ExpressivE such as the ability to capture hierarchy, while it maintains functional capabilities, such as the ability to capture compositional definition. Table 4 reveals that the performance of ExpressivE increases when we remove its spatial capabilities, depicted by the performance gain of Functional over Base ExpressivE. This result hints at FB15k-237 not containing many hierarchy patterns. Thus, FB15k-237 cannot exploit the added capabilities of Base ExpressivE, namely the ability to capture general composition and hierarchy. In contrast, the significant performance gain of Base ExpressivE over Functional ExpressivE on WN18RR is likely due to WN18RR containing many composition and hierarchy patterns ((Abboud et al., 2020), cf. Appendix I.2), exploiting Base ExpressivE's added capabilities.

## 6.3 WN18RR PERFORMANCE ANALYSIS

This section analyses the performance of ExpressivE and its closest spatial relative BoxE (Abboud et al., 2020) and functional relative RotatE (Sun et al., 2019) on WN18RR. Table 5 lists the MRR of ExpressivE, RotatE, and BoxE for each of the 11 relations of WN18RR. Bold values represent the best and underlined values represent the second-best results across the compared models.

Table 5: Relation-wise MRR comparison of ExpressivE, RotatE, and BoxE on WN18RR.

| Relation Name | ExpressivE | RotatE | BoxE |
|---|---|---|---|
| member_meronym | **0.233** | 0.199 | 0.226 |
| hypernym | **0.189** | 0.162 | 0.159 |
| has_part | **0.198** | 0.187 | 0.168 |
| instance_hypernym | 0.352 | 0.326 | **0.425** |
| synset_domain_topic_of | 0.363 | **0.384** | 0.323 |
| member_of_domain_usage | 0.288 | 0.333 | **0.360** |
| member_of_domain_region | 0.123 | 0.188 | **0.189** |
| also_see | **0.649** | 0.631 | 0.517 |
| derivationally_related_from | **0.956** | 0.943 | 0.902 |
| similar_to | **1.000** | **1.000** | **1.000** |
| verb_group | **0.972** | 0.843 | 0.876 |

**Results.** ExpressivE performs very well on many relations, where either only BoxE or only RotatE produces good rankings, empirically confirming that ExpressivE combines the inference capabilities of BoxE (hierarchy) and RotatE (compositional definition). Additionally, ExpressivE does not only reach similar performances as RotatE and BoxE if only one of them produces good rankings but even surpasses both of them significantly on relations such as *verb_group*, *also_see*, and *hypernym*. This gives strong experimental evidence that ExpressivE combines the inference capabilities of functional and spatial models, even extending them by novel capabilities (such as general composition), empirically supporting our extensive theoretical results of Section 5.

## 7 CONCLUSION

In this paper, we have introduced ExpressivE, a KGE that (i) represents inference patterns through spatial relations of hyper-parallelograms, offering an intuitive and consistent geometric interpretation of ExpressivE embeddings and their captured patterns, (ii) can capture a wide variety of important inference patterns, including hierarchy and general composition jointly, resulting in strong benchmark performances (iii) is fully expressive, and (iv) reaches competitive performance on FB15k-237, even outperforming any competing model significantly on WN18RR. In the future, we plan to analyze the performance of ExpressivE on further datasets, particularly focusing on the relation between constrained ExpressivE versions and dataset properties.

## REPRODUCIBILITY STATEMENT

We have made our code publicly available in a GitHub repository[1]. It contains, in addition to the code of ExpressivE, a setup file to install the necessary libraries and a ReadMe.md file containing library versions and running instructions to facilitate the reproducibility of our results. Furthermore, we have provided all information for reproducing our results — including the concrete hyperparameters, further details of our experiment setup, the used libraries (Ali et al., 2021), hardware details, definitions of metrics, properties of datasets, and more — in Appendix M. We have provided the complete proofs for our extensive theoretical results in the appendix and stated the complete set of assumptions we made. Specifically, each theorem states any necessary assumption, and each proof starts by listing any property we assume without loss of generality. We have proven Theorem 5.1 in Appendix D, Theorem 5.3 in Appendix E, and Theorems 5.2 and 5.4 in Appendices F and G.

## ACKNOWLEDGMENTS

We are grateful to Maximilian Beck for helpful discussions and feedback. This work has been funded by the Vienna Science and Technology Fund (WWTF) [10.47379/VRG18013].

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

## A    OVERVIEW OF THE APPENDIX

This appendix contains detailed proofs, analyses, and descriptions of our experimental setup. Section B gives an overview of the used notations. Section C specifies the complete formal definitions for all used terms. Section D contains a detailed proof of Theorem 5.1, i.e., showing that ExpressivE is fully expressive. Section E proves Theorem 5.3, developing technical machinery to support further proofs in this appendix. Sections F and G provide additional propositions and proofs for Theorems 5.2 and 5.4, proving ExpressivE's inference capabilities. Section H proves that ExpressivE can capture more than one step of composition. Section I provides additional empirical evidence for ExpressivE's theoretical capabilities, specifically investigating ExpressivE's performance stratified by cardinalities, captured composition patterns, and reasoning steps. Section J explores the main goals and properties of ExpressivE's distance function introduced in Section 4. Section K further discusses ExpressivE's functional and spatial nature, comparing ExpressivE's inference capabilities with those of spatial and functional models. Section L further analyses the trade-off discovered in Section 6 between high expressive power and low degrees of freedom. Finally, Section M provides further details on the experimental setup, benchmark datasets, and evaluation metrics.

## B    NOTATION

In this section, we give a brief overview of the most important notations we use:

$v$ ... non-bold symbols represent scalars

$\boldsymbol{v}$ ... bold symbols represent vectors, sets or tuples

$\boldsymbol{0}$ ... represents a vector of solely zeros (the same semantics apply to $\boldsymbol{0.5}$, $\boldsymbol{1}$, and $\boldsymbol{2}$)

$\oslash$ ... represents the elementwise division operator

$\odot$ ... represents the elementwise (Hadamard) product operator

$\succeq$ ... represents the elementwise greater or equal operator

$\succ$ ... represents the elementwise greater operator

$\preceq$ ... represents the elementwise less or equal operator

$\prec$ ... represents the elementwise less operator

$\boldsymbol{x}^{|\cdot|}$ ... represents the elementwise absolute value

$||$ ... represents the concatenation operator

$\boldsymbol{v}(j)$ ... represents the $j$-th dimension of a vector $\boldsymbol{v}$

## C    FORMAL DEFINITIONS

In this section, we formally introduce the notions of capturing a pattern in an ExpressivE model that we informally discussed in Section 5. Furthermore, we will introduce some additional notations, which will help us simplify the upcoming proofs and present them intuitively.

**Knowledge Graph.** A tuple $(\boldsymbol{G}, \boldsymbol{E}, \boldsymbol{R})$ is called a knowledge graph, where $\boldsymbol{R}$ is a finite set of relations, $\boldsymbol{E}$ is a finite set of entities, and $\boldsymbol{G} \subseteq \boldsymbol{E} \times \boldsymbol{R} \times \boldsymbol{E}$ is a finite set of triples. W.l.o.g., we assume that any relation is non-empty since assigning an empty hyper-parallelogram to an empty relation would be trivial, just adding unnecessary complexity to the proofs.

**ExpressivE model.** A tuple $M = (\boldsymbol{\epsilon}, \boldsymbol{\sigma}, \boldsymbol{\delta}, \boldsymbol{\rho})$ is called an ExpressivE model, where $\boldsymbol{\epsilon} \subset 2^{\mathbb{R}^d}$ is a finite set of entity embeddings, $\boldsymbol{\sigma} \subset 2^{\mathbb{R}^d}$ is a finite set of center embeddings, $\boldsymbol{\delta} \subset 2^{\mathbb{R}^d}$ is a finite set of width embeddings, and $\boldsymbol{\rho} \subset 2^{\mathbb{R}^d}$ is a finite set of slope vectors.

**Linking Embeddings to KGs.** An ExpressivE model and a KG are linked via the following assignment functions: The entity assignment function $\boldsymbol{f_e} : \boldsymbol{E} \to \boldsymbol{\epsilon}$ assigns an entity embedding $\boldsymbol{e_h} \in \boldsymbol{\epsilon}$ to each entity $e_h \in \boldsymbol{E}$. Based on $\boldsymbol{f_e}$, the virtual assignment function $\boldsymbol{f_v} : \boldsymbol{E} \times \boldsymbol{E} \to \mathbb{R}^{2d}$ defines

for any pair of entities $(e_h, e_t) \in E$ a virtual entity pair embedding $\boldsymbol{f_v}(e_h, e_t) = (\boldsymbol{f_e}(e_h) || \boldsymbol{f_e}(e_t))$, where $||$ represents the concatenation operator. Furthermore, the relation assignment function $\boldsymbol{f_h}(r_i) : \boldsymbol{R} \to \mathbb{R}^{2d} \times \mathbb{R}^{2d} \times \mathbb{R}^{2d}$ assigns a hyper-parallelogram to each relation $r_i$. In more detail, this means that $\boldsymbol{f_h}(r_i) = (\boldsymbol{c_i^{ht}}, \boldsymbol{d_i^{ht}}, \boldsymbol{r_i^{th}})$, where $\boldsymbol{c_i^{ht}} = (\boldsymbol{c_i^h} || \boldsymbol{c_i^t})$ are two concatenated center embeddings with $\boldsymbol{c_i^h}, \boldsymbol{c_i^t} \in \boldsymbol{\sigma}$, where $\boldsymbol{d_i^{ht}} = (\boldsymbol{d_i^h} || \boldsymbol{d_i^t})$ are two concatenated width embeddings with $\boldsymbol{d_i^h}, \boldsymbol{d_i^t} \in \boldsymbol{\delta}$, and where $\boldsymbol{r_i^{th}} = (\boldsymbol{r_i^t} || \boldsymbol{r_i^h})$ are two concatenated slope vectors with $\boldsymbol{r_i^t}, \boldsymbol{r_i^h} \in \boldsymbol{\rho}$. Intuitively, $\boldsymbol{f_h}(r_i)$ defines a hyper-parallelogram in the virtual triple space $\mathbb{R}^{2d}$ as described in Section 4.

**Model Configuration.** We call an ExpressivE model $\boldsymbol{M}$ together with a concrete relation assignment function $\boldsymbol{f_h}$ a relation configuration $\boldsymbol{m_h} = (\boldsymbol{M}, \boldsymbol{f_h})$ and if it additionally has a concrete virtual assignment function $\boldsymbol{f_v}$, we call it a complete model configuration $\boldsymbol{m} = (\boldsymbol{M}, \boldsymbol{f_h}, \boldsymbol{f_v})$.

**Definition of Truth.** A triple $r_i(e_h, e_t)$ holds in some $\boldsymbol{m}$, with $r_i \in \boldsymbol{R}$ and $e_h, e_t \in \boldsymbol{E}$ iff Inequalities 1 and 2 hold for the assigned embeddings of $h, t$, and $r$. This means more specifically that Inequalities 1 and 2 need to hold for $\boldsymbol{f_v}(e_h, e_t) = (\boldsymbol{f_e}(e_h) || \boldsymbol{f_e}(e_t)) = (\boldsymbol{e_h} || \boldsymbol{e_t})$ and $\boldsymbol{f_h}(r_i) = (\boldsymbol{c_i^{ht}}, \boldsymbol{d_i^{ht}}, \boldsymbol{r_i^{th}})$, with $\boldsymbol{c_i^{ht}} = (\boldsymbol{c_i^h} || \boldsymbol{c_i^t})$, $\boldsymbol{d_i^{ht}} = (\boldsymbol{d_i^h} || \boldsymbol{d_i^t})$, and $\boldsymbol{r_i^{th}} = (\boldsymbol{r_i^t} || \boldsymbol{r_i^h})$. At an intuitive level, this means that a triple $r_i(e_h, e_t)$ is true in some complete model configuration $\boldsymbol{m}$ iff the virtual pair embedding $\boldsymbol{f_v}(e_h, e_t)$ of entities $e_h$ and $e_t$ lies within the hyper-parallelogram of relation $r_i$ defined by $\boldsymbol{f_h}(r_i)$.

**Simplifying Notations.** Therefore, to simplify the upcoming proofs, we denote with $\boldsymbol{f_v}(e_h, e_t) \in \boldsymbol{f_h}(r_i)$ that the virtual pair embedding $\boldsymbol{f_v}(e_h, e_t) \in \mathbb{R}^{2d}$ of an entity pair $(e_h, e_t) \in \boldsymbol{E} \times \boldsymbol{E}$ lies within the hyper-parallelogram $\boldsymbol{f_h}(r_i) \subseteq \mathbb{R}^{2d} \times \mathbb{R}^{2d} \times \mathbb{R}^{2d}$ of some relation $r_i \in \boldsymbol{R}$ in the virtual triple space. Accordingly, for sets of virtual pair embeddings $\boldsymbol{P} := \{\boldsymbol{f_v}(e_{h_1}, e_{t_1}), \ldots, \boldsymbol{f_v}(e_{h_n}, e_{t_n})\}$, we denote with $\boldsymbol{P} \subseteq \boldsymbol{f_h}(r_i)$ that all virtual pair embeddings of $\boldsymbol{P}$ lie within the hyper-parallelogram of the relation $r_i$. Furthermore, we denote with $\boldsymbol{f_v}(e_h, e_t) \notin \boldsymbol{f_h}(r_i)$ that a virtual pair embedding $\boldsymbol{f_v}(e_h, e_t)$ does not lie within the hyper-parallelogram of a relation $r_i$ and with $\boldsymbol{P} \nsubseteq \boldsymbol{f_h}(r_i)$ we denote that an entire set of virtual pair embeddings $\boldsymbol{P}$ does not lie within the hyper-parallelogram of a relation $r_i$.

**Capturing Inference Patterns.** Based on the previous definitions, we define capturing patterns formally: A relation configuration $\boldsymbol{m_h}$ captures a pattern $\psi$ *exactly* if for any ground pattern $\phi_{B_1} \wedge \cdots \wedge \phi_{B_m} \Rightarrow \phi_H$ within the deductive closure of $\psi$ and for any instantiation of $\boldsymbol{f_e}$ and $\boldsymbol{f_v}$ the following conditions are satisfied:

- if $\phi_H$ is a triple and if $\boldsymbol{m_h}$ captures the body triples to be true — i.e., $\boldsymbol{f_v}(args(\phi_{B_1})) \in \boldsymbol{f_h}(rel(\phi_{B_1})), \ldots, \boldsymbol{f_v}(args(\phi_{B_m})) \in \boldsymbol{f_h}(rel(\phi_{B_m}))$ — then $\boldsymbol{m_h}$ also captures the head triple to be true — i.e., $\boldsymbol{f_v}(args(\phi_H)) \in \boldsymbol{f_h}(rel(\phi_H))$.
- if $\phi_H = \bot$, then $\boldsymbol{m_h}$ captures at least one of the body triples to be false — i.e., there is some $j \in \{1, \ldots, m\}$ such that $\boldsymbol{f_v}(args(\phi_{B_j})) \notin \boldsymbol{f_h}(rel(\phi_{B_j}))$.

where $args()$ is the function that returns the arguments of a triple and $rel()$ is the function that returns the relation of the triple. Furthermore, a relation configuration $\boldsymbol{m_h}$ captures a pattern $\psi$ *exactly* and *exclusively* if (1) $\boldsymbol{m_h}$ exactly captures $\psi$ and (2) $\boldsymbol{m_h}$ does not capture any *positive* pattern $\phi$ (i.e., $\phi \in \{symmetry, \ inversion, \ hierarchy, \ intersection, \ composition\}$) such that $\psi \not\models \phi$ except where the body of $\phi$ is not satisfied over $\boldsymbol{m_h}$.

**Discussion.** In the following, some intuition of the above definition of capturing a pattern is provided. Capturing a pattern *exactly* is defined straightforwardly by adhering to the semantics of logical implication $\phi := \phi_B \Rightarrow \phi_H$, i.e., a relation configuration $\boldsymbol{m_h}$ needs to be found such that for any complete model configuration $\boldsymbol{m}$ over $\boldsymbol{m_h}$ if the body $\phi_B$ of the pattern is satisfied, then its head $\phi_H$ can be inferred.

Capturing a pattern *exactly* and *exclusively* imposes additional constraints. Here, we do not solely aim at capturing a pattern but at additionally showcasing that a pattern can be captured independently from any other pattern. Therefore, some notion of minimality/exclusiveness of a pattern is needed. As in Abboud et al. (2020), we define minimality by means of *solely* capturing those positive patterns $\phi$ that directly follow from the deductive closure of the pattern $\psi$, except for those $\phi$ that are captured trivially, i.e., except for those $\phi$ where their body is not satisfied over the constructed $\boldsymbol{m_h}$.

As presented in Section 5, we can express any supported pattern by means of spatial relations of the corresponding relation hyper-parallelograms in the virtual triple space. Therefore, we formulate

*exclusiveness* intuitively as the ability to limit the intersection of hyper-parallelograms to only those intersections that directly follow from the captured pattern $\psi$ for any known relation $r_i \in \boldsymbol{R}$, which is in accordance with BoxE's notion of exclusiveness (Abboud et al., 2020).

Note that our definition of capturing patterns solely depends on relation configurations. This is vital for ExpressivE to be able to capture patterns in a *lifted* manner, i.e., ExpressivE shall be able to capture patterns without the need of grounding them first. Furthermore, being able to capture patterns in a lifted way is not only efficient but also natural as we aim at capturing patterns between relations. Thus it would be unnatural if constraints on entity embeddings were necessary to capture such relation-specific patterns.

As outlined in the previous paragraphs, our definition is in accordance with the literature, focuses on efficiently capturing patterns, and gives us a formal foundation for the upcoming proofs, which will show that ExpressivE can capture various logical patterns.

## D    PROOF OF FULLY EXPRESSIVENESS

In this section, we prove Theorem 5.1. We will show by induction that ExpressivE is fully expressive. We will first only consider self-loop-free triples, i.e., triples of the form $r_i(e_j, e_k)$ with $e_j, e_k \in \boldsymbol{E}$, $r_i \in \boldsymbol{R}$ and $j \neq k$ and later remove unwanted self-loops from the constructed model configuration.

Since our proof is highly technical, we will first give some general intuition and then formally state our proof. In the base case, we consider an ExpressivE model that captures the complete graph $G$ over the entity vocabulary $\boldsymbol{E}$ and the relationship vocabulary $\boldsymbol{R}$, i.e., the graph that contains all triples from the universe. In the induction step, we prove that we can adjust our ExpressivE model to make any arbitrary self-loop-free triple of $G$ false while maintaining the truth value of any other triple in the universe.

In the induction step, we make triples $r_i(e_j, e_k)$ false by translating the entity embeddings of $e_j$ and $e_k$ such that a hyper-parallelogram can separate pairs of entity embeddings that shall be true from those that shall be false. Afterward, we translate and shear $r_i$'s hyper-parallelogram to match such a separating shape.

Finally, after the induction step, we add a separate dimension for any possible self-loop, i.e., triple of the form $r_i(e_j, e_j)$ such that we can make any self-loop false. Thereby, we show that ExpressivE can make any triple false and thus that ExpressivE can capture any graph $G$ over $\boldsymbol{R}$ and $\boldsymbol{E}$.

Our proof shares some common ideas with the fully expressiveness proof of BoxE (Abboud et al., 2020), yet differs dramatically in many aspects. BoxE embeds relations with two axis-aligned boxes and entities with two separate embedding vectors, which greatly simplifies the fully expressiveness proof of BoxE, as the two entity embeddings are independent of each other. This grants BoxE some flexibility for adapting model configuration yet imposes substantial restrictions, such as that BoxE cannot capture any notion of composition patterns. Our model does not have these restrictions and uses only one embedding vector per entity instead, pushing the complexity of our model to the relation embeddings by representing relations as hyper-parallelogram in the virtual triple space. This, however, has the consequence that we cannot easily change entity embeddings without moving and sheering relation embeddings as well when we want to make solely one triple false and preserve the truth value of any other triple. In the following proof, we will explain the complex adjustment of relation embeddings and many more novel aspects of our proof in more detail.

We start our proof by making the following assumptions without loss of generality:

1. Any relation $r_i \in \boldsymbol{R}$ and entity $e_j \in \boldsymbol{E}$ is indexed with $0 \leq i \leq |\boldsymbol{R}| - 1$ and $0 \leq j \leq |\boldsymbol{E}| - 1$.

2. The dimensionality of each relation and entity embedding vectors is equal to $|\boldsymbol{E}| * |\boldsymbol{R}|$. Furthermore, $\boldsymbol{v}(i, j)$ represents the dimension $i * |\boldsymbol{E}| + j$ of the vector $\boldsymbol{v}$. Intuitively, the dimensions of $\boldsymbol{v}(i, 0), \dots, \boldsymbol{v}(i, |\boldsymbol{E}| - 1)$ corresponds to the dimensions reserved for relation $r_i$.

3. The slope vectors of relation $r_i \in \boldsymbol{R}$ are positive, i.e., $\boldsymbol{r_i^h}, \boldsymbol{r_i^t} > 0$.

4. Any entity embedding is positive, i.e., for any entity $e_k \in \boldsymbol{E}$ holds that $\boldsymbol{e_k} > 0$.

5. For any pair of entities $e_{k_1}, e_{k_2} \in \boldsymbol{E}$ holds that $\boldsymbol{e_{k_1}}(i, k_1) \geq \boldsymbol{e_{k_2}}(i, k_1) + m$, with $m > 0$.

Building on these assumptions, we prove fully expressiveness by induction as follows:

**Base Case.** We initialize a graph $G$ as the whole universe over $\boldsymbol{E}$ and $\boldsymbol{R}$ and construct a complete model configuration $\boldsymbol{m} = (\boldsymbol{M}, \boldsymbol{f_h}, \boldsymbol{f_v})$ with dimensionality $|\boldsymbol{E}| * |\boldsymbol{R}|$ such that $G$ is captured and all assumptions are satisfied. Concretely, we specify for any dimension $(i, k_1)$ with $0 \le i \le |\boldsymbol{R}| - 1$ and $0 \le k_1 \le |\boldsymbol{E}| - 1$ the embedding values of entity embeddings with index $k_1$ to set $\boldsymbol{e_{k_1}}(i, k_1) = 2$ and with index $k_2 \ne k_1$ to $\boldsymbol{e_{k_2}}(i, k_1) = 1$. Furthermore, we specify for any dimension $(i, k)$ with $0 \le i \le |\boldsymbol{R}| - 1$ and $0 \le k \le |\boldsymbol{E}| - 1$ the embedding of relation $r_i$ to $\boldsymbol{c_i^h}(i, k) = \boldsymbol{c_i^t}(i, k) = 0$, $\boldsymbol{r_i^h}(i, k) = 1$, $\boldsymbol{r_i^t}(i, k) = 2$ and $\boldsymbol{d_i^h}(i, k) = \boldsymbol{d_i^t}(i, k) = 4$. As can be shown easily the constructed complete model configuration satisfies all assumptions and makes any triple over $\boldsymbol{R}$ and $\boldsymbol{E}$ true. Note that in particular, any self-loop is also captured to be true in the constructed complete model configuration.

**Induction step.** In the induction step, we adjust the entity and relation embeddings of the complete model configuration such that a single triple $r_i(e_j, e_k)$ is made false without affecting the truth value of any other triple within the graph $G$. We denote any adjusted embedding with an asterisk $\boldsymbol{v^*}$ and the old value of the embedding with $\boldsymbol{v}$ and perform the following adjustments:

1. Increase any slope vector $\boldsymbol{r_i^{t*}}(i, k) := \boldsymbol{r_i^t}(i, k) + \Delta r_i^t$ with $\Delta r_i^t > 0$ such that:
$$\boldsymbol{e_j}(i, k) - \boldsymbol{r_i^t}(i, k)\boldsymbol{e_k}(i, k) - \boldsymbol{c_i^h}(i, k) - \Delta r_i^t m \le -\boldsymbol{d_i^h}(i, k)$$

2. Since $\boldsymbol{e_k}(i, k)$ is by assumption the largest value in dimension $(i, k)$, we can specify the following two values:
$$\Delta r_i^{max} := \Delta r_i^t \boldsymbol{e_k}(i, k)$$
$$\Delta r_i^{ub} := \Delta r_i^t (\boldsymbol{e_k}(i, k) - m)$$
   with $\Delta r_i^{ub} < \Delta r_i^{max}$.

3. Using this definition, we increase all entity embeddings $\boldsymbol{e_{j'}}$ with $j' \ne j$ in dimension $(i, k)$ by:
$$\boldsymbol{e_{j'}^*}(i, k) := \boldsymbol{e_{j'}}(i, k) + \Delta r_i^{max}$$

4. Furthermore, we increase all entity embeddings $\boldsymbol{e_{j'}}$ with $j' \ne j$ in dimension $(i, k)$ by:
$$\boldsymbol{e_{j'}^*}(i, k) := \boldsymbol{e_{j'}}(i, k) + \Delta r_i^{max}$$

5. For any relation with index $i \ne i'$, we adjust any head band in dimension $(i, k)$ by moving its center downwards and growing the band upwards. This means formally that we update the following embeddings:
$$s := \boldsymbol{r_{i'}^t}(i, k)\Delta r_i^t m + \Delta r_i^{max}$$
$$\boldsymbol{d_{i'}^{h*}}(i, k) := \boldsymbol{d_{i'}^h}(i, k) + \frac{s}{2}$$
$$\boldsymbol{c_{i'}^{h*}}(i, k) := \boldsymbol{c_{i'}^h}(i, k) - \boldsymbol{r_{i'}^t}(i, k)\Delta r_i^{max} + \frac{s}{2}$$

6. We adjust any tail band in dimension $(i, k)$ by moving its center downwards and growing the band upwards. This means formally that we update the following embeddings:
$$s := \boldsymbol{r_{i'}^h}(i, k)\Delta r_i^t m + \Delta r_i^{max}$$
$$\boldsymbol{d_{i'}^{t*}}(i, k) := \boldsymbol{d_{i'}^t}(i, k) + \frac{s}{2}$$
$$\boldsymbol{c_{i'}^{t*}}(i, k) := \boldsymbol{c_{i'}^t}(i, k) - \boldsymbol{r_{i'}^h}(i, k)\Delta r_i^{max} + \frac{s}{2}$$

7. For any relation with index $i$, we adjust any head band in dimension $(i, k)$ by moving its center downwards and growing the band upwards. This means formally that we update the following embeddings:

$$s := (\Delta r_i^t + \boldsymbol{r_i^t}(i, k))\Delta r_i^t m + \Delta r_i^{max}$$

$$\boldsymbol{d_i^{h*}}(i, k) := \boldsymbol{d_i^h}(i, k) + \frac{s}{2}$$

$$\boldsymbol{c_i^{h*}}(i, k) := \boldsymbol{c_i^h}(i, k) - \Delta r_i^t \Delta r_i^{max} - \boldsymbol{r_i^t}(i, k)\Delta r_i^{max} + \frac{s}{2}$$

In the induction step, we adjust the slope vectors (Step 1), the entity embeddings (Step 2-4), and the width and center embeddings (Step 5-7). Intuitively, by changing the slope vector of relation hyper-parallelograms, we sheer the hyper-parallelograms. Furthermore, we translate any desired entity embeddings more than the undesired entity embedding of $e_j$. This allows us to draw a separating hyper-parallelogram between the point defined by $(e_j, e_k)$ and any other pair of entities that shall remain within relation $r_i$. Finally, we must move the sheered hyper-parallelograms into the correct position and stretch it to make all desired triples true.

Our next goal is to show this behavior formally. We will first show that the initially true triple $r_i(e_j, e_k)$ is false, then continue by showing that the truth value of any other triple is preserved.

Since the induction steps perform only adjustments in dimension $(i, k)$, we only have to consider the dimension $(i, k)$ for any embedding vector in the following inequalities. Please note that to state the inequalities concisely, we have omitted the notation $(i, k)$ from any embedding vector $\boldsymbol{v}$ in the following inequalities. For instance, we will denote $\boldsymbol{r_i^t}(i, k)$ with $\boldsymbol{r_i^t}$ henceforth.

Let $s := (\Delta r_i^t + \boldsymbol{r_i^t})\Delta r_i^t m + \Delta r_i^{max}$, then we can show that our induction step makes $r_i(e_j, e_k)$ false as follows:

$$\boldsymbol{e_j} - \boldsymbol{r_i^t}\boldsymbol{e_k} - \boldsymbol{c_i^h} - \Delta r_i^t m \leq -\boldsymbol{d_i^h} \tag{4}$$

$$\boldsymbol{e_j} - \boldsymbol{r_i^t}\boldsymbol{e_k} - \boldsymbol{c_i^h} + \Delta r_i^{ub} - \Delta r_i^{max} - \Delta r_i^t \Delta r_i^{max} + \Delta r_i^t \Delta r_i^{max}$$
$$-\boldsymbol{r_i^t}\Delta r_i^{max} + \boldsymbol{r_i^t}\Delta r_i^{max} + \frac{s}{2} - \frac{s}{2} \leq -\boldsymbol{d_i^h} \tag{5}$$

$$\boldsymbol{e_j} + \Delta r_i^{ub} - (\boldsymbol{r_i^t} + \Delta r_i^t)(\boldsymbol{e_k} + \Delta r_i^{max}) - (\boldsymbol{c_i^h} - \Delta r_i^t \Delta r_i^{max}$$
$$-\boldsymbol{r_i^t}\Delta r_i^{max} + \frac{s}{2}) \leq -(\boldsymbol{d_i^h} + \frac{s}{2}) \tag{6}$$

$$\boldsymbol{e_j^*} - \boldsymbol{r_i^{t*}}\boldsymbol{e_k^*} - \boldsymbol{c_i^{h*}} \leq -\boldsymbol{d_i^{h*}} \tag{7}$$

Inequality 4 follows directly from Induction Step 1. Next, in Inequality 5 we add many terms that eliminate each other and apply $\Delta r_i^{ub} - \Delta r_i^{max} = \Delta r_i^t(\boldsymbol{e_k} - m) - \Delta r_i^t \boldsymbol{e_k} = -m\Delta r_i^t$. Finally, in Inequality 6 we restructure the terms such that we can substitute the terms for the adjusted embedding vectors defined in Steps 1-7. Through this substitution, we obtain Inequality 7, which reveals that the adjusted embeddings $\boldsymbol{e_j^*}, \boldsymbol{e_k^*}$ do not lie within the adjusted hyper-parallelogram of relation $r_i$. Therefore, we have shown that the adjustments of the complete model configuration listed in Steps 1-7 have made the triple $r_i(e_j, e_k)$ false, as required.

Next, we need to show that the truth value of any other self-loop-free triple $r_{i'}(e_{j'}, e_{k'})$ with $j' \neq k'$ is not altered after the induction step. We start by showing that any triple $r_{i'}(e_{j'}, e_{k'})$ that is true in $\boldsymbol{m}$ remains true after the induction step. Since what follows is a highly technical proof, we give some intuition now. We make a case distinction of any possible true triple in $G$ and perform the following steps. First, we assume that the triple is true and therefore instantiate Inequalities 1 and 2 with the embeddings prior to the induction step. Note that it is solely necessary to consider Inequality 1 as the proofs work vice versa for Inequality 2. Thus, we solely consider Inequality 1 henceforth. Next, we add terms that eliminate each other and adjustment terms $a$ such that we can substitute our inequality with the adjusted embedding values $\boldsymbol{v*}$. Finally, we show that Inequality 1 is satisfied for the adjusted embedding values. Note that Inequality 1 defines two inequalities, specifically $\boldsymbol{e_h} - \boldsymbol{c_i^h} - \boldsymbol{r_i^t} \odot \boldsymbol{e_t} \preceq \boldsymbol{d_i^h}$ and $\boldsymbol{e_h} - \boldsymbol{c_i^h} - \boldsymbol{r_i^t} \odot \boldsymbol{e_t} \succeq -\boldsymbol{d_i^h}$. Therefore, we denote with $(<)$ the proof for the first inequality and with $(>)$ the proof for the second inequality. Thereby, we will show that if

we assume the triple $r_{i'}(e_{j'}, e_{k'})$ to be true in the complete model configuration prior to the induction step, we can follow that $r_{i'}(e_{j'}, e_{k'})$ stays true after the adjustments of the induction step. To provide the complete formal side of our proof, we consider the following 12 cases:

1. **Case** $i' = i, j' = j, k' = j, k' \neq k$:

   (<)  Let $s := (\Delta r_i^t + \boldsymbol{r_i^t})\Delta r_i^t m + \Delta r_i^{max}$ and let $a := (\Delta r_i^{max} - \Delta r_i^{ub})(1 - \Delta r_i^t - \boldsymbol{r_i^t}\Delta r^{ub})$. Note that $a$ is positive since $a = \Delta r_i^t m + \Delta r_i^{max}$ holds. Therefore, we can perform the following transformations:

$$\boldsymbol{e_j} - \boldsymbol{r_i^t e_j} - \boldsymbol{c_i^h} \leq \boldsymbol{d_i^h} \tag{8}$$

$$\boldsymbol{e_j} - \boldsymbol{r_i^t e_j} - \boldsymbol{c_i^h} - a + s - s \leq \boldsymbol{d_i^h} \tag{9}$$

$$\boldsymbol{e_j} + \Delta r_i^{ub} - (\boldsymbol{r_i^t} + \Delta r_i^t)(\boldsymbol{e_j} + \Delta r_i^{ub}) - (\boldsymbol{c_i^h} - \Delta r_i^t \Delta r_i^{max}$$
$$-\boldsymbol{r_i^t}\Delta r_i^{max} + \frac{s}{2}) \leq \boldsymbol{d_i^h} + \frac{s}{2} \tag{10}$$

$$\boldsymbol{e_j^*} - \boldsymbol{r_i^{t*} e_j^*} - \boldsymbol{c_i^{h*}} \leq \boldsymbol{d_i^{h*}} \tag{11}$$

   (>)  Let $a := (\Delta r_i^{max} - \Delta r_i^{ub})(\Delta r_i^t + \boldsymbol{r_i^t}) + \Delta r_i^{ub} - \Delta r_i^{max}$ and let $s := (\Delta r_i^t + \boldsymbol{r_i^t})\Delta r_i^t m + \Delta r_i^{max}$. Note that $a$ is positive since (1) $a = m\Delta r_i^t(\Delta r_i^t + \boldsymbol{r_i^t} - 1)$, (2) we initialize $\boldsymbol{r_i^t}$ in the base case to 2 in any dimension and (3) any induction step may only increase $\boldsymbol{r_i^t}$. Therefore, we can perform the following transformations:

$$\boldsymbol{e_j} - \boldsymbol{r_i^t e_j} - \boldsymbol{c_i^h} \geq -\boldsymbol{d_i^h} \tag{12}$$

$$\boldsymbol{e_j} - \boldsymbol{r_i^t e_j} - \boldsymbol{c_i^h} + a + \frac{s}{2} - \frac{s}{2} \geq -\boldsymbol{d_i^h} \tag{13}$$

$$\boldsymbol{e_j} + \Delta r_i^{ub} - (\boldsymbol{r_i^t} + \Delta r_i^t)(\boldsymbol{e_j} + \Delta r_i^{ub}) - (\boldsymbol{c_i^h} - \Delta r_i^t \Delta r_i^{max}$$
$$-\boldsymbol{r_i^t}\Delta r_i^{max} + \frac{s}{2}) \geq -(\boldsymbol{d_i^h} + \frac{s}{2}) \tag{14}$$

$$\boldsymbol{e_j^*} - \boldsymbol{r_i^{t*} e_j^*} - \boldsymbol{c_i^{h*}} \geq -\boldsymbol{d_i^{h*}} \tag{15}$$

2. **Case** $i' = i, j' = j, k' \neq j, k' = k$:

   As can be seen easily this case describes the triple $r_i(e_j, e_k)$, which shall be made false in the induction step. We have shown that the induction step changes the triples truth value to false in Inequalities 4-7 and therefore omitted the case here.

3. **Case** $i' = i, j' = j, k' \neq j, k' \neq k$:

   (<)  Let $s := (\Delta r_i^t + \boldsymbol{r_i^t})\Delta r_i^t m + \Delta r_i^{max}$ and let $a := \Delta r_i^t \boldsymbol{e_{k'}} + s - \Delta r_i^{ub}$. Note that $a$ is positive since $a = \Delta r_i^t(\boldsymbol{e_{k'}} + m(1 + \Delta r_i^t + \boldsymbol{r_i^t}))$ holds. Therefore, we can perform the following transformations:

$$\boldsymbol{e_j} - \boldsymbol{r_i^t e_{k'}} - \boldsymbol{c_i^h} \leq \boldsymbol{d_i^h} \tag{16}$$

$$\boldsymbol{e_j} - \boldsymbol{r_i^t e_{k'}} - \boldsymbol{c_i^h} - a + \Delta r_i^t \Delta r_i^{max} - \Delta r_i^t \Delta r_i^{max} + \boldsymbol{r_i^t}\Delta r_i^{max} - \boldsymbol{r_i^t}\Delta r_i^{max} \leq \boldsymbol{d_i^h} \tag{17}$$

$$\boldsymbol{e_j} + \Delta r_i^{ub} - (\boldsymbol{r_i^t} + \Delta r_i^t)(\boldsymbol{e_{k'}} + \Delta r_i^{max}) - (\boldsymbol{c_i^h} - \Delta r_i^t \Delta r_i^{max}$$
$$-\boldsymbol{r_i^t}\Delta r_i^{max} + \frac{s}{2}) \leq \boldsymbol{d_i^h} + \frac{s}{2} \tag{18}$$

$$\boldsymbol{e_j^*} - \boldsymbol{r_i^{t*} e_{k'}^*} - \boldsymbol{c_i^{h*}} \leq \boldsymbol{d_i^{h*}} \tag{19}$$

($>$) Let $a := \Delta r_i^{ub} - \Delta r_i^t e_{k'}$ and let $s := (\Delta r_i^t + r_i^t)\Delta r_i^t m + \Delta r_i^{max}$. Note that $a$ is positive since $\Delta r_i^{ub} \geq \Delta r_i^t e_{k'}$ holds. Therefore, we can perform the following transformations:

$$e_j - r_i^t e_{k'} - c_i^h \geq -d_i^h \tag{20}$$

$$e_j - r_i^t e_{k'} - c_i^h + a + \Delta r_i^t \Delta r_i^{max} - \Delta r_i^t \Delta r_i^{max} + r_i^t \Delta r_i^{max}$$
$$-r_i^t \Delta r_i^{max} + \frac{s}{2} - \frac{s}{2} \geq -d_i^h \tag{21}$$

$$e_j + \Delta r_i^{ub} - (r_i^t + \Delta r_i^t)(e_{k'} + \Delta r_i^{max}) - (c_i^h - \Delta r_i^t \Delta r_i^{max}$$
$$-r_i^t \Delta r_i^{max} + \frac{s}{2}) \geq -(d_i^h + \frac{s}{2}) \tag{22}$$

$$e_j^* - r_i^{t*} e_{k'}^* - c_i^{h*} \geq -d_i^{h*} \tag{23}$$

4. **Case** $i' = i, j' \neq j, k' = j, k' \neq k$:

($<$) Let $a := \Delta r_i^t e_j$ and let $s := (\Delta r_i^t + r_i^t)\Delta r_i^t m + \Delta r_i^{max}$. Note that $a$ is trivially positive since we initially assumed $e_j > 0$ and since we assumed in Step 1 $\Delta r_i^t > 0$. Therefore, we can perform the following transformations:

$$e_{j'} - r_i^t e_j - c_i^h \leq d_i^h \tag{24}$$

$$e_{j'} - r_i^t e_j - c_i^h - a + \Delta r_i^t \Delta r_i^{max} - \Delta r_i^t \Delta r_i^{max} + r_i^t \Delta r_i^{max}$$
$$-r_i^t \Delta r_i^{max} + s - s \leq d_i^h \tag{25}$$

$$e_{j'} + \Delta r_i^{max} - (r_i^t + \Delta r_i^t)(e_j + \Delta r_i^{ub}) - (c_i^h - \Delta r_i^t \Delta r_i^{max}$$
$$-r_i^t \Delta r_i^{max} + \frac{s}{2}) \leq d_i^h + \frac{s}{2} \tag{26}$$

$$e_{j'}^* - r_i^{t*} e_j^* - c_i^{h*} \leq d_i^{h*} \tag{27}$$

($>$) Let $a := \Delta r_i^{max} - \Delta r_i^t e_j + \Delta r_i^t m(\Delta r_i^t + r_i^t)$ and let $s := (\Delta r_i^t + r_i^t)\Delta r_i^t m + \Delta r_i^{max}$. Note that $a$ is positive since $\Delta r_i^{max} - \Delta r_i^t e_j > 0$. Therefore, we can perform the following transformations:

$$e_{j'} - r_i^t e_j - c_i^h \geq -d_i^h \tag{28}$$

$$e_{j'} - r_i^t e_j - c_i^h + a + \Delta r_i^t \Delta r_i^{max} - \Delta r_i^t \Delta r_i^{max} + r_i^t \Delta r_i^{max}$$
$$-r_i^t \Delta r_i^{max} + \frac{s}{2} - \frac{s}{2} \geq -d_i^h \tag{29}$$

$$e_{j'} + \Delta r_i^{max} - (r_i^t + \Delta r_i^t)(e_j + \Delta r_i^{ub}) - (c_i^h - \Delta r_i^t \Delta r_i^{max}$$
$$-r_i^t \Delta r_i^{max} + \frac{s}{2}) \geq -(d_i^h + \frac{s}{2}) \tag{30}$$

$$e_{j'}^* - r_i^{t*} e_j^* - c_i^{h*} \geq -d_i^{h*} \tag{31}$$

5. **Case** $i' = i, j' \neq j, k' \neq j, k' = k$:

($<$) Let $s := (\Delta r_i^t + r_i^t)\Delta r_i^t m + \Delta r_i^{max}$ and let $a := s + \Delta r_i^t e_k - \Delta r_i^{max}$. Note that $a$ is positive since $a = \Delta r_i^t(e_k + m(\Delta r_i^t + r_i^t))$ holds. Therefore, we can perform the following transformations:

$$e_{j'} - r_i^t e_k - c_i^h \leq d_i^h \tag{32}$$

$$e_{j'} - r_i^t e_k - c_i^h - a + \Delta r_i^t \Delta r_i^{max} - \Delta r_i^t \Delta r_i^{max} + r_i^t \Delta r_i^{max}$$
$$-r_i^t \Delta r_i^{max} \leq d_i^h \tag{33}$$

$$e_{j'} + \Delta r_i^{max} - (r_i^t + \Delta r_i^t)(e_k + \Delta r_i^{max}) - (c_i^h - \Delta r_i^t \Delta r_i^{max}$$
$$-r_i^t \Delta r_i^{max} + \frac{s}{2}) \leq d_i^h + \frac{s}{2} \tag{34}$$

$$e_{j'}^* - r_i^{t*} e_k^* - c_i^{h*} \leq d_i^{h*} \tag{35}$$

(>) Let $s := (\Delta r_i^t + \boldsymbol{r_i^t})\Delta r_i^t m + \Delta r_i^{max}$. Using this definition, we can perform the following transformations:

$$\boldsymbol{e_{j'}} - \boldsymbol{r_i^t}\boldsymbol{e_k} - \boldsymbol{c_i^h} \geq -\boldsymbol{d_i^h} \tag{36}$$

$$\boldsymbol{e_{j'}} - \boldsymbol{r_i^t}\boldsymbol{e_k} - \boldsymbol{c_i^h} + \Delta r_i^{max} - \Delta r_i^{max} + \Delta r_i^t\Delta r_i^{max} - \Delta r_i^t\Delta r_i^{max}$$
$$+ \boldsymbol{r_i^t}\Delta r_i^{max} - \boldsymbol{r_i^t}\Delta r_i^{max} - \frac{s}{2} \geq -\boldsymbol{d_i^h} - \frac{s}{2} \tag{37}$$

$$\boldsymbol{e_{j'}} + \Delta r_i^{max} - (\boldsymbol{r_i^t} + \Delta r_i^t)(\boldsymbol{e_k} + \Delta r_i^{max}) - (\boldsymbol{c_i^h} - \Delta r_i^t\Delta r_i^{max}$$
$$- \boldsymbol{r_i^t}\Delta r_i^{max} + \frac{s}{2}) \geq -\boldsymbol{d_i^h} - \frac{s}{2} \tag{38}$$

$$\boldsymbol{e_{j'}^*} - \boldsymbol{r_i^{t*}}\boldsymbol{e_k^*} - \boldsymbol{c_i^{h*}} \geq -\boldsymbol{d_i^{h*}} \tag{39}$$

6. **Case** $i' = i, j' \neq j, k' \neq j, k' \neq k$:

(<) Let $s := (\Delta r_i^t + \boldsymbol{r_i^t})\Delta r_i^t m + \Delta r_i^{max}$ and let $a := s - \Delta r_i^{max} + \Delta r_i^t\boldsymbol{e_{k'}}$. Note that $a$ is positive since $a = \Delta r_i^t(\boldsymbol{e_{k'}} + m(\Delta r_i^t + \boldsymbol{r_i^t}))$ holds. Therefore, we can perform the following transformations:

$$\boldsymbol{e_{j'}} - \boldsymbol{r_i^t}\boldsymbol{e_{k'}} - \boldsymbol{c_i^h} \leq \boldsymbol{d_i^h} \tag{40}$$

$$\boldsymbol{e_{j'}} - \boldsymbol{r_i^t}\boldsymbol{e_{k'}} - \boldsymbol{c_i^h} - a + \Delta r_i^t\Delta r_i^{max} - \Delta r_i^t\Delta r_i^{max} + \boldsymbol{r_i^t}\Delta r_i^{max}$$
$$- \boldsymbol{r_i^t}\Delta r_i^{max} \leq \boldsymbol{d_i^h} \tag{41}$$

$$\boldsymbol{e_{j'}} + \Delta r_i^{max} - (\boldsymbol{r_i^t} + \Delta r_i^t)(\boldsymbol{e_{k'}} + \Delta r_i^{max}) - (\boldsymbol{c_i^h} - \Delta r_i^t\Delta r_i^{max}$$
$$- \boldsymbol{r_i^t}\Delta r_i^{max} + \frac{s}{2}) \leq \boldsymbol{d_i^h} + \frac{s}{2} \tag{42}$$

$$\boldsymbol{e_{j'}^*} - \boldsymbol{r_i^{t*}}\boldsymbol{e_{k'}^*} - \boldsymbol{c_i^{h*}} \leq \boldsymbol{d_i^{h*}} \tag{43}$$

(>) Let $a := \Delta r_i^{max} - \Delta r_i^t\boldsymbol{e_{k'}}$ and let $s := (\Delta r_i^t + \boldsymbol{r_i^t})\Delta r_i^t m + \Delta r_i^{max}$. Therefore, we can perform the following transformations:

$$\boldsymbol{e_{j'}} - \boldsymbol{r_i^t}\boldsymbol{e_{k'}} - \boldsymbol{c_i^h} \geq -\boldsymbol{d_i^h} \tag{44}$$

$$\boldsymbol{e_{j'}} - \boldsymbol{r_i^t}\boldsymbol{e_{k'}} - \boldsymbol{c_i^h} + a + \Delta r_i^t\Delta r_i^{max} - \Delta r_i^t\Delta r_i^{max} + \boldsymbol{r_i^t}\Delta r_i^{max}$$
$$- \boldsymbol{r_i^t}\Delta r_i^{max} + \frac{s}{2} - \frac{s}{2} \geq -\boldsymbol{d_i^h} \tag{45}$$

$$\boldsymbol{e_{j'}} + \Delta r_i^{max} - (\boldsymbol{r_i^t} + \Delta r_i^t)(\boldsymbol{e_{k'}} + \Delta r_i^{max}) - (\boldsymbol{c_i^h} - \Delta r_i^t\Delta r_i^{max}$$
$$- \boldsymbol{r_i^t}\Delta r_i^{max} + \frac{s}{2}) \geq -(\boldsymbol{d_i^h} + \frac{s}{2}) \tag{46}$$

$$\boldsymbol{e_{j'}^*} - \boldsymbol{r_i^{t*}}\boldsymbol{e_{k'}^*} - \boldsymbol{c_i^{h*}} \geq -\boldsymbol{d_i^{h*}} \tag{47}$$

7. **Case** $i' \neq i, j' = j, k' \neq j, k' = k$:

(<) Let $s := \boldsymbol{r_{i'}^t}\Delta r_i^t m + \Delta r_i^{max}$ and let $a := s - \Delta r_i^{ub}$. Note that $a$ is positive since $a = \Delta r_i^t m(1 + \boldsymbol{r_{i'}^t})$ holds. Therefore, we can perform the following transformations:

$$\boldsymbol{e_j} - \boldsymbol{r_{i'}^t}\boldsymbol{e_k} - \boldsymbol{c_{i'}^h} \leq \boldsymbol{d_{i'}^h} \tag{48}$$

$$\boldsymbol{e_j} - \boldsymbol{r_{i'}^t}\boldsymbol{e_k} - \boldsymbol{c_{i'}^h} - a + \boldsymbol{r_{i'}^t}\Delta r_i^{max} - \boldsymbol{r_{i'}^t}\Delta r_i^{max} \leq \boldsymbol{d_{i'}^h} \tag{49}$$

$$\boldsymbol{e_j} + \Delta r_i^{ub} - \boldsymbol{r_{i'}^t}(\boldsymbol{e_k} + \Delta r_i^{max}) - (\boldsymbol{c_{i'}^h} - \boldsymbol{r_{i'}^t}\Delta r_i^{max} + \frac{s}{2}) \leq \boldsymbol{d_{i'}^h} + \frac{s}{2} \tag{50}$$

$$\boldsymbol{e_j^*} - \boldsymbol{r_{i'}^{t*}}\boldsymbol{e_k^*} - \boldsymbol{c_{i'}^{h*}} \leq \boldsymbol{d_{i'}^{h*}} \tag{51}$$

(>) Let $a := \Delta r_i^{ub}$ and let $s := \boldsymbol{r_{i'}^t}\Delta r_i^t m + \Delta r_i^{max}$. Note that $a$ is trivially positive since $\Delta r_i^{ub}$ is positive. Therefore, we can perform the following transformations:

$$e_j - r_{i'}^t e_k - c_{i'}^h \geq -d_{i'}^h \tag{52}$$

$$e_j - r_{i'}^t e_k - c_{i'}^h + a + r_{i'}^t \Delta r_i^{max} - r_{i'}^t \Delta r_i^{max} + \frac{s}{2} - \frac{s}{2} \geq -d_{i'}^h \tag{53}$$

$$e_j + \Delta r_i^{ub} - r_{i'}^t (e_k + \Delta r_i^{max}) - (c_{i'}^h - r_{i'}^t \Delta r_i^{max} + \frac{s}{2}) \geq -(d_{i'}^h + \frac{s}{2}) \tag{54}$$

$$e_j^* - r_{i'}^{t*} e_k^* - c_{i'}^{h*} \geq -d_{i'}^{h*} \tag{55}$$

8. **Case** $i' \neq i, j' = j, k' \neq j, k' \neq k$:

   As can be seen easily this case generates the same inequalities as the previous case, except that $k' = k$. Therefore, no relevant difference has to be considered, which is why we omit this case.

9. **Case** $(i' \neq i, j' \neq j, k' = j, k' \neq k)$:

   ($<$)  Let $s := r_i^t \Delta r_i^t m + \Delta r_i^{max}$. Using this definition we can make the following transformations:

$$e_{j'} - r_{i'}^t e_j - c_{i'}^h \leq d_{i'}^h \tag{56}$$

$$e_{j'} - r_{i'}^t e_j - c_{i'}^h + s - s \leq d_{i'}^h \tag{57}$$

$$e_{j'} + \Delta r_i^{max} - r_{i'}^t (e_j + \Delta r_i^{ub}) - (c_{i'}^h - r_{i'}^t \Delta r_i^{max} + \frac{s}{2}) \leq d_{i'}^h + \frac{s}{2} \tag{58}$$

$$e_{j'}^* - r_{i'}^{t*} e_j^* - c_{i'}^{h*} \leq d_{i'}^{h*} \tag{59}$$

   ($>$)  Let $a := \Delta r_i^{max} + r_i^t (\Delta r_i^{max} - \Delta r_i^{ub})$ and let $s := r_i^t \Delta r_i^t m + \Delta r_i^{max}$. Note that $a$ is positive since $\Delta r_i^{max} > \Delta r_i^{ub}$. Therefore, we can perform the following transformations:

$$e_{j'} - r_{i'}^t e_j - c_{i'}^h \geq -d_{i'}^h \tag{60}$$

$$e_{j'} - r_{i'}^t e_j - c_{i'}^h + a + \frac{s}{2} - \frac{s}{2} \geq -d_{i'}^h \tag{61}$$

$$e_{j'} + \Delta r_i^{max} - r_{i'}^t (e_j + \Delta r_i^{ub}) - (c_{i'}^h - r_{i'}^t \Delta r_i^{max} + \frac{s}{2}) \geq -(d_{i'}^h + \frac{s}{2}) \tag{62}$$

$$e_{j'}^* - r_{i'}^{t*} e_j^* - c_{i'}^{h*} \geq -d_{i'}^{h*} \tag{63}$$

10. **Case** $i' \neq i, j' \neq j, k' \neq j, k' = k$:

   ($<$)  Let $s := r_{i'}^t \Delta r_i^t m + \Delta r_i^{max}$ and let $a := s - \Delta r_i^{max}$. Note that $a$ is positive since $a = r_i^t \Delta r_i^t m$ holds. Therefore, we can perform the following transformations:

$$e_{j'} - r_{i'}^t e_k - c_{i'}^h \leq d_{i'}^h \tag{64}$$

$$e_{j'} - r_{i'}^t e_k - c_{i'}^h - a - \Delta r_i^{max} + \Delta r_i^{max} - r_{i'}^t \Delta r_i^{max} + r_{i'}^t \Delta r_i^{max} \leq d_{i'}^h \tag{65}$$

$$e_{j'} + \Delta r_i^{max} - r_{i'}^t (e_k + \Delta r_i^{max}) - (c_{i'}^h - r_{i'}^t \Delta r_i^{max} + \frac{s}{2}) \leq d_{i'}^h + \frac{s}{2} \tag{66}$$

$$e_{j'}^* - r_{i'}^{t*} e_k^* - c_{i'}^{h*} \leq d_{i'}^{h*} \tag{67}$$

   ($>$)  Let $s := r_{i'}^t \Delta r_i^t m + \Delta r_i^{max}$ and $a := \Delta r_i^{max}$. Note that $a$ is trivially positive since $\Delta r_i^{max}$ is positive. Therefore, we can perform the following transformations:

$$e_{j'} - r_{i'}^t e_k - c_{i'}^h \geq -d_{i'}^h \tag{68}$$

$$e_{j'} - r_{i'}^t e_k - c_{i'}^h + a + r_{i'}^t \Delta r_i^{max} - r_{i'}^t \Delta r_i^{max} + \frac{s}{2} - \frac{s}{2} \geq -d_{i'}^h \tag{69}$$

$$e_{j'} + \Delta r_i^{max} - r_{i'}^t (e_k + \Delta r_i^{max}) - (c_{i'}^h - r_{i'}^t \Delta r_i^{max} + \frac{s}{2}) \geq -(d_{i'}^h + \frac{s}{2}) \tag{70}$$

$$e_{j'}^* - r_{i'}^{t*} e_k^* - c_{i'}^{h*} \geq -d_{i'}^{h*} \tag{71}$$

11. **Case** $i' \neq i, j' \neq j, k' \neq j, k' \neq k$:

    As can be seen easily this case generates the same inequalities as the previous case, except that $k' = k$. Therefore, no relevant difference has to be considered, which is why we omit this case.

12. **Case** $i' \neq i, j' = j, k' = j, k' \neq k$:

    ($<$)   Let $s := \boldsymbol{r^t_{i'}} \Delta r^t_i m + \Delta r^{max}_i$ and let $a := \Delta r^{max}_i - \Delta r^{ub}_i$. Note that $a$ is positive since $a = \Delta r^t_i m$. Therefore, we can perform the following transformations:

$$\boldsymbol{e_j} - \boldsymbol{r^t_{i'}} \boldsymbol{e_j} - \boldsymbol{c^h_{i'}} \leq \boldsymbol{d^h_{i'}} \tag{72}$$

$$\boldsymbol{e_j} - \boldsymbol{r^t_{i'}} \boldsymbol{e_j} - \boldsymbol{c^h_{i'}} - a - s + s \leq \boldsymbol{d^h_{i'}} \tag{73}$$

$$\boldsymbol{e_j} + \Delta r^{ub}_i - \boldsymbol{r^t_{i'}} (\boldsymbol{e_j} + \Delta r^{ub}_i) - (\boldsymbol{c^h_{i'}} - \boldsymbol{r^t_{i'}} \Delta r^{max}_i + \frac{s}{2}) \leq \boldsymbol{d^h_{i'}} + \frac{s}{2} \tag{74}$$

$$\boldsymbol{e^*_j} - \boldsymbol{r^{t*}_{i'}} \boldsymbol{e^*_j} - \boldsymbol{c^{h*}_{i'}} \leq \boldsymbol{d^{h*}_{i'}} \tag{75}$$

($>$)   Let $s := \boldsymbol{r^t_{i'}} \Delta r^t_i m + \Delta r^{max}_i$ and $a := \Delta r^{ub}_i + \Delta r^t_i m \boldsymbol{r^t_{i'}}$. Note that $a$ is trivially positive since we assumed any parameter to be positive. Therefore, we can perform the following transformations:

$$\boldsymbol{e_j} - \boldsymbol{r^t_{i'}} \boldsymbol{e_j} - \boldsymbol{c^h_{i'}} \geq -\boldsymbol{d^h_{i'}} \tag{76}$$

$$\boldsymbol{e_j} - \boldsymbol{r^t_{i'}} \boldsymbol{e_j} - \boldsymbol{c^h_{i'}} + a + \frac{s}{2} - \frac{s}{2} \geq -\boldsymbol{d^h_{i'}} \tag{77}$$

$$\boldsymbol{e_j} + \Delta r^{ub}_i - \boldsymbol{r^t_{i'}} (\boldsymbol{e_j} + \Delta r^{ub}_i) - (\boldsymbol{c^h_{i'}} - \boldsymbol{r^t_{i'}} \Delta r^{max}_i + \frac{s}{2}) \geq -(\boldsymbol{d^h_{i'}} + \frac{s}{2}) \tag{78}$$

$$\boldsymbol{e^*_j} - \boldsymbol{r^{t*}_{i'}} \boldsymbol{e^*_j} - \boldsymbol{c^{h*}_{i'}} \geq -\boldsymbol{d^{h*}_{i'}} \tag{79}$$

We have shown in any of the twelve discussed cases that if a triple $r_{i'}(e_{j'}, e_{k'})$ with $i' \neq i$ or $j' \neq j$ or $k' \neq k$ was true in the model configuration prior to the induction step, then it is still true in the adjusted model configuration after the induction step. Hence, to show that ExpressivE can capture any self-loop-free graph, it remains to show that any triple that was false remains false after the induction step.

To verify that an initially false tripe $r_{i'}(e_{j'}, e_{k'})$ remains false we solely need to show that the embeddings of $r_{i'}$, $e_{j'}$ and $e_{k'}$ do not satisfy at least one of the Inequalities 1 or 2. We have to consider the following cases:

1. **Case** $k' \neq k$: Any changes to the dimension $\boldsymbol{v}(i, k)$ do not affect the dimension $\boldsymbol{v}(i', k')$. Therefore, if $r_{i'}(e_{j'}, e_{k'})$ for $k' \neq k$ was false before the induction step, it remains false after the induction step, as we solely alter dimension $(i, k)$.

2. **Case** $k' = k, i' = i$: In this case $j' \neq j$ needs to hold as the triple $r_i(e_j, e_k)$ was initially assumed to be true. We can easily show that in this case any triple remains false as follows:

   Let $s := (\Delta r^t_i + \boldsymbol{r^t_i}) \Delta r^t_i m + \Delta r^{max}_i$, then we can show that our induction step makes $r_i(e_{j'}, e_k)$ false as follows:

$$\boldsymbol{e_{j'}} - \boldsymbol{r^t_i} \boldsymbol{e_k} - \boldsymbol{c^h_i} \leq -\boldsymbol{d^h_i} \tag{80}$$

$$\begin{aligned} \boldsymbol{e_{j'}} - \boldsymbol{r^t_i} \boldsymbol{e_k} - \boldsymbol{c^h_i} + \Delta r^{max}_i (1 - 1 + \Delta r^t_i - \Delta r^t_i + \boldsymbol{r^t_i} \\ - \boldsymbol{r^t_i}) - \frac{s}{2} \leq -\boldsymbol{d^h_i} - \frac{s}{2} \end{aligned} \tag{81}$$

$$\begin{aligned} \boldsymbol{e_{j'}} + \Delta r^{max}_i - (\boldsymbol{r^t_i} + \Delta r^t_i)(\boldsymbol{e_k} + \Delta r^{max}_i) - (\boldsymbol{c^h_i} - \Delta r^t_i \Delta r^{max}_i \\ - \boldsymbol{r^t_i} \Delta r^{max}_i + \frac{s}{2}) \leq -\boldsymbol{d^h_i} - \frac{s}{2} \end{aligned} \tag{82}$$

$$\boldsymbol{e^*_{j'}} - \boldsymbol{r^{t*}_i} \boldsymbol{e^*_k} - \boldsymbol{c^{h*}_i} \leq -\boldsymbol{d^{h*}_i} \tag{83}$$

Since we started with the complete graph, any triple that is false was made false by an induction step. We have seen that if we apply our algorithm to make $r_i(e_j, e_k)$ false, then Inequality 7 holds.

Since we assume that $r_i(e_{j'}, e_k)$ was false prior to the current induction step and Inequality 7 describes how induction steps make triples false, we can follow that Inequality 80 needs to hold prior to this induction step. Next, we add in Inequality 81 terms that eliminate each other. Finally, in Inequality 82 we restructure the terms such that we can substitute them for the adjusted embedding vectors defined in 1-7. Through this substitution, we obtain Inequality 83, which reveals that the adjusted embeddings of $e_{j'}^*$ and $e_k^*$ do not lie within the adjusted hyper-parallelogram of relation $r_i$. Therefore, we have shown that the adjustments of the complete model configuration stated in Steps 1-7 preserve the false triples of this case to remain false.

3. **Case** $i' \neq i$: Any changes to the dimension $\boldsymbol{v}(i, k)$ do not affect the dimension $\boldsymbol{v}(i', k')$. Therefore, if $r_{i'}(e_{j'}, e_{k'})$ for $i' \neq i$ was false before the induction step, it remains false after the induction step, as we solely alter dimension $(i', k)$.

Hence, we have shown that we can make any self-loop-free triple false in the induction step while preserving the truth value of the remaining triples in $G$. To show fully expressiveness, it remains to show that we can capture any graph $G$ even with self-loops. We started our proof in the base case with a complete graph, which means that any self-loop was initially true. Furthermore, we have shown in Inequalities 8-15 and 72-79 that any true self-loop remains true after the induction step and that therefore any constructed complete model configuration captures any self-loop to be true. Since there are only $|R| * |E|$ possibilities to generate triples of the form $r_i(e_j, e_j)$ for any $r_i \in \boldsymbol{R}$ and $e_j \in \boldsymbol{E}$ and since we require just a single dimension where the embedding of the entity pair $e_j, e_j$ is outside of $r_i$'s hyper-parallelogram to make the triple $r_i(e_j, e_j)$ false, we can simply add a dimension per self-loop to our embeddings, whose sole purpose is to exclude one undesired self-loop $r_i(e_j, e_j)$. Therefore, ExpressivE can represent any possible graph $G$ in a complete model configuration of $O(|R| * |E|)$ dimensions, and our model is thus fully expressive in $O(|R| * |E|)$ dimensions.

## E  PROOF OF COMPOSITIONALLY DEFINED REGION

In this section, we prove Theorem 5.3, which will serve as further machinery for successive appendices. Since we are going to prove Theorem 5.3 by proving a more specific Theorem, we need to extend the notion of when a compositional definition pattern *holds* in the virtual triple space first such that we can employ it later in our proof. Definition E.1 describes when a compositional definition pattern holds in dependence of the spatial regions of its relations in the virtual triple space. The definition employs the notion of logical implication, i.e., if the body of a pattern is satisfied, then its head can be inferred.

**Definition E.1 (Truth of Compositional Definition in the Virtual Triple Space)** *Let $r_1(X, Y) \wedge r_2(Y, Z) \Leftrightarrow r_d(X, Z)$ be a compositional definition pattern over some relations $r_1, r_2, r_d \in \boldsymbol{R}$ and over arbitrary entities $X, Y, Z \in \boldsymbol{E}$. Furthermore, let $\boldsymbol{f_h}$ be a relation assignment function defined over $r_1$ and $r_2$. Moreover, let $\boldsymbol{s_d}$ be the spatial region of $r_d$ in the virtual triple space. The compositional definition pattern* holds *for the regions of the relations in the virtual triple space, i.e., for $\boldsymbol{f_h}(r_1)$, $\boldsymbol{f_h}(r_2)$ and $\boldsymbol{s_d}$, if: ($\Rightarrow$) for any entity assignment function $\boldsymbol{f_e}$ and virtual assignment function $\boldsymbol{f_v}$ over $\boldsymbol{f_e}$ if $\boldsymbol{f_v}(X, Y) \in \boldsymbol{f_h}(r_1)$ and $\boldsymbol{f_v}(Y, Z) \in \boldsymbol{f_h}(r_2)$, then $\boldsymbol{f_v}(X, Z)$ must be within the region $\boldsymbol{s_d}$ of $r_d$. ($\Leftarrow$) For any entity assignment function $\boldsymbol{f_e}$ and virtual assignment function $\boldsymbol{f_v}$ over $\boldsymbol{f_e}$ if $\boldsymbol{f_v}(X, Z)$ is within the region $\boldsymbol{s_d}$ of $r_d$, then there exists an entity assignment $\boldsymbol{f_e}(Y)$ such that $\boldsymbol{f_v}(X, Y) \in \boldsymbol{f_h}(r_1)$ and $\boldsymbol{f_v}(Y, Z) \in \boldsymbol{f_h}(r_2)$.*

Recall that Theorem 5.3 (reformulated in the definitions of Appendix C and Definition E.1) states that if $\phi := r_1(X, Y) \wedge r_2(Y, Z) \Leftrightarrow r_d(X, Z)$ is a compositional definition pattern defined over relations $r_1, r_2, r_d \in \boldsymbol{R}$ and if $\boldsymbol{f_h}$ is a relation assignment function that is defined over $r_1$ and $r_2$, then there exists a convex region $\boldsymbol{s_d}$ for $r_d$ in the virtual triple space $\mathbb{R}^{2d}$ such that $\phi$ holds for $\boldsymbol{f_h}(r_1)$, $\boldsymbol{f_h}(r_2)$ and $\boldsymbol{s_d}$. In particular, we are not only interested in proving the existence of the compositionally defined region $\boldsymbol{s_d}$, but we will even identify a system of inequalities that describes the shape of $\boldsymbol{s_d}$. Specifically, Theorem E.2 concretely characterizes the shape of $\boldsymbol{s_d}$, which we prove subsequently.

**Theorem E.2** *Let $r_1(X, Y) \wedge r_2(Y, Z) \Leftrightarrow r_d(X, Z)$ be a compositional definition pattern over some relations $r_1, r_2, r_d \in \boldsymbol{R}$ and over arbitrary entities $X, Y, Z \in \boldsymbol{E}$. Furthermore, let $\boldsymbol{f_h}$ be a relation assignment function that is defined over $r_1$ and $r_2$ such that for any $i \in \{1, 2\}$, $\boldsymbol{f_h}(r_i) = (\boldsymbol{c_i^{ht}}, \boldsymbol{d_i^{ht}}, \boldsymbol{r_i^{th}})$ with $\boldsymbol{c_i^{ht}} = (\boldsymbol{c_i^h} || \boldsymbol{c_i^t})$, $\boldsymbol{d_i^{ht}} = (\boldsymbol{d_i^h} || \boldsymbol{d_i^t})$, and $\boldsymbol{r_i^{th}} = (\boldsymbol{r_i^t} || \boldsymbol{r_i^h})$. Moreover, let*

*the slope vectors be positive, i.e., $r_i^{th} \succeq 0$ for $i \in \{1, 2\}$. If Inequalities 84-89 define the region $s_d$ of $r_d$ in the virtual triple space, then $r_1(X, Y) \wedge r_2(Y, Z) \Leftrightarrow r_d(X, Z)$ holds for $f_h(r_1)$, $f_h(r_2)$ and $s_d$ in the virtual triple space.*

$$(x - zr_1^t r_2^t - c_2^h r_1^t - c_1^h)^{|\cdot|} \preceq d_2^h r_1^t + d_1^h \tag{84}$$

$$(zr_2^t + c_2^h - xr_1^h - c_1^t)^{|\cdot|} \preceq d_1^t + d_2^h \tag{85}$$

$$(z - xr_1^h r_2^h - c_1^t r_2^h - c_2^t)^{|\cdot|} \preceq d_1^t r_2^h + d_2^t \tag{86}$$

$$(z + (c_1^h - x)r_2^h \oslash r_1^t - c_2^t)^{|\cdot|} \preceq d_1^h r_2^h \oslash r_1^t + d_2^t \tag{87}$$

$$(x(1 - r_1^h r_1^t) - c_1^t r_1^t - c_1^h)^{|\cdot|} \preceq d_1^t r_1^t + d_1^h \tag{88}$$

$$(z(1 - r_2^h r_2^t) - c_2^h r_2^h - c_2^t)^{|\cdot|} \preceq d_2^h r_2^h + d_2^t \tag{89}$$

**Proof**   Let $r_1(X, Y) \wedge r_2(Y, Z) \Leftrightarrow r_d(X, Z)$ be a compositional definition pattern over some relations $r_1, r_2, r_d \in R$ and over arbitrary entities $X, Y, Z \in E$. Furthermore, let $f_h$ be a relation assignment function that is defined over $r_1$ and $r_2$ such that for any $i \in \{1, 2\}$, $f_h(r_i) = (c_i^{ht}, d_i^{ht}, r_i^{th})$ with $c_i^{ht} = (c_i^h || c_i^t)$, $d_i^{ht} = (d_i^h || d_i^t)$, and $r_i^{th} = (r_i^t || r_i^h)$. Moreover, let the slope vectors be positive, i.e., $r_i^{th} \succeq 0$ for $i \in \{1, 2\}$.

What we want to show is that if Inequalities 84-89 define the region of $r_d$ in the virtual triple space, then $r_1(X, Y) \wedge r_2(Y, Z) \Leftrightarrow r_d(X, Z)$ holds in the virtual triple space, i.e., for any entity assignment function $f_e$ and virtual assignment function $f_v$ over $f_e$ if $f_v(X, Y) \in f_h(r_1)$ and $f_v(Y, Z) \in f_h(r_2)$, then $f_v(X, Z)$ must be within the region of $r_d$. To prove this, we will construct a system of inequalities first that describes $r_d$ and satisfies the compositional definition pattern. Afterward, we will show that the constructed system of inequalities has the same behavior as Inequalities 84-89, proving Theorem E.2.

($\Rightarrow$) First, we choose an arbitrary entity assignment function $f_e$ and virtual assignment function $f_v$ over $f_e$. We will henceforth denote the assigned entity embeddings with $f_e(X) = x$, $f_e(Y) = y$, and $f_e(Z) = z$ to state our proofs concisely. Next, we assume that the left part of $r_1(X, Y) \wedge r_2(Y, Z) \Leftrightarrow r_d(X, Z)$ is true, i.e., that $f_v(X, Y) \in f_h(r_1)$ and $f_v(Y, Z) \in f_h(r_2)$ hold. This means concretely that we can instantiate the following inequalities from Inequalities 1-2:

$$x - c_1^h - r_1^t \odot y - d_1^h \preceq 0 \tag{90}$$

$$x - c_1^h - r_1^t \odot y + d_1^h \succeq 0 \tag{91}$$

$$y - c_1^t - r_1^h \odot x - d_1^t \preceq 0 \tag{92}$$

$$y - c_1^t - r_1^h \odot x + d_1^t \succeq 0 \tag{93}$$

$$y - c_2^h - r_2^t \odot z - d_2^h \preceq 0 \tag{94}$$

$$y - c_2^h - r_2^t \odot z + d_2^h \succeq 0 \tag{95}$$

$$z - c_2^t - r_2^h \odot y - d_2^t \preceq 0 \tag{96}$$

$$z - c_2^t - r_2^h \odot y + d_2^t \succeq 0 \tag{97}$$

Our next goal is to construct a system of inequalities that makes $r_d(X, Z)$ — the right part of the pattern — true, i.e., that defines the region of $r_d$ such that $f_v(X, Z)$ lies within it. To reach this goal, we substitute Inequalities 90-97 into each other to receive a system of inequalities that (1) has the same behavior as the initial set and (2) does not contain the entity embedding $y$. Since we have in the beginning assumed that the slope vectors are positive, we can substitute Inequalities 90-97 into each other as follows:

1. 95 in 91 and 94 in 90 leading to 98
2. 95 in 92 and 94 in 93 leading to 99
3. 93 in 97 and 92 in 96 leading to 100
4. 91 in 96 and 90 in 97 leading to 101.

5. 90 in 92 and 93 in 91 leading to 102.

6. 94 in 96 and 97 in 95 leading to 103.

7. 90 in 91 leading to 104.

8. 93 in 92 leading to 105.

9. 94 in 95 leading to 106.

10. 97 in 96 leading to 107.

These substitutions result in a system of inequalities with the same behavior as the initial system of inequalities. We have listed the result of these substitutions in Inequalities 98-107.

$$(x - zr_1^t r_2^t - c_2^h r_1^t - c_1^h)^{|\cdot|} \preceq d_2^h r_1^t + d_1^h \tag{98}$$

$$(zr_2^t + c_2^h - xr_1^h - c_1^t)^{|\cdot|} \preceq d_1^t + d_2^h \tag{99}$$

$$(z - xr_1^h r_2^h - c_1^t r_2^h - c_2^t)^{|\cdot|} \preceq d_1^t r_2^h + d_2^t \tag{100}$$

$$(z + (c_1^h - x)r_2^h \oslash r_1^t - c_2^t)^{|\cdot|} \preceq d_1^h r_2^h \oslash r_1^t + d_2^t \tag{101}$$

$$(x(1 - r_1^h r_1^t) - c_1^t r_1^t - c_1^h)^{|\cdot|} \preceq d_1^t r_1^t + d_1^h \tag{102}$$

$$(z(1 - r_2^h r_2^t) - c_2^h r_2^h - c_2^t)^{|\cdot|} \preceq d_2^h r_2^h + d_2^t \tag{103}$$

$$d_1^h \preceq -d_1^h \tag{104}$$

$$d_1^t \preceq -d_1^t \tag{105}$$

$$d_2^h \preceq -d_2^h \tag{106}$$

$$d_2^t \preceq -d_2^t \tag{107}$$

Note that Inequalities 98-103 are equivalent to Inequalities 84-89 and that Inequalities 104-107 are tautologies since any width embedding $d_i^p$ is positive by the definition of the ExpressivE model. Therefore, Inequalities 98-107 and Inequalities 84-89 have the same behavior, as required. It remains to show that Inequalities 98-107 define a region $s_d$ containing $f_v(X, Z)$ if $f_v(X, Y) \in f_h(r_1)$ and $f_v(Y, Z) \in f_h(r_2)$. This is trivially true since Inequalities 98-107 directly follow from Inequalities 90-97, which are instantiations of Inequalities 1-2 representing $f_v(X, Y) \in f_h(r_1)$ and $f_v(Y, Z) \in f_h(r_2)$.

Reading the proof bottom-up proves the other direction ($\Leftarrow$), i.e., if $f_v(X, Z)$ is in $s_d$, then there exists an entity assignment $f_e(Y) = y$ such that $f_v(X, Y) \in f_h(r_1)$ and $f_v(Y, Z) \in f_h(r_2)$. Thereby, we have successfully shown that if Inequalities 84-89 describe the region $s_d$ of relation $r_d$ in the virtual triple space, then $r_1(X, Y) \wedge r_2(Y, Z) \Leftrightarrow r_d(X, Z)$ holds for $f_h(r_1)$, $f_h(r_2)$, and $s_d$ in the virtual triple space. □

We have proven Theorem E.2 in this section, i.e., that Inequalities 84-89 define the compositionally defined region for positive slope vectors. The proof works vice versa for any other sign of slope vectors, except that the substitutions of Inequalities 90-97 may vary due to the different signs of slope vectors. Note that by proving Theorem E.2, we have also proven Theorem 5.3 — i.e., that there exists a convex region that describes the compositionally defined region $s_d$ — since (1) we have characterized the compositionally defined region and thereby implicitly proven its existence and since (2) Inequalities 84-89 trivially form a convex region.

## F    DETAILS ON CAPTURING PATTERNS EXACTLY

Before we prove the inference capabilities of ExpressivE in this section, we formally define the considered patterns in Definition F.1.

**Definition F.1** *In accordance with Sun et al. (2019); Abboud et al. (2020), we define the following inference patterns:*

- *Patterns of the form $r_1(X, Y) \Rightarrow r_1(Y, X)$ with $r_1 \in \mathbf{R}$ are called* symmetry patterns.

- *Patterns of the form $r_1(X, Y) \Rightarrow \neg r_1(Y, X)$ with $r_1 \in \boldsymbol{R}$ are called* anti-symmetry patterns.

- *Patterns of the form $r_1(X, Y) \Leftrightarrow r_2(Y, X)$ with $r_1, r_2 \in \boldsymbol{R}$ and $r_1 \neq r_2$ are called* inversion patterns.

- *Patterns of the form $r_1(X, Y) \wedge r_2(Y, Z) \Rightarrow r_3(X, Z)$ with $r_1, r_2, r_3 \in \boldsymbol{R}$ and $r_1 \neq r_2 \neq r_3$ are called* general composition patterns.

- *Patterns of the form $r_1(X, Y) \wedge r_2(Y, Z) \Leftrightarrow r_d(X, Z)$ with $r_1, r_2, r_d \in \boldsymbol{R}$ and $r_1 \neq r_2 \neq r_d$ are called* compositional definition patterns.

- *Patterns of the form $r_1(X, Y) \Rightarrow r_2(X, Y)$ with $r_1, r_2 \in \boldsymbol{R}$ and $r_1 \neq r_2$ are called* hierarchy patterns.

- *Patterns of the form $r_1(X, Y) \wedge r_2(X, Y) \Rightarrow r_3(X, Y)$ with $r_1, r_2, r_3 \in \boldsymbol{R}$ and $r_1 \neq r_2 \neq r_3$ are called* intersection patterns.

- *Patterns of the form $r_1(X, Y) \wedge r_2(X, Y) \Rightarrow \bot$ with $r_1, r_2 \in \boldsymbol{R}$ and $r_1 \neq r_2$ are called* mutual exclusion patterns.

With all definitions in place, we prove the exactness part of Theorems 5.2 and 5.4, i.e., that ExpressivE captures all patterns from Table 1 exactly. Specifically, we do not solely prove that ExpressivE captures the patterns of Table 1 exactly, but that ExpressivE captures these patterns exactly iff its relation hyper-parallelograms meet the properties intuitively described in Section 5. Next, in Section G, we prove that ExpressivE captures patterns exactly and exclusively. For the upcoming proofs, we employ the definitions and formal specifications of Sections C and E:

**Proposition F.1 (Symmetry (Exactly))** *Let $\boldsymbol{m_h} = (\boldsymbol{M}, \boldsymbol{f_h})$ be a relation configuration and $r_1 \in \boldsymbol{R}$ be a symmetric relation, i.e., $r_1(X, Y) \Rightarrow r_1(Y, X)$ holds for any entities $X, Y \in \boldsymbol{E}$. Then $\boldsymbol{m_h}$ captures $r_1(X, Y) \Rightarrow r_1(Y, X)$ exactly iff $r_1$'s relation hyper-parallelogram $\boldsymbol{f_h(r_1)}$ is symmetric across the identity line of any correlation subspace.*

**Proof** $\Rightarrow$ For the first direction, what is to be shown is that if $r_1$'s relation hyper-parallelogram $\boldsymbol{f_h(r_1)}$ is symmetric across the identity line of any correlation subspace, then $\boldsymbol{m_h}$ captures $r_1(X, Y) \Rightarrow r_1(Y, X)$ exactly. We show this by contradiction. Thus, we first assume that $r_1$'s corresponding relation hyper-parallelogram $\boldsymbol{f_h(r_1)}$ of $\boldsymbol{m_h}$ is symmetric across the identity line for any correlation subspace $s_i$. Now to the contrary, we assume that $\boldsymbol{m_h}$ does not capture $r_1(X, Y) \Rightarrow r_1(Y, X)$ exactly. Then, due to the symmetry of the hyper-parallelogram across the identity line in any correlation subspace $s_i$, for any virtual assignment function $\boldsymbol{f_v}$ it holds that if $\boldsymbol{f_v(e_x, e_y)} \in \boldsymbol{f_h(r_1)}$ for arbitrary entities $e_x, e_y \in \boldsymbol{E}$, then $\boldsymbol{f_v(e_y, e_x)} \in \boldsymbol{f_h(r_1)}$. Yet, by the definition of capturing patterns exactly, this means that $\boldsymbol{m_h}$ captures $r_1(X, Y) \Rightarrow r_1(Y, X)$ exactly. This is a contradiction to the initial assumption that $\boldsymbol{m_h}$ does not capture $r_1(X, Y) \Rightarrow r_1(Y, X)$ exactly, proving the $\Rightarrow$ part of the proposition.

$\Leftarrow$ For the second direction, what is to be shown is that if $\boldsymbol{m_h}$ captures $r_1(X, Y) \Rightarrow r_1(Y, X)$ exactly, then $r_1$'s relation hyper-parallelogram $\boldsymbol{f_h(r_1)}$ is symmetric across the identity line of any correlation subspace. We show this by contradiction. Thus, we first assume that $\boldsymbol{m_h}$ captures $r_1(X, Y) \Rightarrow r_1(Y, X)$ exactly, i.e., for any instantiation of $\boldsymbol{f_e}$ and $\boldsymbol{f_v}$ over $\boldsymbol{f_e}$ if $\boldsymbol{f_v(e_x, e_y)} \in \boldsymbol{f_h(r_1)}$, then $\boldsymbol{f_v(e_y, e_x)} \in \boldsymbol{f_h(r_1)}$. Now to the contrary, we assume that $r_1$'s corresponding relation hyper-parallelogram $\boldsymbol{f_h(r_1)}$ of $\boldsymbol{m_h}$ is not symmetric across the identity line in at least one correlation subspace $s_i$. Then, since $\boldsymbol{f_h(r_1)}$ is not symmetric across the identity line in $s_i$, there is an instantiation of $\boldsymbol{f_v}$ and $\boldsymbol{f_e}$ such that $\boldsymbol{f_v(e_x, e_y)} \in \boldsymbol{f_h(r_1)}$ and $\boldsymbol{f_v(e_y, e_x)} \notin \boldsymbol{f_h(r_1)}$ for some entities $e_x, e_y \in \boldsymbol{E}$. Yet, by the definition of capturing patterns exactly, this means that $\boldsymbol{m_h}$ does not capture $r_1(X, Y) \Rightarrow r_1(Y, X)$ exactly. This is a contradiction to the initial assumption that $\boldsymbol{m_h}$ captures $r_1(X, Y) \Rightarrow r_1(Y, X)$ exactly, proving the $\Leftarrow$ part of the proposition. $\square$

**Proposition F.2 (Anti-Symmetry (Exactly))** *Let $\boldsymbol{m_h} = (\boldsymbol{M}, \boldsymbol{f_h})$ be a relation configuration and $r_1 \in \boldsymbol{R}$ be an anti-symmetric relation, i.e., $r_1(X, Y) \Rightarrow \neg r_1(Y, X)$ holds for any entities $X, Y \in \boldsymbol{E}$. Then $\boldsymbol{m_h}$ captures $r_1(X, Y) \Rightarrow \neg r_1(Y, X)$ exactly iff $r_1$'s relation hyper-parallelogram $\boldsymbol{f_h(r_1)}$ is not symmetric across the identity line in at least one correlation subspace.*

Proposition F.2 can be proven analogously to Proposition F.1. Therefore, its proof has been omitted.

**Proposition F.3 (Inversion (Exactly))** *Let $m_h = (M, f_h)$ be a relation configuration and $r_1, r_2 \in R$ be relations where $r_1(X,Y) \Leftrightarrow r_2(Y,X)$ holds for any entities $X, Y \in E$. Then $m_h$ captures $r_1(X,Y) \Leftrightarrow r_2(Y,X)$ exactly iff $f_h(r_1)$ is the mirror image across the identity line of $f_h(r_2)$ for any correlation subspace.*

**Proof** $\Rightarrow$ For the first direction, what is to be shown is that if the relation hyper-parallelogram $f_h(r_1)$ is the mirror image across the identity line of $f_h(r_2)$ for any correlation subspace, then $m_h$ captures $r_1(X,Y) \Leftrightarrow r_2(Y,X)$ exactly. We show this by contradiction. Thus, we first assume that $r_1$'s corresponding relation hyper-parallelogram $f_h(r_1)$ of $m_h$ is the mirror image across the identity line of $f_h(r_2)$ for any correlation subspace $s_i$. Now to the contrary, we assume that $m_h$ does not capture $r_1(X,Y) \Leftrightarrow r_2(Y,X)$ exactly. Then, due to $f_h(r_1)$ being the mirror image of $f_h(r_2)$ in any correlation subspace $s_i$, for any virtual assignment function $f_v$ it holds that if $f_v(e_x, e_y) \in f_h(r_1)$ for arbitrary entities $e_x, e_y \in E$, then $f_v(e_y, e_x) \in f_h(r_2)$. Yet, by the definition of capturing patterns exactly, this means that $m_h$ captures $r_1(X,Y) \Leftrightarrow r_2(Y,X)$ exactly. This is a contradiction to the initial assumption that $m_h$ does not capture $r_1(X,Y) \Leftrightarrow r_2(Y,X)$ exactly, proving the $\Rightarrow$ part of the proposition.

$\Leftarrow$ For the second direction, what is to be shown is that if $m_h$ captures $r_1(X,Y) \Leftrightarrow r_2(Y,X)$ exactly, then the relation hyper-parallelogram $f_h(r_1)$ is the mirror image across the identity line of $f_h(r_2)$ for any correlation subspace. We show this by contradiction. Thus, we first assume that $m_h$ captures $r_1(X,Y) \Leftrightarrow r_2(Y,X)$ exactly, i.e., for any instantiation of $f_e$ and $f_v$ over $f_e$ if $f_v(e_x, e_y) \in f_h(r_1)$, then $f_v(e_y, e_x) \in f_h(r_2)$. Now to the contrary, we assume that $r_1$'s corresponding relation hyper-parallelogram $f_h(r_1)$ of $m_h$ is not the mirror image across the identity line of $f_h(r_2)$ for at least one correlation subspace $s_i$. Then, since $f_h(r_1)$ is not the mirror image across the identity line of $f_h(r_2)$ in $s_i$, there is an instantiation of $f_v$ and $f_e$ such that $f_v(e_x, e_y) \in f_h(r_1)$ and $f_v(e_y, e_x) \notin f_h(r_2)$ for some entities $e_x, e_y \in E$. Yet, by the definition of capturing patterns exactly, this means that $m_h$ does not capture $r_1(X,Y) \Leftrightarrow r_2(Y,X)$ exactly. This is a contradiction to the initial assumption that $m_h$ captures $r_1(X,Y) \Leftrightarrow r_2(Y,X)$ exactly, proving the $\Leftarrow$ part of the proposition. $\square$

**Proposition F.4 (Hierarchy (Exactly))** *Let $m_h = (M, f_h)$ be a relation configuration and $r_1, r_2 \in R$ be relations where $r_1(X,Y) \Rightarrow r_2(X,Y)$ holds for any entities $X, Y \in E$. Then $m_h$ captures $r_1(X,Y) \Rightarrow r_2(X,Y)$ exactly iff $f_h(r_1)$ is subsumed by $f_h(r_2)$ for any correlation subspace.*

**Proof** $\Rightarrow$ For the first direction, what is to be shown is that if the relation hyper-parallelogram $f_h(r_1)$ is subsumed by $f_h(r_2)$ for any correlation subspace, then $m_h$ captures $r_1(X,Y) \Rightarrow r_2(X,Y)$ exactly. We show this by contradiction. Thus, we first assume that $r_1$'s corresponding relation hyper-parallelogram $f_h(r_1)$ of $m_h$ is subsumed by $f_h(r_2)$ for any correlation subspace $s_i$. Now to the contrary, we assume that $m_h$ does not capture $r_1(X,Y) \Rightarrow r_2(X,Y)$ exactly. Then, due to $f_h(r_1)$ being a subset of $f_h(r_2)$ in any correlation subspace $s_i$, for any virtual assignment function $f_v$ it holds that if $f_v(e_x, e_y) \in f_h(r_1)$ for arbitrary entities $e_x, e_y \in E$, then $f_v(e_x, e_y) \in f_h(r_2)$. Yet, by the definition of capturing patterns exactly, this means that $m_h$ captures $r_1(X,Y) \Rightarrow r_2(X,Y)$ exactly. This is a contradiction to the initial assumption that $m_h$ does not capture $r_1(X,Y) \Rightarrow r_2(X,Y)$ exactly, proving the $\Rightarrow$ part of the proposition.

$\Leftarrow$ For the second direction, what is to be shown is that if $m_h$ captures $r_1(X,Y) \Rightarrow r_2(X,Y)$ exactly, then the relation hyper-parallelogram $f_h(r_1)$ is subsumed by $f_h(r_2)$ for any correlation subspace. We show this by contradiction. Thus, we first assume that $m_h$ captures $r_1(X,Y) \Rightarrow r_2(X,Y)$ exactly, i.e., for any instantiation of $f_e$ and $f_v$ over $f_e$ if $f_v(e_x, e_y) \in f_h(r_1)$, then $f_v(e_x, e_y) \in f_h(r_2)$. Now to the contrary, we assume that $r_1$'s corresponding relation hyper-parallelogram $f_h(r_1)$ of $m_h$ is not subsumed by $f_h(r_2)$ for at least one correlation subspace $s_i$. Then, since $f_h(r_1)$ is subsumed by $f_h(r_2)$ in $s_i$, there is an instantiation of $f_v$ and $f_e$ such that $f_v(e_x, e_y) \in f_h(r_1)$ and $f_v(e_x, e_y) \notin f_h(r_2)$ for some entities $e_x, e_y \in E$. Yet, by the definition of capturing patterns exactly, this means that $m_h$ does not capture $r_1(X,Y) \Rightarrow r_2(X,Y)$ exactly. This is a contradiction to the initial assumption that $m_h$ captures $r_1(X,Y) \Rightarrow r_2(X,Y)$ exactly, proving the $\Leftarrow$ part of the proposition. $\square$

**Proposition F.5 (Intersection (Exactly))** *Let* $m_h = (M, f_h)$ *be a relation configuration and* $r_1, r_2, r_3 \in R$ *be relations where* $r_1(X, Y) \wedge r_2(X, Y) \Rightarrow r_3(X, Y)$ *holds for any entities* $X, Y \in E$. *Then* $m_h$ *captures* $r_1(X, Y) \wedge r_2(X, Y) \Rightarrow r_3(X, Y)$ *exactly iff the intersection of* $f_h(r_1)$ *and* $f_h(r_2)$ *is subsumed by* $f_h(r_3)$ *for any correlation subspace.*

**Proof** $\Rightarrow$ For the first direction, what is to be shown is that if the intersection of $f_h(r_1)$ and $f_h(r_2)$ is subsumed by $f_h(r_3)$ for any correlation subspace, then $m_h$ captures $r_1(X, Y) \wedge r_2(X, Y) \Rightarrow r_3(X, Y)$ exactly. We show this by contradiction. Thus, we first assume that the intersection of $f_h(r_1)$ and $f_h(r_2)$ of $m_h$ is subsumed by $f_h(r_3)$ for any correlation subspace $s_i$. Now to the contrary, we assume that $m_h$ does not capture $r_1(X, Y) \wedge r_2(X, Y) \Rightarrow r_3(X, Y)$ exactly. Then, due to the intersection of $f_h(r_1)$ and $f_h(r_2)$ being a subset of $f_h(r_3)$ in any correlation subspace $s_i$, for any virtual assignment function $f_v$ it holds that if $f_v(e_x, e_y) \in f_h(r_1)$ and $f_v(e_x, e_y) \in f_h(r_2)$ for arbitrary entities $e_x, e_y \in E$, then $f_v(e_x, e_y) \in f_h(r_3)$. Yet, by the definition of capturing patterns exactly, this means that $m_h$ captures $r_1(X, Y) \wedge r_2(X, Y) \Rightarrow r_3(X, Y)$ exactly. This is a contradiction to the initial assumption that $m_h$ does not capture $r_1(X, Y) \wedge r_2(X, Y) \Rightarrow r_3(X, Y)$ exactly, proving the $\Rightarrow$ part of the proposition.

$\Leftarrow$ For the second direction, what is to be shown is that if $m_h$ captures $r_1(X, Y) \wedge r_2(X, Y) \Rightarrow r_3(X, Y)$ exactly, then the intersection of $f_h(r_1)$ and $f_h(r_2)$ is subsumed by $f_h(r_3)$ for any correlation subspace. We show this by contradiction. Thus, we first assume that $m_h$ captures $r_1(X, Y) \wedge r_2(X, Y) \Rightarrow r_3(X, Y)$ exactly, i.e., for any instantiation of $f_e$ and $f_v$ over $f_e$ if $f_v(e_x, e_y) \in f_h(r_1)$ and $f_v(e_x, e_y) \in f_h(r_2)$, then $f_v(e_x, e_y) \in f_h(r_3)$. Now to the contrary, we assume that the intersection of $f_h(r_1)$ and $f_h(r_2)$ is not subsumed by $f_h(r_3)$ for at least one correlation subspace $s_i$. Then, since the intersection of $f_h(r_1)$ and $f_h(r_2)$ is not subsumed by $f_h(r_3)$ in $s_i$, there is an instantiation of $f_v$ and $f_e$ such that $f_v(e_x, e_y) \in f_h(r_1)$ and $f_v(e_x, e_y) \in f_h(r_2)$ but $f_v(e_x, e_y) \notin f_h(r_3)$ for some entities $e_x, e_y \in E$. Yet, by the definition of capturing patterns exactly, this means that $m_h$ does not capture $r_1(X, Y) \wedge r_2(X, Y) \Rightarrow r_3(X, Y)$ exactly. This is a contradiction to the initial assumption that $m_h$ captures $r_1(X, Y) \wedge r_2(X, Y) \Rightarrow r_3(X, Y)$ exactly, proving the $\Leftarrow$ part of the proposition. $\square$

**Proposition F.6 (Mutual Exclusion (Exactly))** *Let* $m_h = (M, f_h)$ *be a relation configuration and* $r_1, r_2 \in R$ *be relations where* $r_1(X, Y) \wedge r_2(X, Y) \Rightarrow \bot$ *holds for any entities* $X, Y \in E$. *Then* $m_h$ *captures* $r_1(X, Y) \wedge r_2(X, Y) \Rightarrow \bot$ *exactly iff* $f_h(r_1)$ *and* $f_h(r_2)$ *do not intersect in at least one correlation subspace.*

**Proof** $\Rightarrow$ For the first direction, what is to be shown is that if the relation hyper-parallelograms $f_h(r_1)$ and $f_h(r_2)$ do not intersect in at least one correlation subspace, then $m_h$ captures $r_1(X, Y) \wedge r_2(X, Y) \Rightarrow \bot$ exactly. We show this by contradiction. Thus, we first assume that $f_h(r_1)$ and $f_h(r_2)$ of $m_h$ do not intersect in at least one correlation subspace $s_i$. Now to the contrary, we assume that $m_h$ does not capture $r_1(X, Y) \wedge r_2(X, Y) \Rightarrow \bot$ exactly. Then, since $f_h(r_1)$ and $f_h(r_2)$ do not intersect in at least one correlation subspace $s_i$, for any virtual assignment function $f_v$ it holds that if $f_v(e_x, e_y) \in f_h(r_1)$ for arbitrary entities $e_x, e_y \in E$, then $f_v(e_x, e_y) \notin f_h(r_2)$. Yet, by the definition of capturing patterns exactly, this means that $m_h$ captures $r_1(X, Y) \wedge r_2(X, Y) \Rightarrow \bot$ exactly. This is a contradiction to the initial assumption that $m_h$ does not capture $r_1(X, Y) \wedge r_2(X, Y) \Rightarrow \bot$ exactly, proving the $\Rightarrow$ part of the proposition.

$\Leftarrow$ For the second direction, what is to be shown is that if $m_h$ captures $r_1(X, Y) \wedge r_2(X, Y) \Rightarrow \bot$ exactly, then the relation hyper-parallelograms $f_h(r_1)$ and $f_h(r_2)$ do not intersect in at least one correlation subspace. We show this by contradiction. Thus, we first assume that $m_h$ captures $r_1(X, Y) \wedge r_2(X, Y) \Rightarrow \bot$ exactly, i.e., for any instantiation of $f_e$ and $f_v$ over $f_e$ if $f_v(e_x, e_y) \in f_h(r_1)$, then $f_v(e_x, e_y) \notin f_h(r_2)$ and if $f_v(e_x, e_y) \in f_h(r_2)$, then $f_v(e_x, e_y) \notin f_h(r_1)$. Now to the contrary, we assume that $r_1$'s corresponding relation hyper-parallelogram $f_h(r_1)$ of $m_h$ intersects with $f_h(r_2)$ in any correlation subspace. Then, since $f_h(r_1)$ intersects with $f_h(r_2)$ in any correlation subspace, there is an instantiation of $f_v$ and $f_e$ such that $f_v(e_x, e_y) \in f_h(r_1)$ and $f_v(e_x, e_y) \in f_h(r_2)$ for some entities $e_x, e_y \in E$. Yet, by the definition of capturing patterns exactly, this means that $m_h$ does not capture $r_1(X, Y) \wedge r_2(X, Y) \Rightarrow \bot$ exactly. This is a contradiction to the initial assumption that $m_h$ captures $r_1(X, Y) \wedge r_2(X, Y) \Rightarrow \bot$ exactly, proving the $\Leftarrow$ part of the proposition. $\square$

**Proposition F.7 (General Composition (Exactly))** *Let $r_1, r_2, r_3 \in \boldsymbol{R}$ be relations and let $\boldsymbol{m_h} = (\boldsymbol{M}, \boldsymbol{f_h})$ be a relation configuration, where $\boldsymbol{f_h}$ is defined over $r_1, r_2$, and $r_3$. Furthermore let $r_3$ be the composite relation of $r_1$ and $r_2$, i.e., $r_1(X, Y) \wedge r_2(Y, Z) \Rightarrow r_3(X, Z)$ holds for any entities $X, Y, Z \in \boldsymbol{E}$. Then $\boldsymbol{m_h}$ captures $r_1(X, Y) \wedge r_2(Y, Z) \Rightarrow r_3(X, Z)$ iff the relation hyperparallelogram $\boldsymbol{f_h}(r_3)$ subsumes the compositionally defined region $s_d$ defined by $\boldsymbol{f_h}(r_1)$ and $\boldsymbol{f_h}(r_2)$ for any correlation subspace.*

**Proof** $\Rightarrow$ For the first direction, assume that the compositionally defined region defined by $\boldsymbol{f_h}(r_1)$ and $\boldsymbol{f_h}(r_2)$ is subsumed by $\boldsymbol{f_h}(r_3)$ for any correlation subspace. What is to be shown is that $\boldsymbol{m_h}$ captures $r_1(X, Y) \wedge r_2(Y, Z) \Rightarrow r_3(X, Z)$ exactly. Our proof for this direction is based on the following three results:

1. For an auxiliary relation $r_d \in \boldsymbol{R}$, there exists a convex region $s_d$ in the virtual triple space such that $r_1(X, Y) \wedge r_2(Y, Z) \Leftrightarrow r_d(X, Z)$ holds for $\boldsymbol{f_h}(r_1)$, $\boldsymbol{f_h}(r_2)$, and $s_d$ in any correlation subspace (Theorem E.2).

2. $\boldsymbol{f_h}(r_1)$ subsumes $s_d$ iff $\boldsymbol{m_h}$ captures $r_d(X, Y) \Rightarrow r_3(X, Y)$ exactly (Proposition F.4).

3. $r_1(X, Y) \wedge r_2(Y, Z) \Rightarrow r_3(X, Z)$ logically follows from $\{r_1(X, Y) \wedge r_2(Y, Z) \Leftrightarrow r_d(X, Z), \ r_d(X, Y) \Rightarrow r_3(X, Y)\}$.

For (1), observe that based on Theorem E.2, we know that we can define an auxiliary relation $r_d \in \boldsymbol{R}$ with area $s_d$ such that $r_1(X, Y) \wedge r_2(Y, Z) \Leftrightarrow r_d(X, Z)$ holds for $\boldsymbol{f_h}(r_1)$, $\boldsymbol{f_h}(r_2)$, and $s_d$, i.e., such that $s_d$ is the compositionally defined region of $\boldsymbol{f_h}(r_1)$ and $\boldsymbol{f_h}(r_2)$. For (2), as shown in Proposition F.4, $\boldsymbol{m_h}$ captures $r_d(X, Y) \Rightarrow r_3(X, Y)$ exactly iff $\boldsymbol{f_h}(r_3)$ subsumes $r_d$'s area $s_d$. Therefore, we have shown that if $\boldsymbol{f_h}(r_3)$ subsumes $s_d$, and if $s_d$ is the compositionally defined region of $\boldsymbol{f_h}(r_1)$ and $\boldsymbol{f_h}(r_2)$, then $r_d(X, Y) \Rightarrow r_3(X, Y)$ and $r_1(X, Y) \wedge r_2(Y, Z) \Leftrightarrow r_d(X, Z)$ holds for $\boldsymbol{f_h}(r_1)$, $\boldsymbol{f_h}(r_2)$, $\boldsymbol{f_h}(r_3)$ and $s_d$. Together with the fact that $\boldsymbol{f_h}$ is only defined over $r_1$, $r_2$, and $r_3$, we can infer that $\boldsymbol{m_h}$ exactly captures any pattern — solely consisting of $r_1$, $r_2$, and $r_3$ — that follows from $\psi = \{r_1(X, Y) \wedge r_2(Y, Z) \Leftrightarrow r_d(X, Z), \ r_d(X, Y) \Rightarrow r_3(X, Y)\}$. For (3), by logical deduction, the following statement holds: $\psi \models r_1(X, Y) \wedge r_2(Y, Z) \Rightarrow r_3(X, Y)$. Since $r_1(X, Y) \wedge r_2(Y, Z) \Rightarrow r_3(X, Z)$ (i) solely consists of $r_1$, $r_2$, and $r_3$ and (ii) follows from $\psi$, we have proven that $\boldsymbol{m_h}$ captures $r_1(X, Y) \wedge r_2(Y, Z) \Rightarrow r_3(X, Z)$ exactly if $\boldsymbol{f_h}(r_3)$ subsumes $s_d$, proving the $\Rightarrow$ part of the proposition.

$\Leftarrow$ For the second direction, what is to be shown is that if $\boldsymbol{m_h}$ captures $r_1(X, Y) \wedge r_2(Y, Z) \Rightarrow r_3(X, Z)$ exactly, then the compositionally defined region defined by $\boldsymbol{f_h}(r_1)$ and $\boldsymbol{f_h}(r_2)$ is subsumed by $\boldsymbol{f_h}(r_3)$ for any correlation subspace. We prove this by contradiction. Thus assume that $\boldsymbol{m_h}$ captures $r_1(X, Y) \wedge r_2(Y, Z) \Rightarrow r_3(X, Z)$ exactly, i.e., for any instantiation of $\boldsymbol{f_e}$ and $\boldsymbol{f_v}$ over $\boldsymbol{f_e}$ if $\boldsymbol{f_v}(e_x, e_y) \in \boldsymbol{f_h}(r_1)$ and $\boldsymbol{f_v}(e_y, e_z) \in \boldsymbol{f_h}(r_2)$, then $\boldsymbol{f_v}(e_x, e_z) \in \boldsymbol{f_h}(r_3)$. Now to the contrary, we assume that $r_3$'s corresponding relation hyper-parallelogram $\boldsymbol{f_h}(r_3)$ of $\boldsymbol{m_h}$ does not subsume the compositionally defined region $s_d$ in at least one correlation subspace. The following three points will be used to construct a counter-example: (1) we have shown in Theorem E.2 that we can define an auxiliary relation $r_d \in \boldsymbol{R}$ with area $s_d$ such that $r_1(X, Y) \wedge r_2(Y, Z) \Leftrightarrow r_d(X, Z)$ holds for $\boldsymbol{f_h}(r_1)$, $\boldsymbol{f_h}(r_2)$, and $s_d$, (2) $r_1(X, Y) \wedge r_2(Y, Z) \Rightarrow r_3(X, Z)$ logically follows from $\{r_1(X, Y) \wedge r_2(Y, Z) \Leftrightarrow r_d(X, Z), \ r_d(X, Y) \Rightarrow r_3(X, Y)\}$, stating together with Point (1) and Proposition F.4 that $r_3$ needs to subsume $r_d$'s area $s_d$ such that $\boldsymbol{m_h}$ can capture $r_1(X, Y) \wedge r_2(Y, Z) \Rightarrow r_3(X, Z)$ exactly, and (3) we have initially assumed that $\boldsymbol{f_h}(r_3)$ does not subsume $s_d$. From (1)-(3) we can infer that there exists an instantiation of $\boldsymbol{f_v}$ and $\boldsymbol{f_e}$ such that $\boldsymbol{f_v}(e_x, e_y) \in \boldsymbol{f_h}(r_1)$ and $\boldsymbol{f_v}(e_y, e_z) \in \boldsymbol{f_h}(r_2)$ but $\boldsymbol{f_v}(e_x, e_z) \notin \boldsymbol{f_h}(r_3)$ for some entities $e_x, e_y, e_z \in \boldsymbol{E}$. Yet, by the definition of capturing patterns exactly, this means that $\boldsymbol{m_h}$ does not capture $r_1(X, Y) \wedge r_2(Y, Z) \Rightarrow r_3(X, Z)$ exactly. This is a contradiction to the initial assumption that $\boldsymbol{m_h}$ captures $r_1(X, Y) \wedge r_2(Y, Z) \Rightarrow r_3(X, Z)$ exactly, proving the $\Leftarrow$ part of the proposition. $\square$

**Proposition F.8 (Compositional Definition (Exactly))** *Let $r_1, r_2, r_d \in \boldsymbol{R}$ be relations and let $\boldsymbol{m_h} = (\boldsymbol{M}, \boldsymbol{f_h})$ be a relation configuration, where $\boldsymbol{f_h}$ is defined over $r_1, r_2$, and $r_d$. Furthermore let $r_d$ be the compositionally defined relation of $r_1$ and $r_2$, i.e., $r_1(X, Y) \wedge r_2(Y, Z) \Leftrightarrow r_d(X, Z)$ holds for any entities $X, Y, Z \in \boldsymbol{E}$. Then $\boldsymbol{m_h}$ captures $r_1(X, Y) \wedge r_2(Y, Z) \Leftrightarrow r_d(X, Z)$ iff the relation hyper-parallelogram $\boldsymbol{f_h}(r_d)$ is equal to the compositionally defined region $s_d$ defined by $\boldsymbol{f_h}(r_1)$ and $\boldsymbol{f_h}(r_2)$ for any correlation subspace.*

The proof for Proposition F.8 is straightforward, as Proposition F.8 can be proven analogously to Proposition F.7 with the sole difference that instead of defining a relation embedding $f_h(r_3)$ that subsumes the compositionally defined region $s_d$, we define the compositionally defined relation $r_d$ whose embedding $f_h(r_d)$ is equal to the compositionally defined region $s_d$.

Propositions F.1, F.2, F.3, F.5, F.4, and F.6 together prove the exactness part of Theorem 5.2, i.e., that ExpressivE can capture symmetry, anti-symmetry, inversion, intersection, hierarchy, and mutual exclusion exactly. Propositions F.7 and F.8 prove the exactness part of Theorem 5.4, i.e., that ExpressivE can capture general composition exactly. Now it remains to show that ExpressivE can capture all these patterns exactly and exclusively, which is shown in Section G.

# G  DETAILS ON CAPTURING PATTERNS EXCLUSIVELY

This section proves that ExpressivE can capture all inference patterns of Theorems 5.2 and 5.4 exactly and exclusively. By the definition of capturing a pattern $\psi$ exactly and exclusively, this means that we need to construct a relation configuration $m_h$ such that (1) $m_h$ captures $\psi$ and (2) $m_h$ does not capture any positive pattern $\phi$ such that $\psi \not\models \phi$. Note that we have shown in Propositions F.1-F.7 that we can construct a relation configuration $m_h$ that captures the following patterns by constraining the following geometric properties of $m_h$'s relation hyper-parallelograms:

1. For symmetry and inversion patterns, the mirror images across the identity line of hyper-parallelograms in any correlation subspace need to be constrained (Propositions F.1 and F.3).

2. For hierarchy and intersection patterns the intersections of hyper-parallelograms in any correlation subspace need to be constrained (Propositions F.4 and F.5).

3. For general composition patterns the compositionally defined region needs to be subsumed in any correlation subspace.

Since symmetry, inversion, hierarchy, intersection, and composition are all positive patterns of our considered language of patterns, it suffices to analyze the mirror images (M), intersections (I), and compositionally defined regions (C) of each relation hyper-parallelogram to check which positive patterns have been captured. Furthermore, for the upcoming proofs, Definition G.1 defines head and tail intervals.

**Definition G.1 (Head and Tail Intervals)** *Let $r_i \in R$ be a relation and $m_h = (M, f_h)$ be a relation configuration. We call an interval a head interval $H_{r_i,m_h}$ and respectively a tail interval $T_{r_i,m_h}$ of $r_i$ and $m_h$ if for arbitrary entities $e_h, e_t \in E$, virtual assignment functions $f_v$, and complete model configuration $m$ over $m_h$ and $f_v$ the following property holds: if $m$ captures a triple $r_1(e_h, e_t)$ to be true, then $f_v(e_h) \in H_{r_i,m_h}$ and $f_v(e_t) \in T_{r_i,m_h}$.*

Using the Definition G.1 and the insights provided by (M), (I), and (C), we will followingly prove that ExpressivE captures each considered pattern exactly and exclusively.

**Proposition G.1 (Symmetry (Exactly and Exclusively))** *Let $m_h = (M, f_h)$ be a relation configuration and $r_1 \in R$ be a symmetric relation, i.e., $r_1(X, Y) \Rightarrow r_1(Y, X)$ holds for any entities $X, Y \in E$. Then $m_h$ can capture $r_1(X, Y) \Rightarrow r_1(Y, X)$ exactly and exclusively.*

**Proposition G.2 (Anti-Symmetry (Exactly and Exclusively))** *Let $m_h = (M, f_h)$ be a relation configuration and $r_1 \in R$ be an anti-symmetric relation, i.e., $r_1(X, Y) \Rightarrow \neg r_1(Y, X)$ holds for any entities $X, Y \in E$. Then $m_h$ can capture $r_1(X, Y) \Rightarrow \neg r_1(Y, X)$ exactly and exclusively.*

The proofs for Propositions G.1 and G.2 are straightforward, as the only positive pattern that contains only one relation is symmetry. Furthermore, since (i) Propositions F.1 and F.2 have shown that there is a relation configuration that can capture symmetry/anti-symmetry exactly and (ii) a hyper-parallelogram cannot be symmetric and anti-symmetric simultaneously, we have shown that there is a relation configuration that captures symmetry/anti-symmetry exactly and exclusively, proving Propositions G.1 and G.2.

**Proposition G.3 (Inversion (Exactly and Exclusively))** *Let $m_h = (M, f_h)$ be a relation configuration and $r_1, r_2 \in R$ be relations where $r_1(X, Y) \Leftrightarrow r_2(Y, X)$ holds for any entities $X, Y \in E$. Then $m_h$ can capture $r_1(X, Y) \Leftrightarrow r_2(Y, X)$ exactly and exclusively.*

The proof for Proposition G.3 is straightforward, as the only positive patterns that contain at most two relations are symmetry, hierarchy, and inversion. Furthermore, since (i) Proposition F.3 has shown that there is a relation configuration that can capture inversion exactly and (ii) it is simple to show that a hyper-parallelogram can be the mirror image of another hyper-parallelogram without one of them subsuming the other (hierarchy) or one of them being symmetric across the identity line (symmetry), we have shown that there is a relation configuration that captures inversion exactly and exclusively, proving Proposition G.3.

**Proposition G.4 (Hierarchy (Exactly and Exclusively))** *Let $m_h = (M, f_h)$ be a relation configuration and $r_1, r_2 \in R$ be relations where $r_1(X, Y) \Rightarrow r_2(X, Y)$ holds for any entities $X, Y \in E$. Then $m_h$ can capture $r_1(X, Y) \Rightarrow r_2(X, Y)$ exactly and exclusively.*

The proof for Proposition G.4 is straightforward, as the only positive patterns that contain at most two relations are symmetry, hierarchy, and inversion. Furthermore, since (i) Proposition F.4 has shown that there is a relation configuration that can capture hierarchy exactly and (ii) it is simple to show that a hyper-parallelogram can subsume another hyper-parallelogram without one of them being the mirror image across the identity line of the other (inversion) or one of them being symmetric across the identity line (symmetry), we have shown that there is a relation configuration that captures hierarchy exactly and exclusively, proving Proposition G.4.

**Proposition G.5 (Intersection (Exactly and Exclusively))** *Let $m_h = (M, f_h)$ be a relation configuration and $r_1, r_2, r_3 \in R$ be relations where $r_1(X, Y) \wedge r_2(X, Y) \Rightarrow r_3(X, Y)$ holds for any entities $X, Y \in E$. Then $m_h$ can capture $r_1(X, Y) \wedge r_2(X, Y) \Rightarrow r_3(X, Y)$ exactly and exclusively.*

**Proof** What is to be shown is that $m_h$ can capture intersection $(r_1(X, Y) \wedge r_2(X, Y) \Rightarrow r_3(X, Y))$ exactly and exclusively. We have already shown that $m_h$ can capture $r_1(X, Y) \wedge r_2(X, Y) \Rightarrow r_3(X, Y)$ exactly in Proposition F.5. Now, to show that $m_h$ can capture intersection exactly and exclusively, we construct an instance of $m_h$ such that (1) $m_h$ captures intersection $r_1(X, Y) \wedge r_2(X, Y) \Rightarrow r_3(X, Y)$ and (2) $m_h$ does not capture any positive pattern $\phi$ such that $r_1(X, Y) \wedge r_2(X, Y) \Rightarrow r_3(X, Y) \not\models \phi$.

Table 6: One-dimensional relation embeddings of a relation configuration $m_h$ that captures intersection (i.e., $r_1(X, Y) \wedge r_2(X, Y) \Rightarrow r_3(X, Y)$) exactly and exclusively.

|       | $c^h$  | $d^h$ | $r^t$ | $c^t$ | $d^t$ | $r^h$ |
|-------|--------|-------|-------|-------|-------|-------|
| $r_1$ | $-6$   | 2     | 2     | 8     | 2     | 3     |
| $r_2$ | $-11.5$| 3     | 5     | 11    | 3     | 3     |
| $r_3$ | $-9.5$ | 5     | 5     | 9     | 1     | 3     |

Figure 2 visualizes the hyper-parallelograms defined by the one-dimensional relation embeddings of Table 6. In particular, it displays the hyper-parallelograms of $r_1, r_2, r_3$. As can be easily seen in Figure 2 (and proven using Proposition F.5), the relation configuration $m_h$ described by Table 6 captures $r_1(X, Y) \wedge r_2(X, Y) \Rightarrow r_3(X, Y)$ exactly, as $f_h(r_3)$ subsumes the intersection of $f_h(r_1)$ and $f_h(r_2)$.

Now it remains to show that $m_h$ does not capture any positive pattern $\phi$ such that $r_1(X, Y) \wedge r_2(X, Y) \Rightarrow r_3(X, Y) \not\models \phi$. To show this, we will show that (M) the mirror image of any relation hyper-parallelogram is not subsumed by any other relation hyper-parallelogram (i.e., no unwanted symmetry nor inversion pattern is captured) and (C) the compositionally defined region defined by any pair of hyper-parallelograms is not subsumed by any relation hyper-parallelogram (i.e., no unwanted composition pattern is captured). We do not need to show that (I) no unwanted relation hyper-parallelograms intersect, as by the nature of the intersection pattern, $f_h(r_1)$, $f_h(r_2)$, and $f_h(r_3)$ should intersect.

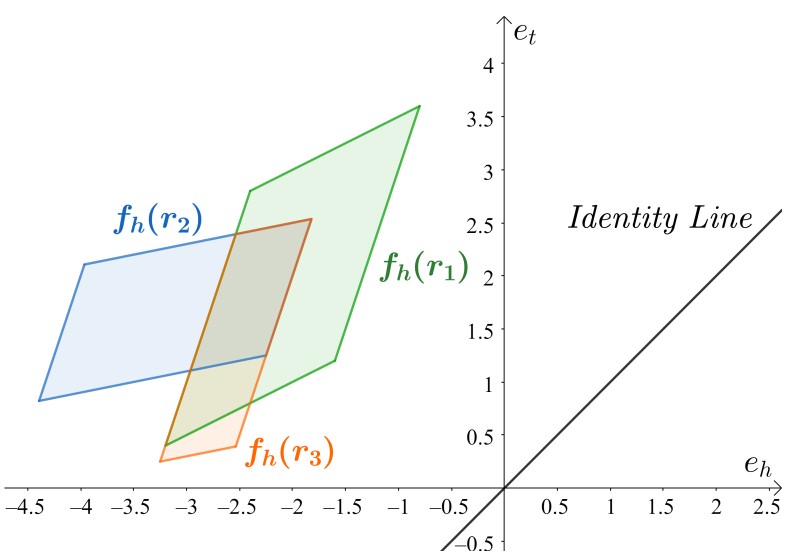

Figure 2: Visualization of the relation configuration $m_h$ described by Table 6.

For (M), observe in Figure 2 that all hyper-parallelograms $f_h(r_1)$, $f_h(r_2)$, and $f_h(r_3)$ of $m_h$ are on the same side of the identity line. Thus, the mirror images of $f_h(r_1)$, $f_h(r_2)$, and $f_h(r_3)$ across the identity line must be on the other side. Therefore, we have shown (M), i.e., that no relation hyper-parallelograms subsume the mirror image of any other relation hyper-parallelogram and thus that $m_h$ does not capture any unwanted symmetry nor inversion pattern.

For (C), observe in Figure 2 that for the displayed relation configuration $m_h$, the head intervals of any relation hyper-parallelogram of $m_h$ contain only negative values and the tail intervals contain only positive values. Thus, for any pair $(r_i, r_j) \in \{r_1, r_2, r_3\}^2$, there is no virtual assignment function $f_v$ such that $m$ over $m_h$ and $f_v$ captures $r_i(x, y)$ and $r_j(y, z)$ for arbitrary entities $x, y, z \in E$. Therefore, no pair of relations $(r_i, r_j)$ defines a compositionally defined region. Thus, we have shown (C) that no compositionally defined region is subsumed by any relation hyper-parallelogram (as no compositionally defined region exists) and thus that $m_h$ does not capture any unwanted general composition pattern.

By Proposition F.5 and by proving (M) and (C), we have shown that the constructed relation configuration $m_h$ of Table 6 captures the intersection pattern $r_1(X, Y) \wedge r_2(X, Y) \Rightarrow r_3(X, Y)$ and does not capture any positive pattern $\phi$ such that $r_1(X, Y) \wedge r_2(X, Y) \Rightarrow r_3(X, Y) \not\models \phi$. This means by the definition of capturing patterns exactly and exclusively that $m_h$ captures intersection $(r_1(X, Y) \wedge r_2(X, Y) \Rightarrow r_3(X, Y))$ exactly and exclusively, proving the proposition. □

**Proposition G.6 (General Composition (Exactly and Exclusively))** *Let $r_1, r_2, r_3 \in R$ be relations and let $m_h = (M, f_h)$ be a relation configuration, where $f_h$ is defined over $r_1, r_2$, and $r_3$. Furthermore let $r_3$ be the composite relation of $r_1$ and $r_2$, i.e., $r_1(X, Y) \wedge r_2(Y, Z) \Rightarrow r_3(X, Z)$ holds for all entities $X, Y, Z \in E$. Then $m_h$ can capture $r_1(X, Y) \wedge r_2(Y, Z) \Rightarrow r_3(X, Z)$ exactly and exclusively.*

**Proof**    What is to be shown is that $m_h$ can capture general composition $(r_1(X, Y) \wedge r_2(Y, Z) \Rightarrow r_3(X, Z))$ exactly and exclusively. We have already shown that $m_h$ can capture $r_1(X, Y) \wedge r_2(Y, Z) \Rightarrow r_3(X, Z)$ exactly in Proposition F.7. Now, to show that $m_h$ can capture general composition exactly and exclusively, we construct an instance of $m_h$ such that (1) $m_h$ captures general composition and (2) $m_h$ does not capture any positive pattern $\phi$ such that $r_1(X, Y) \wedge r_2(Y, Z) \Rightarrow r_3(X, Z) \not\models \phi$.

Figure 3 visualizes the hyper-parallelograms defined by the one-dimensional relation embeddings of Table 7. In particular, it displays the hyper-parallelograms of $r_1, r_2, r_3$, and the compositionally defined region $s_d$ of auxiliary relation $r_d$ such that $r_1(X, Y) \wedge r_2(Y, Z) \Leftrightarrow r_d(X, Z)$ holds for

Table 7: One-dimensional relation embeddings of a relation configuration $m_h$ that captures general composition (i.e., $r_1(X, Y) \wedge r_2(Y, Z) \Rightarrow r_3(X, Z)$) and that captures compositional definition (i.e., $r_1(X, Y) \wedge r_2(Y, Z) \Leftrightarrow r_d(X, Z)$) exactly and exclusively.

|       | $c^h$ | $d^h$ | $r^t$ | $c^t$ | $d^t$ | $r^h$ |
|-------|-------|-------|-------|-------|-------|-------|
| $r_1$ | $-6$  | 0     | 2     | 8     | 5     | 3     |
| $r_2$ | $-35$ | 5     | 5     | $-1$  | 2     | 5     |
| $r_d$ | $-76$ | 10    | 10    | 14    | 2     | 2.5   |
| $r_3$ | $-46$ | 11    | 6     | 19    | 6     | 4     |

$f_h(r_1)$, $f_h(r_2)$, and $s_d$. As can be easily seen in Figure 3 (and proven using Theorem E.2 and Proposition F.7), the relation configuration $m_h$ described by Table 7 captures $r_1(X, Y) \wedge r_2(Y, Z) \Rightarrow r_3(X, Z)$ exactly, as $f_h(r_3)$ subsumes the compositionally defined region $s_d$.

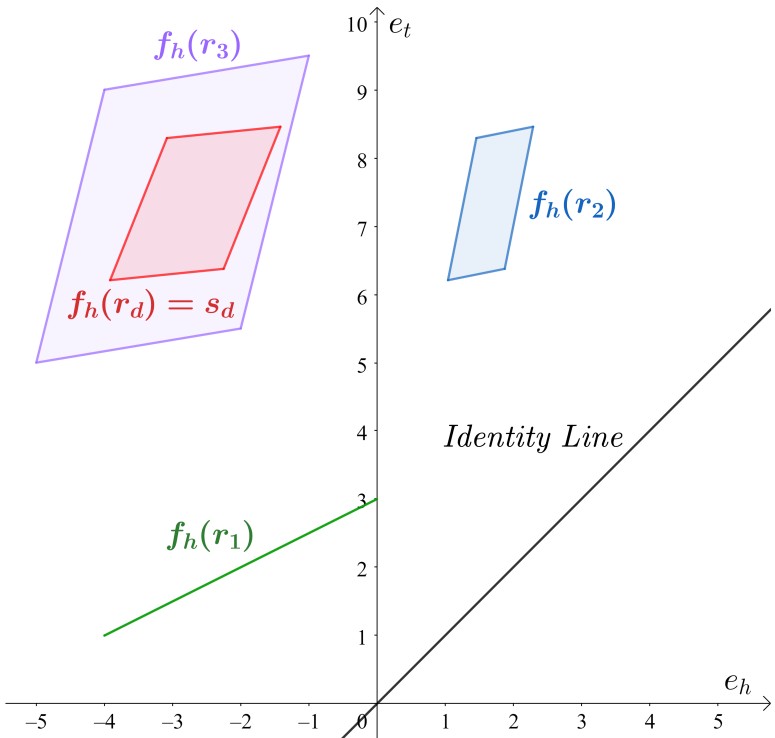

Figure 3: Visualization of the relation configuration $m_h$ described by Table 7.

Now it remains to show that $m_h$ does not capture any positive pattern $\phi$ such that $r_1(X, Y) \wedge r_2(Y, Z) \Rightarrow r_3(X, Z) \not\models \phi$. To show this, we will show that (M) the mirror image of any relation hyper-parallelogram is not subsumed by any other relation hyper-parallelogram (i.e., no unwanted symmetry nor inversion pattern is captured), (I) no relation hyper-parallelograms intersect with each other (i.e., no unwanted hierarchy nor intersection pattern is captured), and (C) solely the compositionally defined region $s_d$ defined by $f_h(r_1)$ and $f_h(r_2)$ is subsumed by $f_h(r_3)$ and no other compositionally defined region is subsumed by any other relation hyper-parallelogram (i.e., no unwanted composition pattern is captured).

For (M), observe in Figure 3 that all hyper-parallelograms $f_h(r_1)$, $f_h(r_2)$, and $f_h(r_3)$ of $m_h$ are on the same side of the identity line. Thus, the mirror images of $f_h(r_1)$, $f_h(r_2)$, and $f_h(r_3)$ across the identity line must be on the other side. Therefore, we have shown (M), i.e., that no relation hyper-parallelograms subsume the mirror image of any other relation hyper-parallelogram and thus that $m_h$ does not capture any unwanted symmetry nor inversion pattern.

For (I), observe in Figure 3 that no relation hyper-parallelograms $f_h(r_1)$, $f_h(r_2)$, and $f_h(r_3)$ of $m_h$ intersect with each other. Thus, we have shown (I), i.e., that $m_h$ does not capture any unwanted hierarchy nor intersection pattern.

For (C), observe in Figure 3 that for the displayed relation configuration $m_h$, the following head and tail intervals can be defined: (i) $H_{r_1,m_h} = [-4, 0]$ and $T_{r_1,m_h} = [1, 3]$, (ii) $H_{r_2,m_h} = [1, 3]$ and $T_{r_2,m_h} = [6, 9]$, and (iii) $H_{r_3,m_h} = [-6, -1]$ and $T_{r_3,m_h} = [4, 10]$. The tail intervals solely overlap with the head intervals for $T_{r_1,m_h}$ and $H_{r_2,m_h}$, i.e., $T_{r_i,m_h} \cap H_{r_j,m_h} = \emptyset, (r_i, r_j) \in \{r_1, r_2, r_3\}^2 \setminus (r_1, r_2)$. Thus, for any pair $(r_i, r_j) \in \{r_1, r_2, r_3\}^2 \setminus (r_1, r_2)$ there is no virtual assignment function $f_v$ such that $m$ over $m_h$ and $f_v$ captures $r_i(x, y)$ and $r_j(y, z)$ for arbitrary entities $x, y, z \in E$. Therefore, $(r_1, r_2)$ is the only pair of relations that defines a compositionally defined region, i.e., no other pair of relations defines a compositionally defined region. Thus, we have shown (C) that no other compositionally defined region is subsumed by any other relation (as no other compositionally defined region exists) and thus that no unwanted composition pattern is captured by $m_h$.

By Proposition F.7 and by proving (I), (M), and (C), we have shown that the constructed relation configuration $m_h$ of Table 7 captures the general composition pattern $r_1(X, Y) \wedge r_2(Y, Z) \Rightarrow r_3(X, Z)$ and does not capture any positive pattern $\phi$ such that $r_1(X, Y) \wedge r_2(Y, Z) \Rightarrow r_3(X, Z) \not\models \phi$. This means by the definition of capturing patterns exactly and exclusively that $m_h$ captures general composition ($r_1(X, Y) \wedge r_2(Y, Z) \Rightarrow r_3(X, Z)$) exactly and exclusively, proving the proposition. $\square$

**Proposition G.7 (Compositional Definition (Exactly and Exclusively))** *Let $r_1, r_2, r_d \in R$ be relations and let $m_h = (M, f_h)$ be a relation configuration, where $f_h$ is defined over $r_1, r_2$, and $r_d$. Furthermore, let $r_d$ be the compositionally defined relation of $r_1$ and $r_2$, i.e., $r_1(X, Y) \wedge r_2(Y, Z) \Leftrightarrow r_d(X, Z)$ holds for all entities $X, Y, Z \in E$. Then $m_h$ can capture $r_1(X, Y) \wedge r_2(Y, Z) \Leftrightarrow r_d(X, Z)$ exactly and exclusively.*

The proof for Proposition G.7 is straightforward, as it can be proven analogously to Proposition G.6 with the only difference that instead of defining a relation embedding $f_h(r_3)$ that subsumes the compositionally defined region, we define the compositionally defined relation $r_d$ whose embedding $f_h(r_d)$ is equal to the compositionally defined region $s_d$. We have stated the relation embeddings for $r_d$ in Table 7 and also visualized $f_h(r_d)$ in Figure 3.

Finally, the sum of Propositions G.1-G.7 proves Theorems 5.2 and 5.4. Thus, we have theoretically shown that ExpressivE can capture any pattern from Table 1 exactly and exclusively.

## H  EXTENDED COMPOSITIONS

This section provides theoretical evidence that ExpressivE is not limited to capturing a single composition pattern. Specifically, we prove that ExpressivE can capture more than one application of a composition pattern. The following theoretical result is empirically backed up by further experimental results of Appendix I.3.

**Proposition H.1** *Let $r_1, r_2, r_3, r_{1,2}, r_{1,2,3} \in R$ be relations and let $m_h = (M, f_h)$ be a relation configuration, where $f_h$ is defined over $r_1, r_2, r_3, r_{1,2}$, and $r_{1,2,3}$. Furthermore, let $r_1(X, Y) \wedge r_2(Y, Z) \Rightarrow r_{1,2}(X, Z)$ and $r_{1,2}(X, Y) \wedge r_3(Y, Z) \Rightarrow r_{1,2,3}(X, Z)$ hold for all entities $X, Y, Z \in E$. Then $m_h$ can capture $r_1(X, Y) \wedge r_2(Y, Z) \Rightarrow r_{1,2}(X, Z)$ and $r_{1,2}(X, Y) \wedge r_3(Y, Z) \Rightarrow r_{1,2,3}(X, Z)$ exactly and exclusively.*

**Proof**  What is to be shown is that $m_h$ can capture $\phi_1 := r_1(X, Y) \wedge r_2(Y, Z) \Rightarrow r_{1,2}(X, Z)$ and $\phi_2 := r_{1,2}(X, Y) \wedge r_3(Y, Z) \Rightarrow r_{1,2,3}(X, Z)$ exactly and exclusively. To show that there is an $m_h$ that captures $\phi_1$ and $\phi_2$ exactly and exclusively, we construct an instance of $m_h$ such that (1) $m_h$ captures $\phi_1$ and $\phi_2$ exactly, and (2) $m_h$ does not capture any positive pattern $\psi$ such that $(\phi_1 \wedge \phi_2) \not\models \psi$.

Figure 4 visualizes the hyper-parallelograms defined by the one-dimensional relation embeddings of Table 8. In particular, it displays the hyper-parallelograms of $r_1$, $r_2$, $r_{1,2}$, $r_3$, $r_{1,2,3}$, and the compositionally defined regions $s_{1,2}^d, s_{2,3}^d, s_{(1,2),3}^d, s_{1,(2,3)}^d$ of auxiliary relation $r_{1,2}^d, r_{2,3}^d, r_{(1,2),3}^d$, and $r_{1,(2,3)}^d$ such that $r_1(X, Y) \wedge r_2(Y, Z) \Leftrightarrow r_{1,2}^d(X, Z)$, $r_2(X, Y) \wedge r_3(Y, Z) \Leftrightarrow r_{2,3}^d(X, Z)$,

Table 8: One-dimensional relation embeddings of a relation configuration $m_h$ that captures two general compositions (i.e., $r_1(X,Y) \wedge r_2(Y,Z) \Rightarrow r_{1,2}(X,Z)$ and $r_{1,2}(X,Y) \wedge r_3(Y,Z) \Rightarrow r_{1,2,3}(X,Z)$) exactly and exclusively.

|           | $c^h$  | $d^h$ | $r^t$ | $c^t$ | $d^t$ | $r^h$ |
|-----------|--------|-------|-------|-------|-------|-------|
| $r_1$     | $-6$   | 0     | 2     | 8     | 5     | 3     |
| $r_2$     | $-35$  | 5     | 5     | $-1$  | 2     | 5     |
| $r_{1,2}$ | $-46$  | 11    | 6     | 19    | 6     | 4     |
| $r_3$     | $-45$  | 3     | 5     | $-20$ | 0     | 4     |
| $r_{1,2,3}$ | $-215$ | 20   | 20    | 22    | 8     | 4     |

$r_{1,2}(X,Y) \wedge r_3(Y,Z) \Leftrightarrow r^d_{(1,2),3}(X,Z)$, and $r_1(X,Y) \wedge r^d_{2,3}(Y,Z) \Leftrightarrow r^d_{1,(2,3)}(X,Z)$ hold for $f_h(r_1)$, $f_h(r_2)$, $f_h(r_3)$, $f_h(r_{1,2})$, $f_h(r_{1,2,3})$, $s^d_{1,2}$, $s^d_{2,3}$, $s^d_{(1,2),3}$, and $s^d_{1,(2,3)}$. Note that from $\phi_1$ and $\phi_2$ together with the auxiliary relation $r^d_{1,(2,3)}$ — defined above — follows that $r^d_{1,2}(X,Y) \Rightarrow r_{1,2}(X,Y)$, $r^d_{(1,2),3}(X,Y) \Rightarrow r_{1,2,3}(X,Y)$, and $r^d_{1,(2,3)}(X,Y) \Rightarrow r_{1,2,3}(X,Y)$ need to be satisfied. Thus, as can be easily seen in Figure 4 (and proven using Theorem E.2 and Proposition F.7), the relation configuration $m_h$ described by Table 8 captures $\phi_1$ and $\phi_2$ exactly, as $f_h(r_{1,2})$ subsumes the compositionally defined region $s^d_{1,2}$ and as $f_h(r_{1,2,3})$ subsumes the compositionally defined regions $s^d_{(1,2),3}$ and $s^d_{1,(2,3)}$.

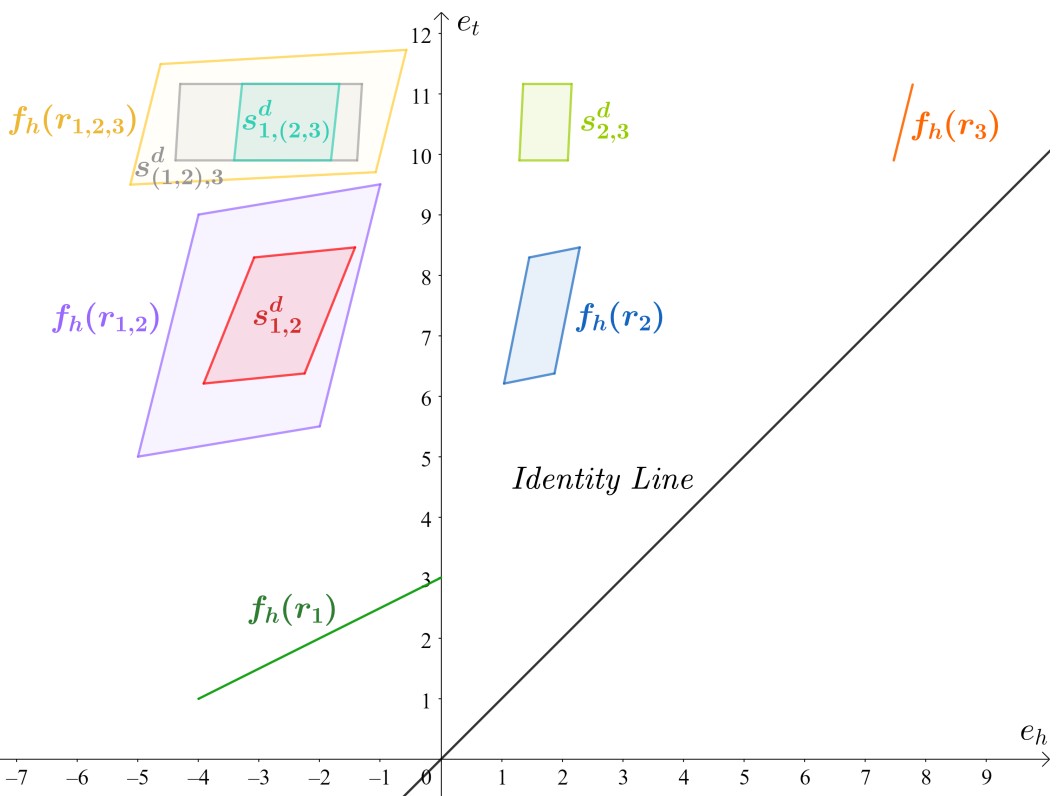

Figure 4: Visualization of the relation configuration $m_h$ described by Table 8.

Now it remains to show that $m_h$ does not capture any positive pattern $\psi$ such that $(\phi_1 \wedge \phi_2) \not\models \psi$. To show this, we will show that (M) the mirror image of any relation hyper-parallelogram is not subsumed by any other relation hyper-parallelogram (i.e., no unwanted symmetry nor inversion pattern

is captured), (I) no relation hyper-parallelograms intersect with each other (i.e., no unwanted hierarchy nor intersection pattern is captured), and (C) solely that $s_{1,2}^d \subseteq f_h(r_{1,2})$ and $(s_{(1,2),3}^d \cup s_{1,(2,3)}^d) \subseteq f_h(r_{1,2,3})$ are satisfied, and no other compositionally defined region is subsumed by any other relation hyper-parallelogram (i.e., no unwanted composition pattern is captured).

For (M), observe in Figure 4 that all hyper-parallelograms $f_h(r_1)$, $f_h(r_2)$, $f_h(r_3)$, $f_h(r_{1,2})$, and $f_h(r_{1,2,3})$ of $m_h$ are on the same side of the identity line. Thus, the mirror images of any of these hyper-parallelograms across the identity line must be on the other side. Therefore, we have shown (M), i.e., that no relation hyper-parallelograms subsume the mirror image of any other relation hyper-parallelogram and thus that $m_h$ does not capture any unwanted symmetry nor inversion pattern.

For (I), observe in Figure 4 that no relation hyper-parallelograms $f_h(r_1)$, $f_h(r_2)$, $f_h(r_3)$, $f_h(r_{1,2})$, and $f_h(r_{1,2,3})$ of $m_h$ intersect with each other. Thus, we have shown (I), i.e., that $m_h$ does not capture any unwanted hierarchy nor intersection pattern.

For (C), recall Definition G.1, describing head and tail intervals. We observe in Figure 4 that for the displayed relation configuration $m_h$, the following head and tail intervals can be defined: (i) $H_{r_1,m_h} = [-4, 0]$ and $T_{r_1,m_h} = [1, 3]$, (ii) $H_{r_2,m_h} = [1, 3]$ and $T_{r_2,m_h} = [6, 9]$, (iii) $H_{r_{1,2},m_h} = [-6, -1]$ and $T_{r_{1,2},m_h} = [4, 9.7]$, (iv) $H_{r_3,m_h} = [7, 9]$ and $T_{r_3,m_h} = [10, 12]$, (v) $H_{r_{1,2,3},m_h} = [-6, 0]$ and $T_{r_{1,2,3},m_h} = [9.8, 12]$, and (vi) $H_{r_{2,3}^d,m_h} = [1, 3]$ and $T_{r_{2,3}^d,m_h} = [9.8, 12]$. The tail intervals solely overlap with the head intervals for the pairs $\{(r_1, r_2), (r_2, r_3), (r_{1,2}, r_3), (r_1, r_{2,3}^d)\}$, i.e., $T_{r_i,m_h} \cap H_{r_j,m_h} = \emptyset, (r_i, r_j) \in \{r_1, r_2, r_3\}^2 \setminus \{(r_1, r_2), (r_2, r_3), (r_{1,2}, r_3), (r_1, r_{2,3}^d)\}$. Thus, for any pair $(r_i, r_j) \in \{r_1, r_2, r_3\}^2 \setminus \{(r_1, r_2), (r_2, r_3), (r_{1,2}, r_3), (r_1, r_{2,3}^d)\}$ there is no virtual assignment function $f_v$ such that $m$ over $m_h$ and $f_v$ captures $r_i(x, y)$ and $r_j(y, z)$ for arbitrary entities $x, y, z \in E$. Therefore, $\{(r_1, r_2), (r_2, r_3), (r_{1,2}, r_3), (r_1, r_{2,3}^d)\}$ are the only pairs of relations that define a compositionally defined region, i.e., no other pair of relations defines a compositionally defined region. Thus we have shown that (1) $m_h$ captures $\phi_1$ and $\phi_2$ exactly — since $s_{1,2}^d \subseteq f_h(r_{1,2})$ and $(s_{(1,2),3}^d \cup s_{1,(2,3)}^d) \subseteq f_h(r_{1,2,3})$ — and (2) the only other existing compositionally defined region $s_{2,3}^d$ is disjoint with any other relation hyper-parallelograms. By (1) and (2), we have shown (C) that no other compositionally defined region (specifically $s_{1,2}^d$) is subsumed by any other relation and thus that no unwanted composition pattern is captured by $m_h$.

By proving that the constructed $m_h$ captures $\phi_1$ and $\phi_2$ exactly and by (I), (M), and (C), we have shown that the constructed relation configuration $m_h$ of Table 8 captures $\phi_1$ and $\phi_2$ and does not capture any positive pattern $\psi$ such that $(\phi_1 \wedge \phi_2) \not\models \psi$. This means by the definition of capturing patterns exactly and exclusively that $m_h$ captures $\phi_1$ and $\phi_2$ exactly and exclusively, proving the proposition. $\qquad \square$

# I ADDITIONAL EXPERIMENTS

This section presents additional experiments, providing further empirical evidence for our theoretical results. Specifically, Section I.1 studies the benchmark performances of ExpressivE and its closest relatives on WN18RR stratified by the cardinality of each relation, providing empirical evidence that ExpressivE performs well on 1-1, 1-N, N-1, and N-N relations. Section I.2 provides empirical evidence that ExpressivE can capture general composition and provides empirical support for a link between ExpressivE's significant performance gain on WN18RR and inference capabilities. Finally, Section I.3 discusses empirical results, revealing that ExpressivE can reason over more than one step of composition patterns.

## I.1 CARDINALITY EXPERIMENTS

This section provides empirical evidence for our theoretical result that ExpressivE performs well on 1-N, N-1, and N-N relations.

**Experiment Setup.** Following the procedure of Bordes et al. (2013), we have categorized the relations of WN18RR into four cardinality classes, specifically 1-1, 1-N, N-1, and N-N. As in Bordes et al. (2013), we have classified a relation $r \in R$ by computing:

- $\mu_{rt}$ the averaged number of head entities $h \in \boldsymbol{E}$ per tail entity $t \in \boldsymbol{E}$, appearing in a triple $r(h,t)$ of WN18RR.
- $\mu_{rh}$ the averaged number of tail entities $t \in \boldsymbol{E}$ per head entity $h \in \boldsymbol{E}$, appearing in a triple $r(h,t)$ of WN18RR.

Following the soft classification of Bordes et al. (2013), a relation is:

- **1-1** if $\mu_{rt} \leq 1.5$ and $\mu_{rh} \leq 1.5$
- **1-N** if $\mu_{rt} \leq 1.5$ and $\mu_{rh} \geq 1.5$
- **N-1** if $\mu_{rt} \geq 1.5$ and $\mu_{rh} \leq 1.5$
- **N-N** if $\mu_{rt} \geq 1.5$ and $\mu_{rh} \geq 1.5$

Table 9: MRR of ExpressivE, RotatE, and BoxE on WN18RR stratified by cardinality classes (1-1, 1-N, N-1, N-N). The best results are bold, and the second-best are underlined.

| Task | **Predicting Head** | | | | **Predicting Tail** | | | |
|------|-----|-----|-----|-----|-----|-----|-----|-----|
| Cardinality | 1-1 | 1-N | N-1 | N-N | 1-1 | 1-N | N-1 | N-N |
| ExpressivE | **0.976** | 0.290 | 0.105 | **0.941** | **0.976** | 0.141 | **0.327** | **0.938** |
| RotatE | 0.833 | **0.294** | 0.103 | 0.930 | 0.875 | 0.107 | 0.288 | 0.925 |
| BoxE | 0.877 | 0.272 | **0.146** | 0.883 | 0.893 | **0.147** | 0.246 | 0.884 |

**Results.** Table 9 summarizes the performance results of ExpressivE and its closest spatial relative BoxE and functional relative RotatE on WN18RR, stratified by the four cardinality classes defined previously. It reveals that ExpressivE almost exclusively reaches a SotA or close-to-SotA performance on 1-N, N-1, and N-N relations. In particular, ExpressivE outperforms both RotatE and BoxE consistently on N-N relations, which are often considered the most complex relations to capture in KGC with regard to cardinalities. Thus, Table 9 provides empirical results supporting our theoretical claim that ExpressivE can capture 1-1, 1-N, N-1, and N-N relations well.

## I.2 GENERAL COMPOSITION AND LINK TO PERFORMANCE GAIN

This section provides empirical evidence for the theoretical result of Appendices F and G that ExpressivE can capture general composition exactly and exclusively. Even more, the experiments of this section give evidence for a direct link between the support of general composition and ExpressivE's performance gain on WN18RR. In the following, we first discuss our experiments' preparation and setup details, followed by the considered hypotheses and final results.

**Pattern Identification.** Our first goal, to provide empirical evidence for the discussed points, was to identify patterns occurring in WN18RR. To reach this goal, we have analyzed patterns mined with AMIE+ (Galárraga et al., 2015) from WN18RR by Akrami et al. (2020) that were provided in a GitHub repository[2]. To identify the most relevant patterns, we have — similar to the discussion of (Galárraga et al., 2013; 2015) — sorted the patterns $\rho = \phi_{B1} \wedge \cdots \wedge \phi_{Bm} \Rightarrow r(X,Y)$ by their head coverage $h(\rho)$, which is formally defined as (Galárraga et al., 2013):

$$h(\rho) = \frac{|\{(x,y) \in \boldsymbol{E}^2 \mid r(x,y) \in \boldsymbol{G} \wedge \exists z_1 \ldots z_k(\phi_{B1}(z_1,z_2) \in \boldsymbol{G} \wedge \cdots \wedge \phi_{Bm}(z_{k-1},z_k) \in \boldsymbol{G})\}|}{|\{(x,y) \in \boldsymbol{E}^2 \mid r(x,y) \in \boldsymbol{G}\}|}$$

On an intuitive level, the head coverage $h(\rho)$ represents the ratio of true triples implied by the pattern $\rho$ on a given knowledge graph $(\boldsymbol{G}, \boldsymbol{E}, \boldsymbol{R})$.

**Pattern Selection.** To analyze the most relevant patterns in the following experiments, we have selected any patterns whose head coverage is greater than 15% (as inspection of the head coverage of AMIE shows a very low number of inferred triples contained in the test set below that). From these patterns, we have left out any pattern with the head relation $\_similar\_to$, as ExpressivE, BoxE, and RotatE already have an MRR of 1 on this relation, thus further stratifying $\_similar\_to$'s test

---

[2]https://github.com/idirlab/kgcompletion

triples will not reveal novel information. This procedure leads to the following set of patterns, where relations $r^{-1}$ represent the inverse counterpart of relations $r \in \boldsymbol{R}$:

$$
\begin{aligned}
S_1 :=& \_verb\_group(Y, X) \Rightarrow \_verb\_group(X, Y) \\
C_2 :=& \_derivationally\_related\_form(X, Y) \wedge \\
& \_derivationally\_related\_form(Y, Z) \Rightarrow \_verb\_group(X, Z) \\
C_3 :=& \_derivationally\_related\_form(X, Y) \wedge \\
& \_derivationally\_related\_form^{-1}(Y, Z) \Rightarrow \_verb\_group(X, Z) \\
C_4 :=& \_derivationally\_related\_form^{-1}(X, Y) \wedge \\
& \_derivationally\_related\_form(Y, Z) \Rightarrow \_verb\_group(X, Z) \\
C_5 :=& \_also\_see(X, Y) \wedge \_also\_see(Y, Z) \Rightarrow \_also\_see(X, Z) \\
C_6 :=& \_also\_see(X, Y) \wedge \_also\_see^{-1}(Y, Z) \Rightarrow \_also\_see(X, Z) \\
S_7 :=& \_also\_see(Y, X) \Rightarrow \_also\_see(X, Y) \\
C_8 :=& \_hypernym(X, Y) \wedge \\
& \_synset\_domain\_topic\_of(Y, Z) \Rightarrow \_synset\_domain\_topic\_of(X, Z)
\end{aligned}
$$

**Experimental Setup.** For each of these patterns $\rho$ we have computed all triples that (i) can be derived by $\rho$ from the data known to our model and (ii) are known to be true in the KG, yet unseen to our models. Thus, for each pattern $\rho$, we have computed the set $s_\rho$, containing all triples that (i) can be derived with $\rho$ from the training set and (ii) are contained in the test set of WN18RR. We have used each of the computed sets of triples $s_\rho$ to evaluate the performance of ExpressivE, BoxE, and RotatE on the corresponding pattern $\rho$.

**Hypotheses.** Note that (as discussed in Appendix K.1) compositional definition $r_1(X, Y) \wedge r_2(Y, Z) \Leftrightarrow r_3(X, Z)$ defines the triples of the composite relation $r_3$ completely, whereas general composition $r_1(X, Y) \wedge r_2(Y, Z) \Rightarrow r_3(X, Z)$ allows $r_3$ to contain more triples than those that the compositional definition pattern can directly infer. Thus, if ExpressivE captures general composition and if RotatE captures compositional definition, we expect the following behavior:

- **H1.** RotatE will perform well solely on relations occurring as the head of maximally one composition pattern, as RotatE solely supports compositional definition.

- **H2.** ExpressivE will perform well even when a relation is defined by multiple composition patterns and/or multiple other patterns since ExpressivE supports general composition.

Table 10: MRR of ExpressivE, RotatE, and BoxE on WN18RR stratified by patterns $S_1$-$C_8$. $S_i$ represents a [S]ymmetry pattern, $C_i$ a [C]omposition pattern ($i \in \{1, \dots, 8\}$).

| Head Rel. | _verb_group | | | | _also_see | | | _syn_dto |
|---|---|---|---|---|---|---|---|---|
| Model | $S_1$ | $C_2$ | $C_3$ | $C_4$ | $C_5$ | $C_6$ | $S_7$ | $C_8$ |
| Base Exp. | **1.000** | **1.000** | **1.000** | **1.000** | **0.818** | **0.907** | **0.985** | **0.621** |
| RotatE | 0.865 | 0.760 | 0.760 | 0.760 | 0.771 | 0.893 | 0.975 | 0.599 |
| BoxE | 0.906 | 0.801 | 0.806 | 0.806 | 0.632 | 0.645 | 0.727 | 0.547 |

**Results.** Table 10 lists for each pattern $S_1$ to $C_8$ the performances of BoxE, RotatE, and ExpressivE on $s_\rho$, where $\rho \in \{S_1, \dots, C_8\}$ and where $S_i$ represents a symmetry pattern and $C_i$ represents a composition pattern. Table 10 provides evidence for both hypotheses:

- **Evidence for H1.** In the case of the relation $\_synset\_domain\_topic\_of$ ($\_syn\_dto$), there is only one pattern that has $\_synset\_domain\_topic\_of$ as its head relation, specifically the composition pattern $C_8$. RotatE achieves comparable performance to ExpressivE on $s_{C_8}$ as RotatE is capable of defining $\_synset\_domain\_topic\_of$ using compositional definition, providing evidence for H1.

- **Evidence for H2.** Yet, when a relation is defined via multiple patterns, RotatE's performance decreases drastically on most composition patterns compared to ExpressivE's performance, as can be seen for the patterns $C_2$, $C_3$, $C_4$, and $C_5$, giving evidence for H2.

**Conclusion** Thus, these experiments provide empirical evidence for (1) ExpressivE can capture general composition, as ExpressivE and RotatE perform as expected by H1 and H2 under the assumption that ExpressivE captures general composition and that RotatE captures compositional definition. Furthermore, the experiments also provide evidence for (2) ExpressivE's ability to capture general composition contributes to the performance gain on WN18RR, as ExpressivE consistently outperforms RotatE and BoxE on the predicted triples of composition patterns.

### I.3   MULTIPLE STEPS OF COMPOSITION

In this section, we provide empirical evidence for the theoretical results of Appendix H. To evaluate how well ExpressivE supports more than one step of a composition pattern, our first goal was to identify multi-step patterns (i.e., patterns that can be "chained" in multiple steps) occurring in WN18RR. We now recall parts of Appendix I.2 for the self-containedness of this section – readers who have read that section can skip ahead to the "experimental setup" paragraph. To reach the goal of identifying multi-step patterns occurring in WN18RR, we have analyzed patterns mined with AMIE+ (Galárraga et al., 2015) from WN18RR by Akrami et al. (2020) that were provided in a GitHub repository[3]. To identify the most relevant patterns, we have — similar to the discussion of (Galárraga et al., 2013; 2015) — sorted the patterns $\rho = \phi_{B1} \wedge \cdots \wedge \phi_{Bm} \Rightarrow r(X, Y)$ by their head coverage $h(\rho)$, which is formally defined as (Galárraga et al., 2013):

$$h(\rho) = \frac{|\{(x,y) \in \boldsymbol{E}^2 \mid r(x,y) \in \boldsymbol{G} \wedge \exists z_1 \ldots z_k (\phi_{B1}(z_1, z_2) \in \boldsymbol{G} \wedge \cdots \wedge \phi_{Bm}(z_{k-1}, z_k) \in \boldsymbol{G})\}|}{|\{(x,y) \in \boldsymbol{E}^2 \mid r(x,y) \in \boldsymbol{G}\}|}$$

On an intuitive level, the head coverage $h(\rho)$ represents the ratio of true triples implied by the pattern $\rho$ on a given knowledge graph $(\boldsymbol{G}, \boldsymbol{E}, \boldsymbol{R})$.

Next, we present the four multi-step patterns with head coverage of at least 15%, as discussed in Appendix I.2:

$$R_1 := \_hypernym(X, Y) \wedge$$
$$\_synset\_domain\_topic\_of(Y, Z) \Rightarrow \_synset\_domain\_topic\_of(X, Z)$$
$$R_2 := \_also\_see(X, Y) \wedge \_also\_see(Y, Z) \Rightarrow \_also\_see(X, Z)$$
$$R_3 := \_also\_see(X, Y) \wedge \_also\_see^{-1}(Y, Z) \Rightarrow \_also\_see(X, Z)$$
$$R_4 := \_also\_see^{-1}(X, Y) \wedge \_also\_see^{-1}(Y, Z) \Rightarrow \_also\_see(X, Z)$$

The relation $\_also\_see^{-1}$ of $R_3$ and $R_4$ represents the inverse relation of $\_also\_see$.

**Experimental Setup.** For each of the selected multi-step patterns $\rho \in \{R_1, R_2, R_3, R_4\}$, we have generated three datasets, the *1-Step*, *2-Steps*, and *3-Steps* sets. Specifically, we have generated for each $\rho$ a *j-Step(s)* set by computing all triples that (i) can be derived by $\rho$ in $j$ steps from the data known to our model and (ii) are known to be true in the KG, yet unseen to our model. Thus, we have computed for each $\rho$ a *j-Step(s)* set, containing all triples that (i) can be derived with $\rho$ by $j$ applications on the training set and (ii) are contained in the test set of WN18RR. The performance of ExpressivE on the computed datasets is summarised in Table 11.

**Results.** We report the performance of at most two steps of $R_1/R_3/R_4$ as after applying $R_1/R_3/R_4$ twice on the training set; no new triples are derived. Similarly, no new triples are derived after at most three steps of $R_2$ on the training set. We can see that the performance of ExpressivE increases by a large margin when more than one step of reasoning is considered, depicted by the performance gain of the 2-Steps and 3-Steps set over the 1-Step set. Interestingly, a small exception for this is $R_1$, where we see a slightly worse behavior – inspection of the results shows that this is due to a single triple. In total, Table 11 provides empirical evidence that ExpressivE can capture chained composition patterns and thus perform more than one step of reasoning.

---
[3]https://github.com/idirlab/kgcompletion

Table 11: ExpressivE's MRR on WN18RR in dependence on the number of reasoning steps. Hyphens represent that no new triples can be inferred with additional steps.

|       | 1-Step | 2-Steps | 3-Steps | 4-Steps+ |
|-------|--------|---------|---------|----------|
| $R_1$ | **0.627** | 0.621 | - | - |
| $R_2$ | 0.720 | 0.804 | **0.818** | - |
| $R_3$ | 0.768 | **0.907** | - | - |
| $R_4$ | 0.716 | **0.922** | - | - |

## J  DETAILS OF THE DISTANCE FUNCTION

In this section, we give additional details on the distance function of Equation 3. As in Section 4, let $\boldsymbol{\tau}_{r_i(h,t)}$ denote the embedding of a triple $r_i(h,t)$, i.e. $\boldsymbol{\tau}_{r_i(h,t)} = (\boldsymbol{e}_{ht} - \boldsymbol{c}_i^{ht} - \boldsymbol{r}_i^{th} \odot \boldsymbol{e}_{th})^{|\cdot|}$, with $\boldsymbol{e}_{xy} = (\boldsymbol{e}_x || \boldsymbol{e}_y)$ and $\boldsymbol{a}_i^{xy} = (\boldsymbol{a}_i^x || \boldsymbol{a}_i^y)$ for $\boldsymbol{a} \in \{\boldsymbol{c}, \boldsymbol{r}, \boldsymbol{d}\}$ and $x, y \in \{h, t\}$.

The distance function $D : \boldsymbol{E} \times \boldsymbol{R} \times \boldsymbol{E} \to \mathbb{R}^{2d}$ of Equation 3 — measuring the distance of entity pair embeddings (points) to relation embeddings (hyper-parallelograms) — is split into two parts:

- $D_i(h, r_i, t) = \boldsymbol{\tau}_{r_i(h,t)} \oslash \boldsymbol{w_i}$ for points inside the corresponding relation hyper-parallelogram, i.e., $\boldsymbol{\tau}_{r_i(h,t)} \preceq \boldsymbol{d_i}$.

- $D_o(h, r_i, t) = \boldsymbol{\tau}_{r_i(h,t)} \odot \boldsymbol{w_i} - \boldsymbol{k}$ for points outside the corresponding relation hyper-parallelogram, i.e., $\boldsymbol{\tau}_{r_i(h,t)} \npreceq \boldsymbol{d_i}$.

**Intuition.** As briefly explained in Section 4, the general idea of splitting the distance function is to assign high scores to entity pair embeddings within a hyper-parallelogram and low scores to entity pair embeddings outside the hyper-parallelogram. Specifically, if a triple $r_i(h,t)$ is captured to be true by an ExpressivE embedding, i.e., if $\boldsymbol{\tau}_{r_i(h,t)} \preceq \boldsymbol{d}_i^{ht}$, then the distance correlates inversely with the hyper-parallelogram's width — through the width-dependent factor $\boldsymbol{w_i}$ — keeping low distances/gradients for points within the hyper-parallelogram. Otherwise, the distance correlates — again through the width-dependent factor $\boldsymbol{w_i}$ — linearly with the width to penalize points outside larger parallelograms.

## K  EXPRESSIVE'S TWO NATURES

In this section, we analyze functional and spatial models in more detail and outline how ExpressivE combines the capabilities of both model families. ExpressivE has two natures, specifically:

- ExpressivE has a functional nature (in the spirit of functional models such as TransE and RotatE), allowing it to capture functional composition, discussed in detail in Appendix K.1.

- ExpressivE has a spatial nature (in the spirit of spatial models such as BoxE), allowing it to capture hierarchy, discussed in detail in Appendix K.2.

The combination of the functional and spatial nature is precisely the reason that allows ExpressivE to capture hierarchy and composition patterns jointly. In the following, we review the inference capabilities of spatial and functional models and discuss how ExpressivE combines both the spatial and functional nature.

### K.1  ANALYSIS OF FUNCTIONAL MODELS

We recall the definition of *functional models* provided in Section 3, which states that functional models basically embed relations as functions $\boldsymbol{f}_{r_i} : \mathbb{K}^d \to \mathbb{K}^d$ and entities as vectors $\boldsymbol{e}_j \in \mathbb{K}^d$ over some field $\mathbb{K}$. These models represent true triples $r_i(e_h, e_t)$ as $\boldsymbol{e_t} = \boldsymbol{f}_{r_i}(\boldsymbol{e_h})$ in the embedding space.

Our analysis has revealed that the root cause that functional models cannot capture general composition patterns lies within the functional nature of these models. In essence, these models employ mainly functions to embed relations. This allows them to employ functional composition $\boldsymbol{f}_{r_d} = \boldsymbol{f}_{r_2} \circ \boldsymbol{f}_{r_1}$ to

capture composition patterns. Yet, employing functional composition *defines* the composite relation $r_d$ completely and thus represents a more restricted pattern that we call *compositional definition* $r_1(X, Y) \wedge r_2(Y, Z) \Leftrightarrow r_d(X, Z)$.

In contrast, general composition $r_1(X, Y) \wedge r_2(Y, Z) \Rightarrow r_3(X, Z)$ does *not* completely define its composite relation $r_3$. This means that in the case of general composition, the composite relation $r_3$ may contain more triples than those that are directly *inferable* by compositional definition patterns. Due to this notion of extensibility, we can describe general composition as a combination of compositional definition and hierarchy, i.e., a general composition pattern defines its composite relation $r_3$ as a superset (hierarchy component) of the compositionally defined relation $r_d$. This explains why no KGE has managed to capture general composition, as any SotA KGE that supports some notion of composition cannot represent hierarchy and vice versa (as will be discussed in Appendix K.2) , yet both are essential to support general composition. Therefore, to capture general composition, ExpressivE combines hierarchy and compositional definition patterns, as discussed in more detail in Section 5.2.

### K.2 ANALYSIS OF SPATIAL MODELS

Spatial models embed a relation $r \in \boldsymbol{R}$ via spatial regions in the embedding space. Furthermore, they embed an entity $e_a \in \boldsymbol{E}$ in the role of a head and tail entity with two independent embeddings $\boldsymbol{e_a^h} \in \mathbb{K}^d$ and $\boldsymbol{e_a^t} \in \mathbb{K}^d$. A triple $r(e_h, e_t)$ is true for spatial models if the embeddings of the entities $e_h$ and $e_t$ lie within the respective spatial regions of the relation $r$. Thus, spatial models may capture hierarchy patterns via the spatial subsumption of the regions defined by the relations. However, since there is no relation between $\boldsymbol{e_a^h}$ and $\boldsymbol{e_a^t}$, spatial models — such as BoxE (Abboud et al., 2020) — cannot capture composition.

ExpressivE embeds relations as regions (spatial nature). Yet to achieve the functional nature, it cannot use two independent entity embeddings in the typical embedding space - as we discussed above. The solution and key difference to BoxE is to define the virtual triple space, which is formed by concatenating head and tail entity embeddings of the same embedding space (as described in detail in Section 4). More specifically, any line through the virtual triple space defines a function between head and tail entity embeddings of the same space - the key to the functional nature:

- **Functional nature.** Regions in this virtual triple space establish a mathematical relation between head and tail entities of the same space, by which composition can be captured.

- **Spatial nature.** At the same time, regions can subsume each other, by which - as is intuitive - hierarchy patterns can be captured.

Finally, it is precisely the combination of the functional and spatial nature that allows ExpressivE to capture general composition, as described in detail in Section 5.2.

## L  TRADE-OFF: EXPRESSIVE POWER VS. DEGREES OF FREEDOM

This section discusses the trade-off between a higher expressive power and lower degrees of freedom, observable in the results of Table 3. Specifically, this trade-off manifests in Table 3's benchmark results in the following way:

- **Functional ExpressivE** has a lower expressive power compared to Base ExpressivE as it effectively loses the ability to capture hierarchy patterns. The effect of the reduced expressive power of Functional ExpressivE can be seen in the performance drop on WN18RR over Functional ExpressivE in Table 3. However, since Functional ExpressivE uses fewer parameters than Base ExpressivE, it has a lower degree of freedom, making it less likely to stop in a local minimum than Base ExpressivE as can be seen on Functional ExpressivE's performance on FB15k-237 in Table 3.

- **Base ExpressivE** has the full expressive powers - the high degree of freedom heightening the chance of ending in a local minimum. Table 3 reveals the significant performance increase of Base ExpressivE over Functional ExpressivE on WN18RR, giving evidence that the expressive power is helpful, but the downside being that its higher degrees of freedom

> may make it likelier to stop in a local optimum, manifesting in its performance drop over Functional ExpressivE on FB15k-237.

Further analyzing this trade-off to establish a link between dataset properties and the necessary expressive power of a KGE will be subject for interesting future work.

## M    Experimental Details

This section discusses our experiment setup, benchmark datasets, and evaluation metrics in detail. The concrete experiment setups, including details of our implementation, used hardware, learning setup, and chosen hyperparameters, are discussed in Subsection M.1. Subsection M.2 lists properties of the used benchmark datasets and Subsection M.3 lists properties of the used ranking metrics.

### M.1    Experiment Setup and Emissions

**Implementation Details.**   We have implemented ExpressivE in PyKEEN 1.7 (Ali et al., 2021), which is a Python library that uses the MIT license and supports many benchmark KGs and KGEs. Thereby, we make ExpressivE comfortably accessible to the community for future benchmarks and experiments. We have made our code publicly available in a GitHub repository[4]. It contains, in addition to the code of ExpressivE, a setup file to install the necessary libraries and a ReadMe.md file containing library versions and running instructions to facilitate the reproducibility of our results.

**Training Setup.**   Each model was trained and evaluated on one of 4 GeForce RTX 2080 GPUs of our internal cluster. Specifically, the training process uses the Adam optimizer (Kingma & Ba, 2015) to optimize the self-adversarial negative sampling loss (Sun et al., 2019). ExpressivE is trained with gradient descent for up to 1000 epochs with early stopping, finishing the training if after 100 epochs the Hits@10 score did not increase by at least $0.5\%$ for WN18RR and $1\%$ for FB15k-237. We have increased the patience for OneBand ExpressivE to 150 epochs for FB15k-237, as it converges slower than the other ablation versions of ExpressivE. We use the model of the final epoch for testing. Each experiment was repeated three times to account for small performance fluctuations. In particular, the MRR values fluctuate by less than 0.003 between runs for Base and Functional ExpressivE on any dataset. We performed hyperparameter tuning over the learning rate $\lambda$, embedding dimensionality $d$, number of negative samples $neg$, loss margin $\gamma$, adversarial temperature $\alpha$, and minimal denominator $D_{min}$. Specifically, two mechanisms were employed to implicitly regularize the hyper-parallelogram: (1) the hyperbolic tangent function $tanh$ was element-wise applied to each entity embedding $e_p$, slope vector $r_i^p$, and center vector $c_i^p$, projecting them into the bounded space $[-1,1]^d$, and (2) the size of each hyper-parallelogram is limited by the novel $D_{min}$ parameter. In the following, we will briefly introduce the $D_{min}$ parameter and its function.

**Minimal Denominator $D_{min}$.**   As can be easily shown, Equations 108 describe the relation hyper-parallelogram's center, and Equations 109-110 its corners in the virtual triple space.

$$center_i^h = \frac{c_i^h + r_i^t c_i^t}{1 - r_i^h r_i^t} \qquad \text{and} \qquad center_i^t = \frac{r_i^h c_i^h + c_i^t}{1 - r_i^h r_i^t} \tag{108}$$

$$cornA_i^h = center_i^h \pm \frac{d_i^h + r_i^t d_i^t}{1 - r_i^h r_i^t} \qquad \text{and} \qquad cornA_i^t = center_i^t \pm \frac{r_i^h d_i^h + d_i^t}{1 - r_i^h r_i^t} \tag{109}$$

$$cornB_i^h = center_i^h \pm \frac{d_i^h - r_i^t d_i^t}{1 - r_i^h r_i^t} \qquad \text{and} \qquad cornB_i^t = center_i^t \pm \frac{r_i^h d_i^h - d_i^t}{1 - r_i^h r_i^t} \tag{110}$$

Note that the denominator of each term is equal to $(1 - r_i^h r_i^t)$. Since a small denominator in Equations 109 and 110 produces large corners and, therefore, a large hyper-parallelogram, we have introduced the hyperparameter $D_{min}$, allowing ExpressivE to tune the maximal size of its hyper-parallelograms. In particular, $D_{min}$ constrains the relation embeddings such that $(1 - r_i^h r_i^t) \preceq D_{min}$, thereby constraining the maximal size of a hyper-parallelogram as required.

---

[4]https://github.com/AleksVap/ExpressivE

**Hyperparameter Optimization.**    Following Abboud et al. (2020), we have varied the learning rate by $\lambda \in \{a * 10^{-b} | a \in \{1, 2, 5\} \wedge b \in \{-2, -3, -4, -5, -6\}\}$, the margin $m$ by integer values between 3 and 24 inclusive, the adversarial temperature by $\alpha \in \{1, 2, 3, 4\}$, and the number of negative samples by $neg \in \{50, 100, 150\}$. Furthermore, we have varied the novel minimal denominator parameter by $D_{min} \in \{0, 0.5, 1\}$. We have tuned the hyperparameters of ExpressivE manually within the specified ranges. Finally, to allow a direct performance comparison of ExpressivE to its closest spatial relative BoxE and its closest functional relative RotatE, we chose for each benchmark the embedding dimensionality and negative sampling strategy of the best-performing RotatE and BoxE model (Abboud et al., 2020; Sun et al., 2019). Concretely we chose self-adversarial negative sampling (Sun et al., 2019) and the embedding dimensionalities listed in Table 12. The best performing hyperparameters for ExpressivE on each benchmark dataset are listed in Table 12. We have used the hyperparameters of Table 12 for any considered version of ExpressivE — namely Base, Functional, EqSlopes, NoCenter, and OneBand ExpressivE —, which are described in the ablation study of Section 6.2.

Table 12: Hyperparameters for the best-performing ExpressivE models on WN18RR and FB15k-237.

| Dataset | Embedding Dimensionality | Margin | Learning Rate | Adversarial Temperature | Negative Samples | Batch Size | Minimal Denominator |
|---|---|---|---|---|---|---|---|
| WN18RR | 500 | 3 | $1 * 10^{-3}$ | 2 | 100 | 512 | 0 |
| FB15k-237 | 1000 | 4 | $1 * 10^{-4}$ | 4 | 150 | 1024 | 0.5 |

**CO2 Emission Related to Experiments.**    The computation of the reported experiments took below 200 GPU hours. On an RTX 2080 (TDP of 215W) with a carbon efficiency of 0,432 kg/kWh (based on the OECD's 2014 yearly carbon efficiency average), 200 GPU hours correspond to a rough $CO_2$ emission of 18.58 kg $CO_2$-eq. The estimations were conducted using the MachineLearning Impact calculator (Lacoste et al., 2019).

## M.2   BENCHMARK DATASETS

This section briefly discusses some details of the standard KGC benchmark datasets WN18RR (Dettmers et al., 2018) and FB15k-237 (Toutanova & Chen, 2015). In particular, Table 13 lists the following characteristics of the benchmark datasets, namely their number of: entities $|\boldsymbol{E}|$, relation types $|\boldsymbol{R}|$, training, testing, and validation triples. Both WN18RR and FB15k-237 provide training, testing, and validation splits, which were directly used in our experiments.

Table 13: Benchmark dataset characteristics.

| Dataset | $|\boldsymbol{E}|$ | $|\boldsymbol{R}|$ | Training Triples | Validation Triples | Testing Triples |
|---|---|---|---|---|---|
| FB15k-237 | 14,541 | 237 | 272,115 | 17,535 | 20,466 |
| WN18RR | 40,943 | 11 | 86,835 | 3,034 | 3,034 |

We have not found licenses for FB15k-237 nor WN18RR. WN18RR is a subset of WN18 (Bordes et al., 2013), whose license is also unknown, yet FB15k-237 is a subset of FB15k (Bordes et al., 2013) that uses the CC BY 2.5 license.

## M.3   METRICS

We have evaluated ExpressivE by measuring the ranking quality of each test set triple $r_i(e_h, e_t)$ over all possible head $e'_h$ and tail $e'_t$: $r_i(e'_h, e_t)$ for all $e'_h \in \boldsymbol{E}$ and $r_i(e_h, e'_t)$ for all $e'_t \in \boldsymbol{E}$. The mean reciprocal rank (MRR), and Hits@k are the standard evaluation metrics for this evaluation (Bordes et al., 2013). In particular, we have reported the filtered metrics (Bordes et al., 2013), i.e., where all triples that occur in the training, validation, and testing set (except the test triple that shall be ranked) are removed from the ranking, as ranking these triples high does not represent a faulty inference. Furthermore, the filtered MRR, Hits@1, Hits@3, and Hits@10 are the most widely used metrics for evaluating KGEs (Sun et al., 2019; Trouillon et al., 2016; Balazevic et al., 2019; Abboud et al., 2020).

Finally, we will briefly discuss the definitions of these metrics: the MRR represents the average of inverse ranks ($1/rank$), and Hits@k represents the proportion of true triples within the predicted triples whose rank is at maximum k.

