# OpenReview forum: "ExpressivE: A Spatio-Functional Embedding For Knowledge Graph Completion"
_ICLR.cc/2023/Conference — ICLR 2023 notable top 25%_

### Official Review · Reviewer_MZLX · 2022-10-17

**Confidence:** 3
**Correctness:** 4
**Technical Novelty And Significance:** 4
**Empirical Novelty And Significance:** 4
**Recommendation:** 8

**Clarity, Quality, Novelty And Reproducibility:**

- This paper is written clearly, and the contents are well-organized.
- The proposed method is novel and easy to reproduce.

**Strength And Weaknesses:**

Pros
- This paper can fully capture vital inference patterns(e.g., capturing both general composition and hierarchy jointly), which is meaningful for KGC.
-  Interpreting multiple inference patterns within the virtual triple space is interesting.
- The analysis of ExpressivE's expressive power is convincing.
- The experiment results(especially on WN18RR) are satisfied.

Cons
- Currently, there are no critical problems in this paper for me.

**Summary Of The Paper:**

This paper proposed a spatio-functional embedding model named ExpressivE, which has the benefits of both region-based and functional models. Compared to present methods, ExpressivE can (1) fully capture vital inference patterns, (2) capture prominent logical rules jointly, and (3) provide an intuitive interpretation of captured patterns. The following experiments on the KGC benchmarks showed the great performance of ExpressivE.

**Summary Of The Review:**

This paper solves previous KGC embedding methods' incapability of several inference patterns.
The proposed methods are both powerful and explainable.
The article is valuable and worth being accepted.

---

> ### Author Response · Authors · 2022-11-13
> **Thank you!**
>
> Thank you for your interest, positive comments on the significance, and kind words.

---

### Official Review · Reviewer_KcUq · 2022-10-20

**Confidence:** 5
**Correctness:** 3
**Technical Novelty And Significance:** 3
**Empirical Novelty And Significance:** 2
**Recommendation:** 8

**Clarity, Quality, Novelty And Reproducibility:**

## Clarity
This paper is clearly written and well organized. The authors provide a sufficient background introduction and thus the paper is easy to follow.

## Quality
Most of the claims are well supported by either rigorous proofs or experimental results, while some needs more clarification. (See Strength And Weaknesses) I do not check the theoretical proof in the Appendix but they look nice.

## Novelty
The idea of using hyper-parallelograms to design a geometric KGE model is mainly inspired by BoxE, but the way to realize it seems novel to me.

## Reproducibility
Detailed experimental settings are provided for reproducibility. However, the codes are not uploaded as the authors claimed.


**Strength And Weaknesses:**

## Strength
1. The idea of using hyper-parallelograms to design a new KGE model seems novel and interesting to me.
2. The proposed model has many good properties and the authors provide rigorous theoretical proof for them.
3. The model performs well while using only half the parameters compared with the previous state-of-the-art methods.
4. The authors did a good job of introducing the background and related work.
5. This paper is clearly written and well organized.

## Weakness
1. **Abstract**
    The authors claim that ExpressivE can capture prominent *logical rules* jointly. However, the term *logical rules* never appears again in the main text. So what does this claim mean?
2. **Motivation**
    The main motivation of this work is to capture general composition and the authors claim that the ability to capture it is vital for KGE models to reason over graphs. However, is there any evidence to support this claim? I admit that it'd be better if a model can capture general composition, but there may not exist a strong connection between this ability and the final KGC performance.
3. **Method**
    3.1 What does the term $\mathbf{w}_i$ in Equation (3) stand for?
    3.2 The authors list some key properties of ExpressivE on Page 7, while there lacks necessary support for them. For example, the authors may want to give some graphic examples to show the intuitive geometric interpretation. Experimental results for 1-n, n-1, and n-n relations should be provided to support the third properties.
4. **Experiments**
    4.1 I am confused about the worse performance of Base ExpressivE compared with Functional ExpressivE on FB15k-237. Even if FB15k-237 does not contain many hierarchy patterns, I expect that the base version of ExpressivE performs no worse than the functional one, since the width can be learned to be zero. Otherwise, the proposed method can only perform well on knowledge graphs with some special properties, which will significantly limit its applicability.
    4.2 It'd be better conduct experiments to show that ExpressivE can indeed model general composition, since it is the main contribution of this work.
5. **Reproducibility**
    The authors claim that they have included the supplementary material via a link in an OpenReview comment. However, I cannot find the link.

**Summary Of The Paper:**

This paper proposes a new geometry-based knowledge graph embedding method named ExpressivE, which represents entities as points and relations as hyper-parallelograms. Extensive theoretical analyses show that ExpressivE is fully expressive and is capable of capturing various inference patterns. Experiments demonstrate that ExpressivE achieves competitive performance.

**Summary Of The Review:**

This paper proposes a novel geometry-based knowledge graph embedding model for knowledge graph completion. It has many theoretical merits and obtains competitive results. However, I think this paper can be greatly improved if some concerns about the Strength And Weaknesses can be properly addressed, and I am open to raising my score if the authors manage to do it.

======Update After Rebuttal======
I am satisfied with the authors' response and raise my score accordingly.

---

> ### Author Response · Authors · 2022-11-13
> **Brief Response Summary**
>
> We want to thank the reviewer for their constructive and helpful feedback, and in particular their significant time investment. Addressing the raised questions has led us to perform more experiments, which provide even more empirical evidence supporting the extensive theoretical capabilities of ExpressivE. To ease the identification of additions to our paper, we have marked added sections and texts in response to your questions in green. In the following, we summarize the actions taken in response to your questions.
>
> **Brief Summary**
>
> In response to your questions, we have added:
> * Experimental results for ExpressivE's performance on 1-1, 1-N, N-1, and N-N relations (Appendix I.1).
> * Experimental evidence for ExpressivE's capability to capture general composition, which in addition provides support for the link between ExpressivE's inference capabilities and the performance improvement on WN18RR (Appendix I.2).
> * Discussion of the trade-off of high expressive power vs. low degrees of freedom (Appendix L).
> * Responses to all other points raised by the reviewer (terminology, etc.).
>
>
> Regarding your question about the availability of our code: Before the discussion phase, it was not possible for us to post a comment for some reason. In response to your question, we have immediately posted the OpenReview comment **Reproducibility: Sharing Code and Pre-trained Models**, containing a link to our code and pre-trained models.
>
> Finally, we have addressed your main questions in detail in the following comment.

---

> > ### Author Response · Authors · 2022-11-13
> > **Response to Major Questions**
> >
> > **3.2. Performance on 1-N, N-1, and N-N**
> >
> > As requested by the reviewer, we have provided experimental results to support our theoretical result that ExpressivE performs well on 1-N, N-1, and N-N relations.
> >
> > |Prediction|**Head**|**Head**|**Head**|**Head**|**Tail**|**Tail**|**Tail**|**Tail**|
> > |-|-|-|-|-|-|-|-|-|
> > |Cardinality|1-1|1-N|N-1|N-N|1-1|1-N|N-1|N-N|
> > |ExpressivE|**0.976**|*0.290*|*0.105*|**0.941**|**0.976**|*0.141*|**0.327**|**0.938**|
> > |RotatE|0.833|**0.294**|0.103|*0.930*|0.875|0.107|*0.288*|*0.925*|
> > |BoxE|*0.877*|0.272|**0.146**|0.883|*0.893*|**0.147**|0.246|0.884|
> >
> >
> > Following the procedure of Bordes et al. (2013), we have categorized the relations of WN18RR into four cardinality classes: 1-1, 1-N, N-1, and N-N (details in Appendix I.1.).
> > Table 9 (above) lists the MRR of ExpressivE and its closest spatial relative BoxE and functional relative RotatE on WN18RR, stratified by the four cardinality classes. It reveals that ExpressivE almost exclusively reaches a (close-to-)state-of-the-art performance on 1-N, N-1, and N-N relations. In particular, ExpressivE outperforms both RotatE and BoxE consistently on the N-N class, which is often considered the most complex cardinality class. Thus, Table 9 empirically supports our theoretical claim that ExpressivE can capture 1-1, 1-N, N-1, and N-N relations well.
> >
> > As mentioned, we have added this discussion and experimental details in Appendix I.1.
> >
> > &nbsp;
> >
> > **Evidence for 4.2. capturing general composition and 2. link to performance improvement**
> >
> > To provide the requested empirical evidence that ExpressivE can capture general composition (Point 4.2) and that capturing general composition is linked to its performance gain on WN18RR (Point 2), our first goal was to identify the patterns occurring in WN18RR. To reach this goal, we have analyzed the patterns of WN18RR provided by Akrami et al. (2020) in a public GitHub repository and identified the most relevant patterns (details in Appendix I.2).
> >
> > **Experimental Setup.** For each of the selected patterns $\rho$ we have computed all triples that (i) can be derived by $\rho$ from the data known to our model and (ii) are known to be true in the KG, yet unseen to our models. Finally, we have computed the MRR of ExpressivE on each of these sets of triples (details in Appendix I.2).
> >
> > **Hypotheses**. Note that (as discussed in Appendix K.1) compositional definition $r_1(X,Y) \land r_2(Y,Z) \iff r_3(X,Z)$ defines the triples of the composite relation $r_3$ completely, whereas general composition $r_1(X,Y) \land r_2(Y,Z) \implies r_3(X,Z)$ allows $r_3$ to contain more triples than those that the compositional definition pattern can directly infer. Thus, if ExpressivE captures general composition and RotatE captures compositional definition, we expect the following behavior:
> > * **H1**. RotatE will perform well solely on relations occurring as the head of maximally one composition pattern, as RotatE solely supports compositional definition.
> > * **H2**. ExpressivE will perform well even when a relation is defined by multiple composition patterns and/or multiple other patterns since ExpressivE supports general composition.
> >
> >
> >
> > |Head Rel.|**\_v\_g**|**\_v\_g**|**\_v\_g**|**\_v\_g**|**\_a\_s**|**\_a\_s**|**\_a\_s**|**\_syn\_dto**|
> > |-|-|-|-|-|-|-|-|-|
> > |Model|$S_1$|$C_2$|$C_3$|$C_4$|$C_5$|$C_6$|$S_7$|$C_8$|
> > |Base Exp.|**1.000**|**1.000**|**1.000**|**1.000**|**0.818**|**0.907**|**0.985**|**0.621**|
> > |RotatE|0.865|0.760|0.760|0.760|0.771|0.893|0.975|0.599|
> > |BoxE|0.906|0.801|0.806|0.806|0.632|0.645|0.727|0.547|
> >
> > **Results**. Table 10 (above) lists for each pattern $S_1$ to $C_8$ the performances of BoxE, RotatE, and ExpressivE stratified by pattern $\rho \in \{S_1, \dots, C_8\}$, where $S_i$ represents a symmetry pattern and $C_i$ represents a composition pattern. Table 10 provides evidence for both hypotheses:
> > * **Evidence for H1**. In the case of the relation *\_syn\_dto*, there is only one pattern with *\_syn\_dto* as its head relation, specifically the composition pattern $C_8$. RotatE achieves comparable performance to ExpressivE on $s_{C_8}$ as RotatE is capable of defining *\_syn\_dto* using compositional definition, providing evidence for H1.
> > * **Evidence for H2**. Yet, when a relation is defined via multiple patterns, RotatE’s performance decreases drastically on most composition patterns compared to ExpressivE’s performance, as can be seen for the patterns $C_2$, $C_3$, $C_4$, and $C_5$, giving evidence for H2.
> >
> > **Conclusion** Thus, these experiments provide empirical evidence for Point 4.2, as ExpressivE and RotatE perform as expected by H1 and H2 under the assumption that ExpressivE captures general composition and that RotatE captures compositional definition. Furthermore, the experiments also provide evidence for Point 2, as ExpressivE consistently outperforms RotatE and BoxE on each composition pattern $C_i$.
> >
> > As mentioned before, the complete details of the experiments discussed here are included in Appendix I.2.

---

> > > ### Author Response · Authors · 2022-11-13
> > > **Response to Minor Questions**
> > >
> > > In addition to addressing the main questions, we have addressed the minor questions raised by the reviewer as follows:
> > >
> > > &nbsp;
> > >
> > > **1.1. Abstract (What does "logical rules" mean)**
> > >
> > > Thank you for catching this - the term "logical rules" in the abstract was used synonymously for “inference patterns” (short: patterns). To improve our terminology's consistency, we have replaced "logical rules" with "patterns" in the abstract.
> > >
> > > &nbsp;
> > >
> > > **3.1. Method (What does $w_i$ stand for?)**
> > >
> > > The term $w_i$ stands for a width-dependent factor as it grows proportional to the width of the parallelogram. Specifically, it is used to scale the distance function in dependence of the parallelogram's width, penalizing points outside larger hype-parallelograms.
> > >
> > > To aid the reader in quickly understanding the distance function, we have added a note on this in the paragraph "intuition" of Appendix J, which discusses the goals and properties of the distance function in more detail.
> > >
> > > &nbsp;
> > >
> > > **4.1. Trade-Off: High Expressive Power vs. Low Degrees of Freedom**
> > >
> > > The trade-off between high expressive power and low degrees of freedom is a good point and should be discussed:
> > > * **Functional ExpressivE** has a lower expressive power than Base ExpressivE as it effectively loses the ability to capture hierarchy patterns. The effect of the reduced expressive power of Functional ExpressivE can be seen in the performance drop on WN18RR over Functional ExpressivE in Table 3. However, since Functional ExpressivE uses fewer parameters than Base ExpressivE, it has a lower degree of freedom, making it less likely to stop in a local minimum than Base ExpressivE, as can be seen on Functional ExpressivE's performance on FB15k-237 in Table 3.
> > > * **Base ExpressivE** has the full expressive powers - the high degree of freedom heightening the chance of ending in a local minimum. Table 3 reveals the significant performance increase of Base ExpressivE over Functional ExpressivE on WN18RR, giving evidence that the expressive power is helpful, but the downside being that its higher degrees of freedom may make it likelier to stop in a local optimum, manifesting in its performance drop over Functional ExpressivE on FB15k-237.
> > >
> > >
> > > This trade-off is interesting to the reader - we have added discussion on it to the paper (Appendix L) - thank you.
> > >
> > > &nbsp;
> > >
> > > **5. Reproducibility (Is the code available?)**
> > >
> > > It was previously not possible to upload the material via OpenReview for some reason. It works now - and we have made the code available.
> > >
> > > In response to the reviewers' question, we immediately made the code of ExpressivE available to the reviewers, the program, and area chairs via the OpenReview comment **Reproducibility: Sharing Code and Pre-trained Models**.

---

> > > > ### Comment · Reviewer_KcUq · 2022-11-14
> > > > **Thanks for your detailed response**
> > > >
> > > > Your response has addressed my concerns and now I have raised my score from 5 to 8.

---

### Official Review · Reviewer_kHNH · 2022-10-24

**Confidence:** 5
**Clarity, Quality, Novelty And Reproducibility:** see above
**Correctness:** 4
**Technical Novelty And Significance:** 4
**Empirical Novelty And Significance:** 4
**Recommendation:** 10

**Strength And Weaknesses:**

Strengths:
1. The paper is well motivated and presented.
2. The proposed method is insightful and elegant.
3. The experiments demonstrate the effectiveness.

Weaknesses:

I didn't see any weaknesses of the paper, except some minor questions:

1) Will the proposed method bring too much expressiveness, and thus weaken the generalization ability to capture more inference pattersn using limited data?

2) In the section parameter interpretation, why does the intersection of two bands result in another band?

3) In Section Scoring Function, please provide more explanation of why k keeps the function continuous.

**Summary Of The Paper:**

The paper proposes a novel spatial KGC model that can capture many inference patterns, including compositional and general compositional. In specific, the authors define each entity as points in the vector space, and each relation as hyper-parallelograms in the virtual triple space, which is constructed by the concatenation of head and tail entity vectors and the jth and (d+j)th dimensions bring a correlation subspaces. Thus, the intersection of two parallel half-spaces constitute a band and two bands form a parallelogram to post constraints on the triple. Furthermore, the authors provide proof on the proposed model's capacity in modeling various types of inference patterns. In experiments, the results on two widely used datasets show improvements.

**Summary Of The Review:**

see above

---

> ### Author Response · Authors · 2022-11-13
> **Response to Questions**
>
> We want to thank the reviewer for their positive feedback and the extremely interesting questions raised. To ease the identification of additions to our paper, we have marked added sections and texts in response to your questions in blue. Specifically, we have answered the raised questions as follows:
>
> **1. Will the proposed method bring too much expressiveness, and thus weaken the generalization ability to capture more inference patterns using limited data?**
>
> Thank you - this question is very interesting - the gist of it could be summarized as the following three points:
> * In **general** with limited data, given that ExpressivE uses half the number of parameters as BoxE and RotatE, one would expect ExpressivE to have better generalization capabilities than BoxE and RotatE.
> * In **specific** cases with limited patterns in the data, there is interesting evidence that datasets may not exploit this additional expressive power: The results of Base ExpressivE compared to Functional ExpressivE on FB15k-237 gives such cases (see Table 3).
> * Now, a solution for cases with limited patterns in data is the one we have chosen in Table 3, namely constraining the parameters in ExpressivE to reduce its expressive power, obtaining e.g., the model we call Functional ExpressivE - which led to the significant performance gain on FB15k-237 over Base ExpressivE.
>
> Still, we think that this line of work - exploring models’ behavior in the presence of limited patterns in the data - is a very interesting research question that could establish a new line of work, and want to thank the reviewer for this.
>
> We have extended the conclusion with promising future work in this regard - thank you!
>
> &nbsp;
>
> **2. In the section parameter interpretation, why does the intersection of two bands result in another band?**
>
> This only happens in a special case: Specifically, when one band subsumes the other, the intersection of two bands will result in another band. In all other cases, the intersection results in a bounded parallelogram. For our constructions, it is sufficient to consider bounded parallelograms solely.
>
> We have added a note on this in Section 4 (paragraph "parameter interpretation").
>
> &nbsp;
>
> **3. In Section Scoring Function, please provide more explanation of why k keeps the function continuous.**
>
> We want to thank the reviewer for this question, as it aided us in adding more explanations that hopefully help the reader progress through this section more smoothly. We have added a section in Appendix J that provides additional details on the distance function, in particular including why $\mathbf{k}$ keeps the distance function continuous. We have included the relevant parts in the following:
>
> As in Section 4, let $\mathbf{\tau_{r_i(h,t)}}$ denote the embedding of a triple $r_i(h, t)$, i.e., $\mathbf{\tau_{r_i(h,t)}} = (\mathbf{e_{ht}} - \mathbf{c_i^{ht}} - \mathbf{r_i^{th}} \odot \mathbf{e_{th}})^{|.|}$, with $\mathbf{e_{xy}} = (\mathbf{e_{x}} || \mathbf{e_{y}})$ and $\mathbf{a_i^{xy}} = (\mathbf{a_i^{x}} || \mathbf{a_i^{y}})$ for $\mathbf{a} \in \{\mathbf{c}, \mathbf{r} , \mathbf{d}\}$ and $\mathbf{x}, \mathbf{y} \in \{\mathbf{h}, \mathbf{t}\}$.
>
> The distance function $D: \mathbf{E} \times \mathbf{R} \times \mathbf{E} \rightarrow \mathbb{R}^{2d}$ of Equation 3 — measuring the distance of entity pair embeddings (points) to relation embeddings (hyper-parallelograms) — is split into two parts:
> * $D_i(h,r_i,t) = \mathbf{\tau_{{r_i}(h,t)}} \oslash \mathbf{w_i}$ for points inside the corresponding relation hyper-parallelogram, i.e., $\mathbf{\tau_{r_i(h, t)}} \preceq \mathbf{d_i}$.
> * $D_o(h,r_i,t) = \mathbf{\tau_{{r_i}(h,t)}} \odot \mathbf{w_i} - \mathbf{k}$ for points outside the corresponding relation hyper-parallelogram, i.e., $\mathbf{\tau_{r_i(h, t)}} \succ \mathbf{d_i}$.
>
> **Continuity**. Since $D$ consists of two piece-wise linear functions, measures need to be taken to prevent discontinuities. Specifically, if the entity pair embedding $\mathbf{e_{ht}} = (\mathbf{e_{h}} || \mathbf{e_{t}})$ of the entities $h, t \in \mathbf{E}$ lies at the border of the hyper-parallelogram of relation $r_i \in \mathbf{R}$ (i.e., if $\mathbf{\tau_{r_i(h, t)}} = \mathbf{d_i}$), then we want to ensure that $D_i(h, r_i, t) = D_o(h, r_i, t)$ such that the overall distance $D$ defined over $D_i$ and $D_o$ is continuous, leading to:
>
> $D_i(h, r_i, t) = D_o(h, r_i, t)$
>
> $\mathbf{\tau_{r_i(h, t)}} = \mathbf{d_i}$
>
> Substituting the definitions of $D_i$ and $D_o$ in the equations above and solving the resulting system of equations for $\mathbf{k}$ results in the solution $\mathbf{k} = \mathbf{0.5} \odot (\mathbf{w_i} - \mathbf{1}) \odot (\mathbf{w_i} - \mathbf{1} \oslash \mathbf{w_i})$. Thereby, the coefficient $\mathbf{k}$ keeps the inside distance $D_i$ and outside distance $D_o$ equal at the parallelogram’s border and thus - as briefly mentioned in Section 4 - keeps $D$ continuous.
>
> As mentioned above, we have added this explanation to Appendix J.

---

### Official Review · Reviewer_cMBH · 2022-10-24

**Confidence:** 4
**Correctness:** 4
**Technical Novelty And Significance:** 3
**Empirical Novelty And Significance:** 2
**Recommendation:** 6

**Clarity, Quality, Novelty And Reproducibility:**

The paper is well-written. The proposed model is novel and the detail for reproducing the results are presented in the paper.

**Strength And Weaknesses:**

**Strengths**

Many existing papers have tried to model inference patterns and one that many baselines fail to capture is general composition.

**Weaknesses**


Datasets: Recent works [1, 2] show that previous existing KG completion benchmarks have quality issues. For instance, [1] has shown that a large percentage of relations in FB15K-237 could be covered by a trivial frequency rule. Also, there are recent KGs available in ogbl-biokg and wikikg datasets that are of larger scale and closer to practical cases.

[1] Tara Safavi and Danai Koutra. CoDEx: A Comprehensive Knowledge Graph Completion Benchmark, EMNLP 2020.

[2] Genet Asefa Gesese, Mehwish Alam, and Harald Sack. LiterallyWikidata - A Benchmark for Knowledge Graph Completion Using Literals, SEMWEB 2021.


**Summary Of The Paper:**

This paper proposes a knowledge graph completion model that is fully expressive and can capture some inference patterns. The most important inference pattern that the proposed model is able to capture is general composition. This is mainly because many existing models cannot capture composition. The experiments show the model is competitive with the state-of-the-art on FB15k-237 and outperforms the baselines on WN18RR.

**Summary Of The Review:**

**Questions**

"Vital" inference patterns: I would like the authors to elaborate more on why the set of inference patterns in Table 1 is considered vital. Also, what are some of the patterns that the proposed model cannot capture (limitations of the proposed model)?

Reading through the main paper, I do not seem to get what is the main property of ExpressivE that causes it to be able to capture composition. The proposed mode is similar to the class of models that BoxE belongs to. Why BoxE cannot model composition but ExpressivE can. Can the authors elaborate more on this with intuitions?

Can the authors elaborate more on whether ExpressivE can model $r_1(X, Y) \land r_2(Y,Z) -> r_3(X, Z)$ and $r_3(X, Y) \land r_4(Y, Z) -> r5(X, Z)$? This is more challenging as there are multiple steps of reasoning needed to be done to be able to capture a composition of multiple steps.

**Minor**

Some references in the appendix to the Theorems seem to be broken.

---

> ### Author Response · Authors · 2022-11-13
> **Brief Response Summary**
>
> We want to thank the reviewer for their highly interesting questions, which guided us in further empirical and theoretical investigations of our model, providing even more evidence for ExpressivE's inference capabilities. We have in the following briefly summarized and answered your questions.
>
> **Brief Summary**
>
> In response to your questions, we have added:
>
> * A detailed discussion of the source of ExpressivE's capability to capture composition compared to other spatial models (Appendix K and K.2).
>
> * A theoretical proof for ExpressivE's capability to capture the multi-step pattern mentioned by the reviewer (Appendix H).
>
> * Empirical evidence for ExpressivE's capability to capture multi-step patterns occurring in the real-world dataset WN18RR (Appendix I.3).
>
> * Discussion on the use of composition and hierarchy patterns in state-of-the-art models and limitations (in more detail in the response, and as far as space permitted, in the introduction and the conclusion).
>
> * Discussion of further datasets (including multimodal datasets) for which we added a note on interesting future work in the conclusion.
>
> We have marked additions to our paper in response to your questions in red and provide extensive details on these points in the comments below.

---

> > ### Author Response · Authors · 2022-11-13
> > **Response to Questions 1**
> >
> > **How does ExpressivE capture composition and why does BoxE not?**
> >
> > Thank you - the answer to this question touches the core of ExpressivE. Section 5.2 currently explores how ExpressivE captures general composition. However, comparing its capabilities in detail to spatial models (such as BoxE) will further highlight their differences. In response to your comment:
> >
> > * We present a compact comparison of ExpressivE to spatial models below.
> >
> > * So that future readers benefit from it, we have also added this as Appendices K and K.2 of the revised paper.
> >
> > ExpressivE has a:
> >
> > * functional nature, allowing it to capture functional composition.
> >
> > * spatial nature, allowing it to capture hierarchy.
> >
> > The combination of the functional and spatial nature is precisely the reason that allows ExpressivE to capture hierarchy and composition jointly.
> > In response to your questions, we review the inference capabilities of spatial models and discuss how ExpressivE extends them:
> >
> > **Spatial models** embed a relation $r \in \mathbf{R}$ via spatial regions in the embedding space. Furthermore, they embed an entity $e_a \in \mathbf{E}$ in the head/tail role with two independent embeddings $\mathbf{e^h_{a}} \in \mathbb{K}^d$ and $\mathbf{e^t_{a}} \in \mathbb{K}^d$. A triple $r(e_h,e_t)$ is true for spatial models if the entity embeddings $e_h$ and $e_t$ lie within the respective spatial regions of $r$. Thus, spatial models may capture hierarchy via spatial subsumptions. However, since there is no relation between $\mathbf{e^h_{a}}$ and $\mathbf{e^t_{a}}$, spatial models - such as BoxE (Abboud et al., 2020) - cannot capture composition.
> >
> > **ExpressivE** embeds relations as regions (spatial nature). Yet to achieve the functional nature, it cannot use two independent entity embeddings in the typical embedding space - as discussed above. The solution and key difference to BoxE is to define the virtual triple space, which is formed by concatenating head and tail entity embeddings of the same space (as described in detail in Section 4) - the key to the functional nature:
> >
> > * **Functional nature**. Regions in this virtual triple space establish a mathematical relation between head and tail entities of the same space, by which composition can be captured.
> >
> > * **Spatial nature**. At the same time, regions can subsume each other, by which - as is intuitive - hierarchy patterns can be captured.
> >
> > Finally, the combination of these two natures allows ExpressivE to capture general composition as described in detail in Section 5.2.
> >
> > We have added this explanation in Appendix K and K.2.
> >
> > &nbsp;
> >
> > **Can ExpressivE capture $r_1(X,Y) \land r_2(Y,Z) \implies r_3(X,Z)$ and $r_3(X,Y) \land r_4(Y,Z) \implies r_5(X,Z)$?**
> >
> > We want to thank the reviewer for this highly interesting question. In response to your question:
> >
> > * We have added a proof in Appendix H showing that ExpressivE can capture the described scenario.
> >
> > * To further support this theoretical claim, we have provided some preliminary experimental evidence in Appendix I.3. We present the experimental results of Appendix I.3 in what follows.
> >
> > To evaluate how well ExpressivE supports more than one step of composition, our first goal was to identify multi-step patterns (i.e., patterns of the shape the reviewer described) occurring in WN18RR. To reach this goal, we have analyzed patterns of WN18RR provided by Akrami et al. (2020) in a public GitHub repository and identified the most relevant multi-step patterns (details in Appendix I.3).
> >
> > For each of the selected multi-step patterns $\rho \in \{R_1, R_2, R_3, R_4\}$, we have generated three datasets, the 1-Step, 2-Steps, and 3-Steps sets. Specifically, we have generated for each $\rho$ a j-Step(s) set by computing all triples that (i) can be derived by $\rho$ in $j$ steps from the data known to our model and (ii) are known to be true in the KG, yet unseen to our model (details in Appendix I.3).
> > The MRR of ExpressivE on the computed datasets is summarised in Table 11 (listed below).
> >
> > | | 1-Step | 2-Steps | 3-Steps | 4-Steps+ |
> > |-|-|-|-|-|
> > | $R_1$ | **0.627** | 0.621 | - | - |
> > | $R_2$ | 0.720 | 0.804 | **0.818** | - |
> > | $R_3$ | 0.768 | **0.907** | - | - |
> > | $R_4$ | 0.716 | **0.922** | - | - |
> >
> > Hyphens in the table represent that no new triples can be inferred with additional steps. We can see that the performance of ExpressivE increases by a large margin when more than one step of reasoning is considered, depicted by the performance gain of the 2-Steps and 3-Steps set over the 1-Step set. Interestingly, a small exception for this is $R_1$, where we see slightly worse behavior - inspection of the results shows that this is due to a single triple. In total, Table 11 provides empirical evidence that ExpressivE can capture multi-step composition.
> >
> > We have added the complete details in Appendix I.3 and - as mentioned - the theoretical proof in Appendix H.

---

> > > ### Author Response · Authors · 2022-11-13
> > > **Response to Questions 2**
> > >
> > > **Why are the considered inference patterns vital?**
> > >
> > > **State-of-the-art motivation**.
> > > While there are many patterns that could be considered vital, our main focus here in this paper is hierarchy and composition, given the extensive research on capturing
> > >
> > > * composition (Bordes et al., 2013; Sun et al., 2019; Zhang et al., 2019; Lu \& Hu, 2020; Gao et al., 2020), and
> > > * hierarchy (Yang et al., 2015a; Trouillon et al., 2016; Kazemi \& Poole, 2018; Abboud et al., 2020).
> > >
> > > Each of the mentioned works focuses on either of these aspects, identifying either composition or hierarchy individually as vital - over the same datasets - but the lack of models which jointly consider these two patterns was the core motivation of our work.
> > >
> > > **Empirical motivation**.
> > > Beyond this particular motivation, given the density of work on these topics, there is also a clear empirical motivation: As ExpressivE's performance gain of at least 3\% over state-of-the-art models on WN18RR in Table 3 shows, significant performance gains can be achieved by considering these patterns jointly.
> > >
> > > **Limitations**.
> > > Regarding your question about limitations - there are, of course, limitations to most of the works mentioned above, including our work: the support of multimodal knowledge graphs (including new forms of patterns which can be formed over such multimodel knowledge graphs) is an important aspect that has not seen as much coverage by our community as it should, including datasets such as LiterallyWikidata. We take the extension of ExpressivE to multimodal knowledge graphs as interesting future work, with a range of new challenges from the multi-modality of data.
> > >
> > > In response to your question, we have added a comment about why composition and hierarchy are vital patterns in the introduction. Furthermore, we have added the extension of ExpressivE to multimodal knowledge graphs as promising future work in the conclusion section.
> > >
> > > &nbsp;
> > >
> > > **Datasets suggested by the reviewer**
> > >
> > > We are thankful for the suggested literature and datasets provided by the reviewer.
> > > This point is interesting, as it is an effort that our community has to take that goes beyond what we can do in this rebuttal timeframe but is giving a general direction that we agree with the reviewer that we as a community should take:
> > >
> > > * CoDEx is a promising choice for future evaluations of our KGC model. To achieve a fair comparison, one would need to get not only performance results on our work but performance results for many of the considered state-of-the-art KGC models (such as BoxE, RotatE, and DistMult) - many of which have not been benchmarked against CoDEx yet. This critically applies to hyper-parameter optimization, not only for our model but also for some competing models to achieve a truly representative study, going clearly beyond the timeframe of this rebuttal and which would presumably be a research paper of its own.
> > >
> > > * The LiterallyWikidata dataset focuses on multimodal knowledge graphs. As mentioned in the previous question, we think it is an interesting new set of challenges for our community that should form a new line of research for the next many years.
> > >
> > > Finally, we want to mention that WN18RR and FB15k-237 are currently the standard KGC benchmarks. Thus, the evaluation of Table 3 on both datasets provides results that are comparable across many different approaches, allowing ExpressivE to be put into the context of contemporary literature. Still, we are thankful for the reviewer's suggestions that we will keep in mind for our future work.

---

### Decision · Program_Chairs · 2023-01-20

**Decision:**

Accept: notable-top-25%

**Justification For Why Not Higher Score:**

The task of knowledge graph completion is a bit dated, and KGC is not an end task (e.g., Question Answering).
Only one task is considered, but other graph-related problems are not included.

**Justification For Why Not Lower Score:**

The paper brings a significant amount of new knowledge to KGC, and it should be introduced as a spotlight paper.

**Metareview: Summary, Strengths And Weaknesses:**

In this paper, the authors propose a novel solution to AI's classic problem of knowledge graph completion (KGC). Specifically, they use hyper-parallelograms and design a new approach to capture the comprehensive inference relations among entities and relations. The paper has accompanied theoretical analysis that adds to the technical depth of the approach. The empirical results perform well with only half of the parameters compared to some previous models. The authors have answered very well to some of the initial questions asked by reviewers, which resulted in a score increase. Overall, the AC believes this is a refreshing paper on KGC this year. A minor comment from the AC is that the paper is intrinsically related to "Holographic embeddings of knowledge graphs" and "Poincaré Embeddings for Learning Hierarchical Representations", and it will be useful to discuss these two papers and provide comparison during camera ready.

**Note From Pc:**

if the above contains the word "oral" or "spotlight" please see: "oral" presentation means -> notable-top-5% and "spotlight" means -> notable-top-25%. As stated in our emails, we are disassociating presentation type from AC recommendations

**Summary Of Ac-Reviewer Meeting:**

This is not a borderline paper, and no AC-reviewer meeting was scheduled.